# COGITAO: A PROCEDURAL OBJECT-CENTRIC FRAMEWORK TO EVALUATE COMPOSITIONAL GENERALIZATION

## ABSTRACT

The ability to compose learned concepts and apply them in novel settings is key to human intelligence, but remains a key challenge in state-of-the-art machine learning models. To address this issue, we introduce COGITAO, a modulable data-generation framework to evaluate compositional and systematic generalization in object-centric domains. Drawing inspiration from ARC-AGI's environment and problem-setting, COGITAO constructs rule-based tasks to be solved by applying a set of transformations to objects in grid-based environments. It supports composition over a set of 28 interoperable transformations, at adjustable composition-depth, along with extensive control over grid parametrization and object properties. This flexibility enables the creation of millions of unique task rules – surpassing existing datasets by several orders of magnitude – across a broad range of difficulties, while allowing virtually unlimited sample generation per rule. Alongside open-sourcing our flexible data-generation framework, we release benchmark datasets and provide baseline results with several state-of-the-art architectures that incorporate inductive biases well-suited for compositionality, such as diffusion-based Transformers (LLaDA) or recurrent Transformers with Adaptive Computation Time (Universal Transformer/PonderNet). Despite strong in-domain performance, these models consistently fail to generalize to novel combinations of familiar elements – highlighting a persistent challenge in compositional and systematic generalization, which COGITAO allows to precisely characterize.

## 1 INTRODUCTION

Compositional and systematic generalization are core principles of human cognition (1). From a few examples of 'atomic' concepts, humans can later effortlessly combine these in exponentially many new ways and apply them in contexts far removed from those in which they were learned. As Lake and Baroni illustrate, "Once a person learns the meaning of a new verb 'dax,' they can immediately understand the meaning of 'dax twice' or 'sing and dax'" (2). Machine learning systems still struggle with these forms of generalization, making them a central research challenge (3; 4; 5; 6; 7; 8; 9; 10).

To foster progress in this direction, several benchmarks have been proposed, both in language (2; 11; 7; 12; 13) and vision (14; 15; 16; 17; 18; 19). Yet in vision, existing benchmarks lack the flexibility of their language counterparts: they provide limited control over compositional structure, offer a narrow range of tasks, and conflate visual complexity with relational structure – distracting from the essence of compositional generalization.

To close this gap, we introduce **CO**mpositional **G**eneralization **I**n **T**ransformations **A**nd **O**bjects: **COGITAO** – a procedural object-centric framework designed to generate simple, controllable datasets to probe compositional and systematic generalization in an abstract visual domain. Our generator's strength stems from its capacity to compose freely, at arbitrary compositional depth, a set of 28 atomic transformations –

thus enabling to create millions of unique transformation sequences, with a virtually infinite amount of task samples, across a wide range of difficulties.

In our COGITAO framework, models can learn to apply a given sequence of object-transformations and generate an 'output grid' given an 'input grid' and a transformation sequence (see Figure 1). Furthermore, models can be implemented in various environment configurations (e.g., with different numbers or types of objects, or grid dimensions).

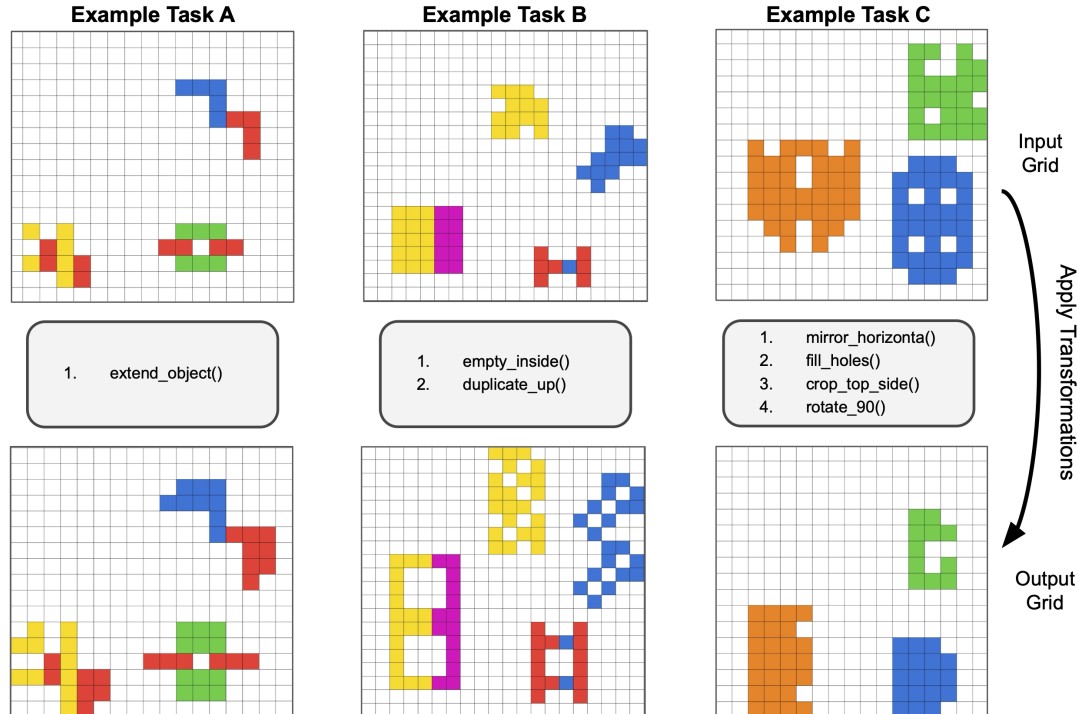

Figure 1: Set of input-output pair examples from our COGITAO generator, with input grids on top rows, and corresponding output grids (after transforming input) on the bottom rows. Each input-output pair follows a different transformation sequence (see boxes on middle row) and a given grid/object parametrization.

The key test of compositionality comes from varying transformation sequences and environment parameters between training and testing time. Analogous to the linguistic example "dax twice" or "sing and dax" (2), a model trained to "rotate" objects once should generalize to "rotate twice" or "rotate and translate," provided it has seen the individual rotate and translate transformations during training.

COGITAO is deliberately abstract and synthetic, which enables to precisely isolate core compositional reasoning from confounding visual complexity. Despite its simplicity and grid-based nature, it captures compositional and systematic generalization in ways essential for real-world vision tasks. Indeed, the object-centric, order-dependent operations exemplified in COGITAO– such as selectively modifying or relocating objects while preserving the surrounding scene – mirror the challenges studied in robotics (20) and world-model research (21; 22; 23; 24). We therefore position COGITAO not only as a static image benchmark, but also as an image-and-action framework that explicitly links perception to interaction (see Appendix A). To facilitate transfer to natural vision, COGITAO additionally offers RGB renderings of its tasks to train standard computer-vision models (see Appendix B), while preserving all its diagnostic power.

To demonstrate our framework's utility and support further research, we release a suite of benchmark datasets built from different COGITAO configurations. We train state-of-the-art architectures that incorporate inductive biases well suited for systematic and compositional generalization (Diffusion Transformers (25) and Pondering Looped Transformers (26; 27)). We also compare to baseline Transformers (optionally infused with object-centric inductive biases (28) and ResNet (29). Our experiments intentionally target the raw inductive biases of well-specified architectures rather than large foundation models, whose training data and precise design details often remain opaque [1]. Despite solid in-domain performance, we report that these models consistently fail to generalize to novel combinations of familiar visual elements. Our findings highlight the need for architectures that move beyond pattern matching and memorization (30; 31; 32; 33) towards truly structured, compositional understanding – COGITAOprovides a simple framework to foster progress in this direction.

In this paper, our main contributions are:

1. We introduce COGITAO: a procedural, grid-based and object-centric framework that freely composes 28 atomic object-transformations to generate millions of unique, controllable input–output rules at adjustable composition depth.

2. We extend COGITAO beyond simple grids to RGB renderings for real-world vision transfer (see appendix B), and from single input–output pairs to sequences of images and actions for world-model research (see appendix A).

3. We create and release multiple benchmark datasets targeting specific aspects of compositional and systematic generalization, enabling reproducible and scalable experimentation.

4. We provide cohesive baselines with state-of-the-art models known for their reasoning capabilities, and show that they consistently fail to generalize to out-of-distribution compositions — highlighting the open challenge of systematic compositional generalization in object-centric domains.

## 2   RELATED WORK

Our proposed framework builds upon prior research in compositional and systematic generalization – applied to both language and vision data. While these areas are often interconnected, we find that compositional generalization benchmarks in the visual domain lag behind their language counterpart in flexibility and scope. This is mainly due to the difficulty of procedurally generating visual data of adequate task fidelity compared to language-like data.

**Compositional Generalization in Language**   Several datasets and benchmarks have enabled targeted focus on systematic and compositional generalization in deep learning architectures for natural language processing. The SCAN dataset (2) is most akin to our work – it consists of commands mapped to action sequences. Models must compose commands, both within and outside the training distribution, to execute them correctly. Our dataset is similar to SCAN in that it contains various experimental settings in which the difference between training and testing distribution varies (across primitives, combinations, or sequence length). Several works have shown that models can achieve good performance on SCAN with specific architectures, representation methods, or data augmentation techniques (34; 35; 36; 37; 32), as well as different training strategies such as meta-learning (6), but there is still no consensus on the best architecture for compositional tasks. Analogous to Scan, COGS (12) or PCFG (13) also leverage a modular language to construct compositional tasks. In a similar but more formal spirit, the CFQ dataset (7) is designed to maximize compound divergence while minimizing atomic divergence between train and test sets, which the authors argue is an optimal setting

---

[1]We note that we also provide baseline foundation models results by adapting COGTIAO as an In-Context-Learning task for frontier LLMs, and highlight similar failure modes to models trained from scratch (see appendix C)

to study compositional generalization. Our approach follows this logic, as primitive elements are shared, but their combinations differ between training and test time. Other datasets reproduce this logic in other sequential modalities, such as within the SQL language (38) or mathematical reasoning (39).

**Compositional Generalization in Vision** Compositional generalization in vision is often studied in pairs with visual reasoning, which has gained significant traction in recent years. Most aligned to our work are benchmarks such as ARC-AGI (40), Raven's Progressive Matrices (41), and Procedurally Generated Matrices (16), which use handcrafted shapes and environments to simplify visual processing in light of reasoning. With similar handcrafted shapes and environments, but with a more targeted focus on compositional and systematic generalization as opposed to reasoning, we find other important datasets such as dSprites (42), CLEVR (14), Compositional Visual Reasoning (15), SVRT (17), and SVIB (43). These datasets are all built out of procedurally generated shapes with varying properties (e.g., color, size, texture) and make use of different types of rules, as well as composition of properties to evaluate models' compositional capabilities and object-centric representations - which can entail a different type of compositional generalization (5; 9; 10). Most aligned with our objectives are the CVR (15) and SYGAR (18) datasets. CVR targets compositional visual relations in an odd-one-out classification format, whereas SYGAR requires grid prediction but adopts the Meta-Learning (MLC) framework (6) and does not providing clear benchmarks. Both are more limited in scope and compositional and environmental control compared to COGITAO. Other datasets, such as CATER (19), contrary to the aforementioned datasets which use procedurally generated shapes, focus on compositional generalization in the real-world visual domain. However, we believe these inevitably conflate compositional reasoning ability with visual complexity, preventing a focused study on compositional generalization.

## 3 COGITAO GENERATOR

Our motivation stemmed from attempting to make progress at the original ARC challenge (40). Similar to the authors of conceptARC (44), we observed that the data regime of the ARC-AGI was too small (low data availability), too diverse (high variance between individual tasks) to make sound scientific progress on some of the abilities crucial to the challenge and lacking from modern deep learning approaches. While other researchers adopted the path of scale, big-data regime and novel inference-time methods such as test-time-training and chain of thoughts reasoning to tackle the challenge (45; 46; 47), we chose instead to follow a more principled approach, as this allows us to directly evaluate the capabilities of raw architectures and inductive biases to compose and generalize on basic sets of problems. As such, we designed a generator tailored to gauge the systematic and compositional generalization abilities of models, specifically through composing object-centric transformations. Our primary tasks consist of rule-based input-output pairs, where each rule is defined by applying a sequence of transformations to objects arranged on grids of variable sizes. This setup enables the creation of a wide spectrum of tasks with a large difficulty range. Our framework becomes valuable when varying the transformation sequences and environment parametrizations between training and testing sets – which is how we propose to gauge models' compositional. Specifically, we control systematic and compositional generalization along two axes:

- **Compositional Generalization**: COGITAO enables the composition of multiple object transformations at various levels of depth (i.e., the number of transformations applied sequentially) through a set of 28 primitive transformations (e.g., translations, rotations, mirroring).
- **Environmental Generalization**: COGITAO allows control over additional parameters to assess the generalization capabilities of models, i.e., their ability to apply learned transformations in varied environments. Users can modify the number of objects per grid, the complexity, size, and color of shapes, and the grid size, enabling the generation of different environment versions between training and testing phases.

Both axes of research are also enabled through our RGB rendering and sequential rendering, which are respectively introduced in Appendix B and A. While we view these two extensions as essential for future

development in natural vision and World-Model research, the main manuscript focuses primarily on the simpler, grid-based input-transformation-output framework.

## 3.1 GENERATOR OVERVIEW

**COGITAO Objects:** COGITAO samples from a collection of 23,000 pre-generated objects varying in size, shape, symmetry, connectivity, and color pattern. Each object uses pixel colors 1–9 (0 is reserved for the background) and is placed to avoid overlap and contact with other objects, ensuring clear boundaries between them (see D.1 for details).

**COGITAO Transformations:** We provide a set of 28 simple object-transformations such as translations, padding, duplicating, etc.; see D.2. Each transformation has been crafted to respect two core rules: (i) each transformation should be composable with all other transformations; (ii) each transformation should modify the object in a way that should not be systematically equivalent to a combination of other transformations. Rule 1 is critical to ensure that our generator maximizes the number of possible tasks that can be generated and avoids degenerate cases. Rule 2 avoids redundancy in the transformations (see Appendix D.2 for details). We ensure each transformation is individually learnable by all models used for training (see E for further details).

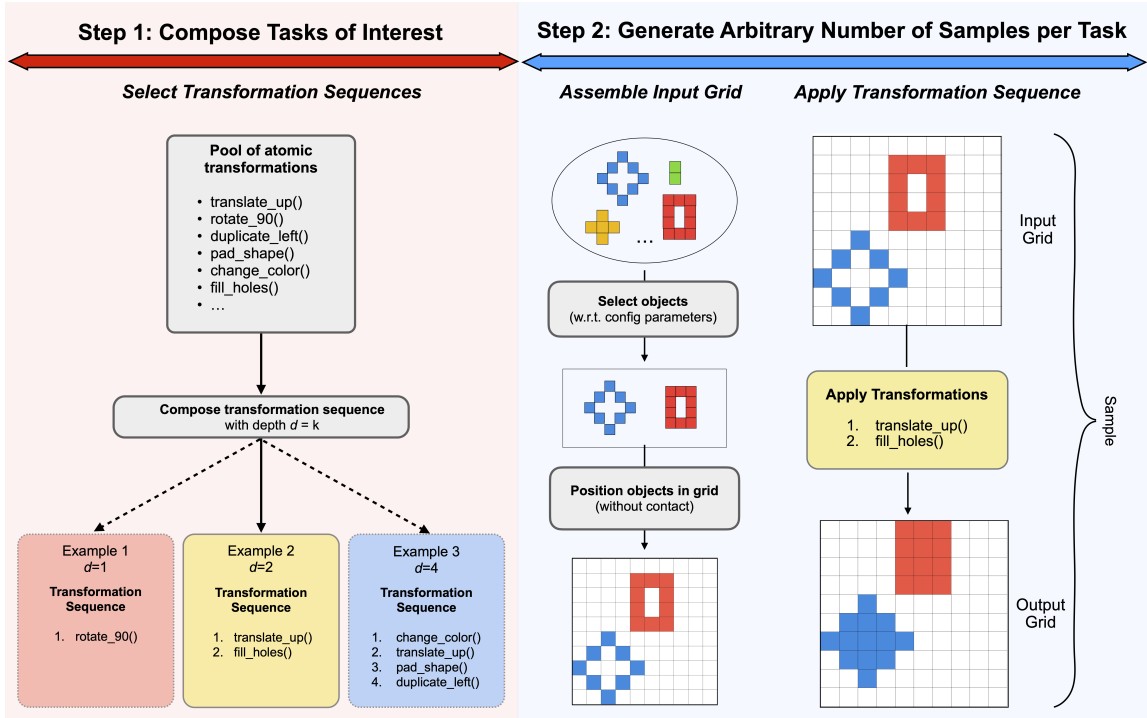

Figure 2: COGITAO is a Python-based procedural and object-centric data generator, inspired by the ARC-AGI (40) grid-based environment. COGITAO samples transformation sequences, then, given the configuration requested by the user and the sampled transformations, randomly samples and position objects in an input grid. Once objects are positioned in the input grid, each transformation is sequentially applied to each object.

**COGITAO Generation**    The generator can either randomly sample a sequence of transformations from a specified pool or use a user-defined sequence. After selecting the transformation sequence, it assembles random objects (w.r.t. config parameters) on the input grid, allowing each transformation to be applied to an output grid. Because some transformations (e.g., cropping) are irreversible and certain combinations are non-commutative, the order of transformations matters. With 28 available transformations and a maximum transformation depth $k$, the theoretical upper bound on the number of distinct tasks is $N = n_{\text{transforms}}^k = 28^k$. For example, at dept $d = 5$ and an adequate[2] number of objects and grid size, the generator could theoretically create $n_{\text{tasks}} = 28^5 \approx 1.7 \times 10^8$ unique tasks. Provided the grid is sufficiently large and generation time is not a limiting factor, the depth of compositional transformations can be arbitrarily large, making the space of possible tasks effectively unbounded. Further details on the generation method can be found in D.3.

## 4    EXPERIMENTS

To illustrate the utility of the COGITAO generator, we designed a set of benchmark experiments that highlight its potential for studying compositional and systematic generalization in the visual domain. These experiments define the *initial* benchmark for the community to beat, while also serving as illustrative examples—COGITAO can generate richer and more challenging tasks than those presented here. In the Discussion section, we outline additional configurations and encourage researchers to explore this broader task space once the current benchmark is mastered.

### 4.1    EXPERIMENT DETAILS

We outline two "studies" - one focusing on the composition of transformations, the **Compositional Generalization** *(CompGen)* study, and the other on environment variations (with fixed transformation sequence): the **Environmental Generalization** *(EnvGen)* study. In each case, we measure the capacity of models to perform tasks that differ in some way from the tasks seen during training—either with respect to transformation composition in the *CompGen* study or with respect to environment parametrization in the *EnvGen* study.

For both studies, we define different *experiment settings* which focus on different aspects of compositional and systematic generalization. For each *experiment setting* we create 5 "experiments": they are different instances of the *experiment settings* in which we only vary the transformation sequences. This is to ensure that the results we report are robust across different transformation combinations (e.g., translation-based transformation sequences might be easier than rotation-based ones; see Appendix E).

For each experiment, we train on 100,000 unique training samples, and test on two unique and distinct sets of 1,000 samples; one test-set follows the same distribution (i.e., in-domain (ID) set) as the training set, and the other is out-of-distribution (OOD), from which we evaluate generalization.

In the *CompGen* study, we train and test a model on multiple transformation sequences within a single experiment. This contrasts with the *EnvGen* study, where models are trained on only a single transformation per experiment. To handle this complexity, we provide context that indicates to the model which sequence to perform. We do so by appending a task embedding to the input sequence, similar to the approach in CVR (15). This embedding is a sequence of tokens that specifies the transformations in the correct order. For example, the task "translate_up-rotate_90" would have an embedding like ['T', 'R'], while the inverse task "rotate_90-translate_up" would be ['R', 'T']. For OOD testing, the models may be evaluated on new or longer sequences of these transformations. However, every individual token would have been seen during training.

We provide below an outline of each experiment setting for each of our two studies, and refer the reader to the Appendix F for more details.

---

[2]"Adequate" refers to grid and object configurations that avoid frequent generation failures—for instance, attempting to place ten objects in a $5 \times 5$ grid while performing four object transformations is impractical.

***CompGen* – Compositional Generalization**

- *C1 – Atomic + Composite → Unseen Composite*: Train on atomic (depth 1) and composite (depth 2) tasks; test on unseen depth 2 composites built out of the same atomic transformations as training.
- *C2 – Restricted Composite → Unseen Composite*: Train only on a subset of composites (depth 2); test on unseen depth 2 composites built out of the same atomic as training.
- *C3 – Atomic + Composite → Deeper Composite*: Train on atomic and depth 2 composites; test on depth 3 composites built out of the same atomics as training.

***EnvGen* – Environmental Generalization**

- *G1 – More Objects*: Train with 1–2 objects; test with 3–4 objects (fixed grid size and object complexity).
- *G2 – Larger Grids*: Train on 10×10–15×15 grids; test on 16×16–20×20 grids (fixed object number and complexity).
- *G3 – Larger Objects*: Train on objects sized 1×1–5×5; test on objects sized 6×6–10×10 (fixed grid size and number of objects).
- *G4 – More Complex Objects*: Train on symmetric, single-colored objects; test on asymmetric, multi-colored (fixed grid size and number of objects).
- *G5 – Combined*: Train on simple cases (1–2 objects, 10×10–15×15 grids, symmetric single-colored 1×1–5×5 objects); test on all harder variants simultaneously (3-4 objects, 16x16-20x20 grids, multi-colored asymmetric objects).

## 4.2    MODELS

We evaluate four encoder architectures, each paired with a two-layer MLP head for grid tokens classification. All models are of comparable size (approximately 1 million parameters) to ensure a fair evaluation. Empirically, increasing the model size did not substantially improve generalization.

- **Vanilla TF**: We use a standard Vanilla Transformer (TF)(48) with learned absolute positional encodings (49). Vision transformers are considered state-of-the-art for vision tasks and offer a strong baseline for COGITAO tasks.
- **Grid TF**: To better capture the structure of grid-based reasoning tasks, we introduce a Grid TF, an adapted version of ViTARC (28) that incorporates task-specific biases: object positional encoding (OPE) (28), PEMixer modules (28), register tokens (50), and modified positional encoding schemes (51; 52). These additions are designed to support spatial reasoning over structured grid-based data.
- **Pondering Looped TF (PL-TF)**: Based on PonderNet (26; 27), we evaluate a Transformer architecture that incorporates recurrence through iterative weight sharing and adaptive computation time. This architecture aims to reflect an inductive bias towards iterative reasoning and composition through multi-step transformations, making it particularly well suited for compositional tasks.
- **LLaDA**: In addition to vision-like and grid-specific architectures, as well as models with a recurrent architecture, we investigate the performance of language models. Specifically, we evaluate LLaDA (25), a diffusion-based language model yielding state-of-the-art performance on symbolic and logical tasks.

The selected models encompass a diverse set of architectural paradigms based on the current state-of-the-art in both vision and sequence modeling. This selection is designed to address the heterogeneous computational requirements of abstract visual reasoning tasks and spans a wide spectrum of modeling characteristics, thus providing baselines for COGITAO. Comprehensive architectural details are provided in Appendix I.

### 4.3 TRAINING

All models are trained[3] from scratch using supervised learning. For each experiment, we evaluate and test in-domain (ID) and out-of-domain (OOD) data. We aim to train all models as similar as possible for a fairer comparison among them. The models are trained for 10 epochs on the 100K samples (except for LLaDA, which is trained for 20 epochs, as on average 50% of the tokens are masked). We found that neither increasing the number of training steps, nor the number of unique samples improved generalization performance (see Appendix H for more details).

All models are trained using the AdamW optimizer (53) with a linear warm-up for 200 steps, followed by cosine annealing of the learning rate. We use a batch size of 64 for training and 50 for evaluation and testing. Further training details for all models are provided in Appendix J.

### 4.4 EXPERIMENT RESULTS

To provide robust empirical baselines for the COGITAO benchmark and to systematically evaluate the compositional and generalization capabilities of different model architectures, we evaluate the models on all the experiments of the three *CompGen* settings (*C1–C3*) and the five *EnvGen* settings (*G1–G5*) described in Section 4.1.

Table 1 provides an overview of the results across these settings, reporting both in-domain (ID) and out-of-domain (OOD) grid accuracy. Grid accuracy is defined as the percentage of samples for which the predicted grid structure matches the ground truth exactly (i.e., the entire grid is predicted correctly). We present further details on metrics in Appendix G. Performance values are averaged across 5 variations of the experiment setting (i.e., different transformation sequences; see F for more details) with 3 seeds each.

| ES | Vanilla-TF | | | Grid-TF | | | PL-TF | | | LLaDA | | |
|---|---|---|---|---|---|---|---|---|---|---|---|---|
| | ID | OOD | Δ | ID | OOD | Δ | ID | OOD | Δ | ID | OOD | Δ |
| C1 | 16.5 | 0.0 | 16.5 | 59.8 | 0.0 | 59.8 | **81.0** | **0.1** | 80.9 | 44.9 | 0.0 | 44.9 |
| C2 | 17.8 | 0.0 | 17.8 | 68.5 | 0.0 | 68.5 | **79.0** | **0.1** | 78.9 | 46.2 | 0.0 | 46.2 |
| C3 | 29.4 | 4.0 | 25.4 | 63.3 | **8.3** | 55.0 | 82.0 | 7.2 | 74.8 | **84.1** | 7.8 | 76.3 |
| G1 | 98.4 | 78.9 | 19.5 | 99.3 | **90.1** | 9.2 | 93.8 | 85.5 | 8.3 | **99.5** | 90.1 | 9.4 |
| G2 | 82.0 | 2.1 | 79.9 | 98.0 | **77.0** | 21.0 | 92.6 | 54.9 | 37.6 | **98.5** | 62.4 | 36.1 |
| G3 | 57.6 | 22.5 | 35.1 | 85.0 | 26.6 | 58.4 | **92.0** | **27.2** | 64.8 | 82.4 | 26.8 | 55.6 |
| G4 | 46.2 | 21.8 | 24.4 | **86.9** | 19.0 | 67.9 | 86.8 | **37.5** | 49.3 | 81.3 | 32.0 | 49.3 |
| G5 | 72.5 | 0.0 | 72.5 | **95.0** | 0.2 | 94.8 | 80.7 | 8.9 | 71.8 | 70.0 | **10.1** | 60.0 |

Table 1: Performance of models on experiments across experiment settings (ES) of the main studies. We report the ID (in-domain) and OOD (out-of-domain) results for test grid accuracy (i.e., % of perfect matches) and the ID to OOD relative drop Δ averaged over all experiments within the experiment setting.

The experimental results reveal several key trends (Table 1). Grid-TF, our grid-specialized transformer, provides strong in-domain (ID) accuracy across most settings and remains the most stable baseline overall - particularly in the *EnvGen* study. However, the PL-TF model consistently matches or surpasses Grid-TF in several critical out-of-domain (OOD) tests. In the *CompGen* study, PL-TF attains the best ID accuracy in C1 and C2 and competitive OOD performance in C3. It is the only model that solves a task on C1 and

---

[3]Training of the Vanilla and Grid-TF models was performed using an NVIDIA GeForce RTX 3090 GPU. Traiing of PL-TF was performed on NVIDIA A100 GPU. Training of LLaDA used an NVIDIA V100 GPU.

C2 OOD. Within the *EnvGen* settings, PL-TF delivers the best OOD scores in G3 (27.2) and G4 (37.5), and strong ID results in G3–G4, indicating improved robustness to object scale and complexity changes. LLaDA remains a strong performer, achieving the highest OOD accuracy in G1 and competitive results in G2 and G5,. Vanilla-TF, by contrast, performs pooly, with sharp ID–OOD drops across nearly all tasks. Overall, these findings underscore both the difficulty of the COGITAO benchmark and the promise of PL-TF's architectural choices for compositional and environmental generalization, while confirming that no current model fully solves the challenge.

Our complementary analysis reveal two important failure modes for which do not generalize.

- **ID Bias:** The model applies the transformation it observes during training (e.g., translate_right) even when the OOD task specifies another transformation (translate_up).
- **Structural Composition Failure:** When moving from depth-1 compositions to depth-2 (CompGen Setting 3), models are able to apply the transformation representing a sequence of atomic transformations–highlighted by a high ID performance–but they fail to decompose such sequences to extract their atomic transformations and recompose them as expected for the OOD case; this again highlights a concrete failure in compositional generalization ability.

We refer the reader to our appendix H, where we present further analysis that sheds light on these issues.

## 5 DISCUSSION

**Summary** COGITAO offers a controlled and targeted framework for studying compositional and systematic generalization, specifically in object-centric environments. With millions of possible tasks and abundant training samples, COGITAO provides an unmatched degree of compositional control in abstract visual domains. By leveraging simple grid-based environments and object-centric transformations, it enables isolating core compositional capabilities while avoiding the visual and data complexity of more naturalistic datasets. Indeed, our core hypothesis is that models that cannot exhibit compositional generalization in this controlled, simple and abstract environment are unlikely to succeed in the far noisier and less structured settings of real-world vision. Nevertheless, to bridge the gap to real-world vision, we provide an RGB extension to COGITAO onto which both the *CompGen* and *EnvGen* benchmarks can be applied.

Unlike classification-based benchmarks such as CVR (15), RPM (41), SVRT (17), and PGM (16), our generator requires models to *generate* output grids - a real test of compositional understanding. It aligns more closely with challenges like ARC-AGI (40) and its successors (54; 18; 44; 55). We also extend our input-output generation framework to a sequential framework (Appendix A), with set of frames and object-centric "actions", thus making it highly relevant sequential manipulation tasks.

In our benchmark experiments, we evaluate Vanilla TF, Grid TF, Pondering Looped TF, and LLaDA across the COGITAO tasks. These models were chosen to represent the state-of-the-art in vision and sequence modeling, including both general-purpose and grid-specialized architectures. While they provide a diverse and strong baseline, future work should explore additional architectures. The experiments reveal a consistent trend: while these models perform well on in-domain tasks, they fail dramatically in out-of-distribution scenarios requiring compositional understanding. In our *CompGen* study, while models "learned to learn" ID transformations sequences, they all failed when faced with unseen OOD transformation sequences. Similarly, *EnvGen* performance degrades significantly with increased task complexity. These results support growing evidence that current state-of-the-art sequence and vision models rely heavily on pattern recognition rather than systematic compositional reasoning (31). COGITAO provides a controlled environment to diagnose these fundamental limitations and guide the development of more robust, generalizable architectures and inductive biases.

**Future Work**  COGITAO offers a compelling platform for driving progress in compositional generalization, and it supports several promising research directions. First, extending the framework to *in-context learning*, for example, by providing demonstration examples, could allow evaluation of generalization in settings known to benefit from longer contexts (46), particularly in large foundation models (56; 57; 58). Second, due to its controllable environment and adjustable difficulty, COGITAO is well-suited for *curriculum learning,* where task complexity is gradually increased to guide learning. Third, analyzing *internal model representations* trained on COGITAO may reveal whether and how models develop object-centric or transformation-centric abstractions.

**Outlook**  Mastering the existing hardest COGITAO settings would be a significant milestone toward genuinely compositional architectures. Furthermore, doing so using only generator-produced data would show that a model can identify primitive operations, apply them sequentially, and recombine them in novel settings—akin to a core human cognition (1).

## 6 REPRODUCIBILITY STATEMENT

We have made every effort to ensure that our results are fully reproducible. Detailed descriptions of the COGITAO generator, object creation process, and transformation suite are provided in Sections 3–4 of the main paper, with additional implementation details, algorithmic pseudo-code, and full experimental settings in Appendix 7–9. All training hyper-parameters, model architectures, and evaluation procedures are explicitly specified (Sections 4.1–4.4 and Appendix 8–9). To facilitate independent verification, we release the complete source code and data generation framework as anonymous supplementary material, with the same version available at our GitHub repository [4]. The benchmark datasets used in our experiments can be reproduced or directly accessed via HuggingFace. These resources include scripts to regenerate all figures, tables, and experimental results reported in the paper.

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

## A    SEQUENTIAL-COGITAO

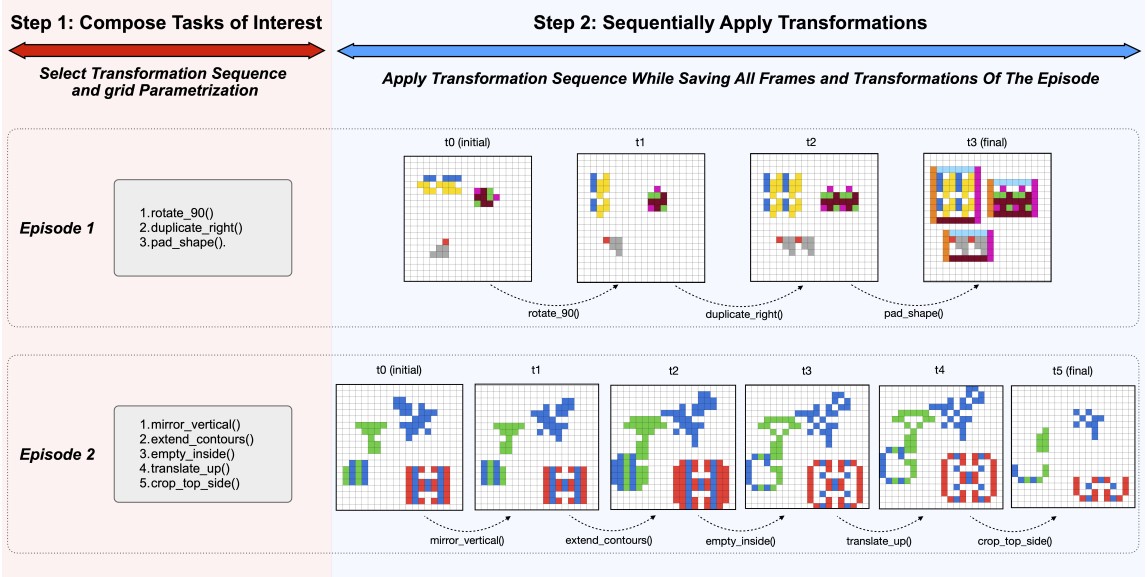

Figure 3: Overview of Sequential-COGITAO. Episodes of are generated from an initially sampled transformation sequence and grid parametrization. Each individual frame is saved, along with its corresponding transformations. We note that transitions for each individual objects are also available, but not shown here for visualization constraints.

Standard COGITAO tasks present a single input grid and expect a model to generate the final output after a composed transformation sequence. Sequential COGITAO extends this by exposing every intermediate state in the transformation chain. Instead of training solely on start–end pairs, models must either predict each intermediate grid or reason over an explicit temporal trajectory of transformations. This is particularly relevant for World Model research (21), where applying action to objects while preserving the environment is of particular importance (22; 23; 24). Both *CompGen* and *EnvGen* methodologies can seamlessly be applied to Sequential-COGITAO (also in a RGB rendering; see B), enabling targeted and compositionality-focused research on perception and interaction tasks—a needed addition to the field.

## B   RGB RENDERING OF COGITAO

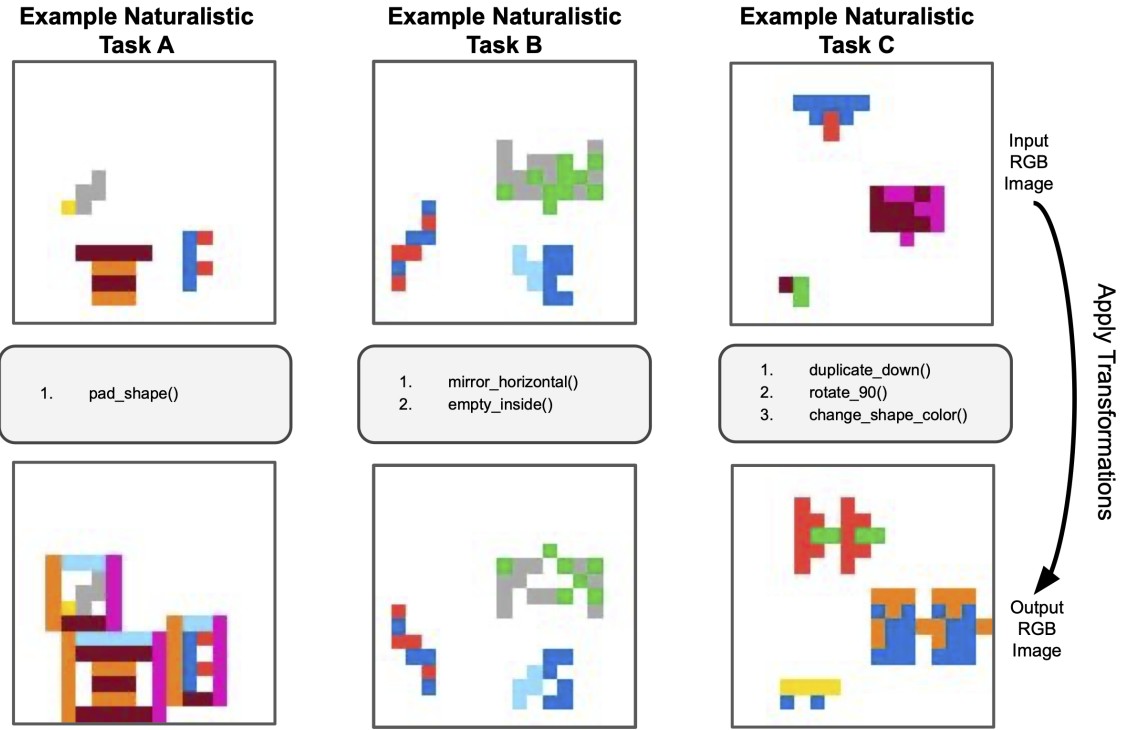

Figure 4: Set of input-output pair examples from our RGB rendering of COGITAO with input *images* on top rows, and corresponding output *images* (after transforming input) on the bottom rows. The images are 128x128x3, saved as .jpeg. Gray borders are added for visualization purposes only - they are not present on the images. Note: Objects are purposely blurry to outline their RGB nature - crisper rendering are straightforward.

To assess how compositional generalization persists beyond synthetic grids, we introduce COGITAO-RGB, a rendering of the benchmark in standard RGB images. Each sample is a $128 \times 128$ RGB image (with configurable resolution) in which the discrete grid visualization is removed and objects are drawn on a plain white (or black) background. We convert the benchmark datasets presented in 4.1 to this format, excluding G1-2 and G1-5 where variable grid sizes are essential; G1-1, G1-3, and G1-4 translate directly. All 28 transformations from §D.2 remain fully applicable, but models must now infer object location and scale without explicit grid cues, increasing perceptual difficulty. Generation and train/test splits follow the original procedure. COGITAO-RGB thus bridges abstract grid worlds and natural image statistics, enabling researchers to test whether compositional and systematic generalization tested in the grid setting carries over to more realistic visual conditions.

# C IN-CONTEXT LEARNING OF COGITAO WITH FRONTIER MODELS

We conducted supplementary experiments to evaluate how frontier LLMs performed on tasks akin to COGITAO's various experiment settings. We remind the reader that COGITAO's goal is to evaluate how models solve task *in light of a specific training distribution*, which is unfortunately not possible with frontier models given the opacity of the exact training data as well the lack of information on specific aspects of models. However, we believe interesting to evaluate how these models perform on tasks similar to COGITAO. As such, we have conducted In-Context Learning (ICL) experiments on frontier models to provide a reference baseline.

## C.1 EXPERIMENTAL SETUP

We prompted frontier LLMs in different conditions which resemble the various experiment settings we have. Specifically, we gave 5 in-context examples to various LLMs, as well as an input example that it must apply the novel transformation on. We do so according to the following ID/OOD control splits:

- **CompGen-ID1:** context built of a set of 5 transformations (simple + composed). Test on one of the composed ones seen during training.
- **CompGen-ID2:** context built of a set of 5 transformations (composed only). Test on one of the composed ones seen during training.
- **CompGen-OOD1:** context with a set of 5 transformations (simple + composed). Test on a composition not seen during training.
- **CompGen-OOD2:** context built of a set of 5 transformations (composed only). Test on a composition not seen during training.
- **CompGen-OOD3:** context built of a set of 5 transformations (simple + composed). Test on a deeper compositions not seen during training.
- **EnvGen-Object:** context built of 5 examples with the same simple transformation and 1-2 objects. Test on the same transformation but 3-4 objects.
- **EnvGen-Grid:** context built of 5 examples with 10x10–15x15 grids. Test on the same transformation but 16x16–20x20 grids.

We followed the same transformation combinations (5 transformation combinations) as presented in our main experiments (see F), with 10 unique examples per experiment. As such, there were $5 \times 10 = 50$ different unique prompts per setting. Each prompt was only sent to models once due to cost constraints [5].

### C.1.1 PROMPTING

We tested two prompting regimes:

**Explicit:** Task codes reveal the transformation name (e.g., ["translate_up", "rotate90"]).

**Coded:** Task codes are abstract (e.g., ["t1", "t2"]), requiring the model to infer the rule from context.

To account for the differences between the different experiment settings, we crafted 4 different "base prompts" - all inspired by previous work on ARC-AGI grids and LLMs (45). Each "base prompt" was used for its relevant "category" of experiment setting; we present these in details, as well as a full example prompt in L

---

[5]We ran all experiments through the OpenRouter API for a total cost of 314 USD.

## C.2 RESULTS

### C.2.1 PERFECT MATCH RATE WITH "EXPLICIT" TASK EMBEDDING

|              | deepseek-r1t2 | openai gpt-4.1-mini | openai o3 | x-ai/grok-code-fast-1 | gemini-3-pro-preview |
|--------------|---------------|---------------------|-----------|------------------------|----------------------|
| CompGen-ID1  | 8 / 50        | 1 / 50              | 19 / 50   | 7 / 48                 | 39 / 50              |
| CompGen-ID2  | 10 / 50       | 5 / 50              | 18 / 50   | 11 / 50                | 35 / 50              |
| CompGen-OOD1 | 0 / 50        | 0 / 50              | 4 / 50    | 0 / 47                 | 14 / 50              |
| CompGen-OOD2 | 0 / 50        | 0 / 50              | 0 / 50    | 0 / 48                 | 11 / 50              |
| CompGen-OOD3 | 1 / 50        | 0 / 49              | 6 / 50    | 2 / 50                 | 24 / 50              |
| envGen-size  | 10 / 50       | 0 / 49              | 19 / 50   | 6 / 50                 | 34 / 50              |
| envGen-objects | 8 / 50      | 1 / 50              | 13 / 50   | 7 / 48                 | 33 / 50              |

### C.2.2 PERFECT MATCH RATE WITH "CODED" TASK EMBEDDING

|              | deepseek-r1t2 | openai gpt-4.1-mini | openai o3 | x-ai/grok-code-fast-1 | gemini-3-pro-preview |
|--------------|---------------|---------------------|-----------|------------------------|----------------------|
| CompGen-ID1  | 5 / 50        | 1 / 50              | 13 / 50   | 1 / 49                 | 28 / 50              |
| CompGen-ID2  | 11 / 49       | 2 / 50              | 20 / 50   | 5 / 49                 | 27 / 50              |
| CompGen-OOD1 | 0 / 50        | 0 / 50              | 8 / 50    | 0 / 46                 | 18 / 50              |
| CompGen-OOD2 | 0 / 50        | 0 / 50              | 1 / 50    | 0 / 47                 | 4 / 49               |
| CompGen-OOD3 | 1 / 49        | 0 / 50              | 13 / 50   | 0 / 48                 | 15 / 50              |
| envGen-size  | 13 / 50       | 1 / 50              | 24 / 50   | 5 / 45                 | 36 / 49              |
| envGen-objects | 13 / 50     | 4 / 50              | 15 / 50   | 5 / 49                 | 39 / 50              |

### C.2.3 INTERPRETATION

Even state-of-the-art models like Gemini 3 Pro and OpenAI o3 exhibit the exact failure mode COGITAO highlights. While they achieve reasonable performance on In-Distribution (ID) tasks and Environmental Generalization (size/objects), their performance drops precipitously on Out-of-Distribution (OOD) Composition. This confirms that COGITAO is not merely solved by scale; it exposes a fundamental reasoning gap in how current architectures (even frontier ones) decompose and recompose information.

We also note that the models do significantly better when given explicit task codes as opposed to coded ones, which underlies that trained models perform best when leverage other internal biases from training data to solve these tasks.

# D    Further Details on COGITAO Core Generator

## D.1    COGITAO Objects

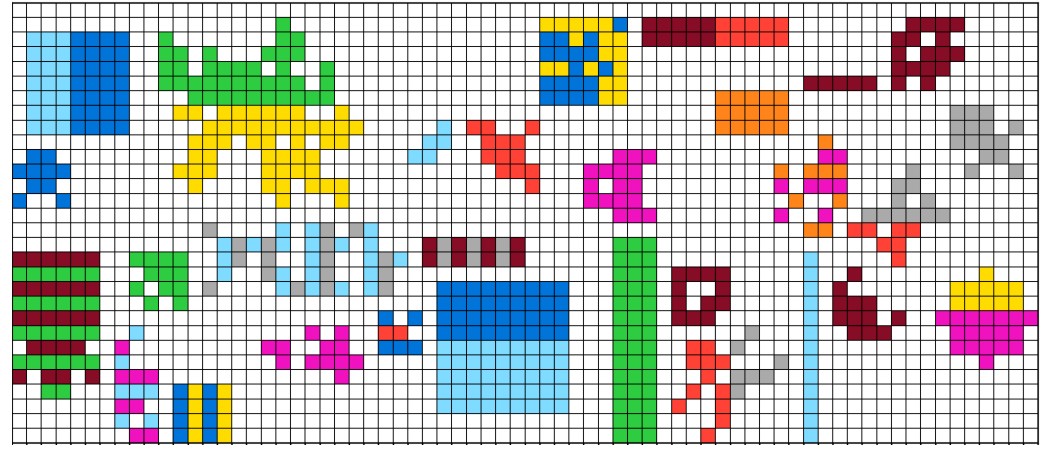

Figure 5: Random example of generated objects. The objects are generated with a variety of properties, including size, symmetry, connectivity, colors, color patterns, and footprints. Note: objects are not allowed to overlap, touch, or be inside one another in our generator environment, as reflected in the above image.

Rather than generating objects on the fly, which can be more costly computationally, we favor a method where we pre-generate a large set of 23,000 objects, and compute a table of properties that these objects satisfy. This allows us to efficiently sort through objects at generation time with respect to the constraints specified by the user, as well as the object constraints that the transformation suite may impose. We share the file with these objects along with our code publication, as well as the code used to generate these, so the community can expand the number of objects at will. Below are all the properties through which we iterate, creating combinations of every single parameter, up until a maximum object dimension of 15×15 pixels.

- **Size:** Number of rows, columns, and pixels.
- **Symmetry**: Horizontal, vertical, diagonal, point, and no symmetry.
- **Connectivity**: "4 connected" (only connected through adjacent edges), "8 connected" (only connected through adjacent or diagonal edges), or even "distance" (object can be composed of unconnected blocks).
- **Colors**: Single colored or multi-colored.
- **Color Pattern**: Uniform (single color), column stripes, row stripes, diagonal stripes, top-bottom coloring (object split in two colors), right-left coloring (object split in two colors), or random.
- **Footprints**: Predefined objects such as rectangle, disk, square, diamond, or ellipse.

## D.2    COGITAO Transformations

We provide a set of 28 object-transformations, which the community can further expand. As noted in the main manuscript, each transformation was chosen to respect two core rules: (1) each transformation should be composable with all other transformations; and (2) each transformation should modify the object in a way that should not be systematically equivalent to a combination of other transformations. Rule 1 is critical to ensure that our generator maximizes the number of possible tasks that can be generated and

avoids degenerate cases. Rule 2 avoids redundancy in the transformations - for instance, we could have implemented an individual transformation `translate_up_right`, but this would always be equivalent to the transformation sequence `translate_up` and `translate_right`. Rule 2, however, doesn't imply that each transformation suite *necessarily* yields a unique output grid that could not have been reached with another transformation suite. For instance, for symmetric objects, applying mirror transformations could yield the same output objects as applying the rotation transformations twice - this is not the case for non-symmetric objects, thereby still satisfying Rule 2. We refer the reader to Appendix E for an evaluation of the learnibility of each of the transformations.

We summarize below all 10 families of transformations available in the generator, and a description of the available variations.

- **Translate** (Up, Down, Left, and Right): Translates the entire object by one pixel in any of the 4 dimensions.
- **Mirror** (Horizontal and Vertical): Mirrors the object with respect to the vertical or the horizontal symmetry axis.
- **Rotate** (90 degrees): Rotates the object by 90 degrees.
- **Crop** (Top Side, Bottom Side, Right Side, Left Side, Contours): Crops the object's specified side(s).
- **Change Color** (`mod` $9 + 1$): Changes object color to `new_color = original_color` `mod` $9 + 1$ (e.g., if the object's color is 7, it is changed to the color 8).
- **Fill** (Same Color, Different Color): Fills object holes with its color, or the `mod` $9 + 1$ color.
- **Empty**: Empties inside of the object, and leaves contours.
- **Extend** (Same Color, Different Color): Extends the outermost edges of the object with either the same color or (`mod` $9 + 1$) color.
- **Pad** (Top, Bottom, Left, Right, Full object): Pads in the specified direction with a fixed color.
- **Duplicate** (Top, Bottom, Left, Right, Quadruple): Duplicates the object in the specified direction.

**Note regarding the `mod` $9 + 1$:** Object colors take values $c \in \{1, \ldots, 9\}$, with $c = 0$ reserved for **background**. The color-change operation `mod` $9 + 1$ is defined as a cyclic increment over the nine object colors:

$$c' = \begin{cases} ((c - 1) \bmod 9) + 1, & c \in \{1, \ldots, 9\} \\ 0, & c = 0. \end{cases}$$

This rule increments each object color by one in a cyclic fashion (sending $9 \mapsto 1$) while ensuring that the background value remains unchanged. Every transformation that involves a color change (e.g. `change_color`, `fill`, `extend`, `pad`) follows this rule - which is fully deterministic and easily learnable by models.

### D.3 COGITAO GENERATION ALGORITHM

Below is an outline pseudocode of the COGITAO generation framework.

The `sampleTransformations()` function simply iterates through the pool of transformations at the desired depth $d$ to create the transformation sequence. An important function of our algorithm is `setInitialGrid()`, which sets up the randomly sampled objects in the grid given the init_params (e.g. number of objects, desired object properties) and the sampled transformation sequence. Indeed, each transformation is defined with a set of object constraints, which are object properties for which the transformation can be applied without ambiguity. This function thus ensures that the random sampling of objects aligns with the sampled transformation sequence to avoid unexpected errors. The `setInitialGrid()` function positions objects to keep them fully within the grid and to avoid contact with other objects. Finally, the `transformAndPosition` function applies the transformations to objects and positions them back to the

---

**Algorithm 1** `GenerateCogitaoTask(init_params)` - Below is a simplified pseudo-code for the general algorithmic logic of the `COGITAO` generator: Generator

---

1: **for** d in transformation_depth **do**
2:     transformation_suite ← `sampleTransformations`(init_params, possible_transformations)
3: **while** trial_n < max_trials & example_n ≤ wanted_examples **do**
4:     input_grid, objects ← `setInitialGrid`(init_params, transform)
5:     output_grid ← input_grid
6:     **for** object in objects **do**
7:         **for** transformation in transformation_suite **do**
8:             output_grid ← `transformAndPosition`(output_grid, object, transformation)
9:     task ← (input_grid, output_grid, transformation_suite)
10: **Return:** task, transform

---

grid. Importantly, this function verifies that the transformed objects are still fully within grid dimension and do not collide with other objects (although adjacent contact is allowed at the transformation stage). If contact occurs, or objects go out of bounds, the entire input-output pair is discarded and generation is restarted.

For a standard 20x20 grid, with 4 objects (smaller than 6x6) and a sequence of 2 transformations applied, the average input-output pair generation time is at $0.005s \pm 0.002s$ [6]. Some configurations are more difficult and therefore more time-consuming to generate. For instance, decreasing the grid size and increasing the number of objects can significantly increase the generation time. Some transformation combinations are also more challenging to generate, such as series of object duplications which quickly yield objects too large for the grids.

---

[6]These figures were obtained on an Ubuntu 22.04.5 LTS machine equipped with two AMD EPYC 7742 64-Core Processors (128 cores total) running at up to 2.25 GHz, and 256 GB of system RAM

# E COGITAO TRANSFORMATIONS LEARNABILITY

To assess the learnability of each transformation introduced, we performed experiments to evaluate how effectively the models in the paper learn to apply transformations to new objects. This was conducted through a "Sample Efficiency" study, where models were given varying amounts of training data to determine the data required to master each transformation. This approach provides direct insights into 1) how easily each model learns each transformation and 2) what models are more efficient at learning given transformations. The latter idea is useful to evaluate the inductive biases of models.

For this purpose, we designed four experimental settings with different numbers of distinct training samples: 100, 1,000, 10,000, and 100,000 examples. These sample sizes were selected based on the known sample inefficiency of Transformer models. While 100 examples are generally insufficient for Transformers to learn effectively—a limitation not observed in humans—100,000 examples approach a scale where Transformers typically achieve competitive performance.

Each setting involves the same set of tasks, with each task corresponding to an elementary or atomic transformation from one of 10 distinct transformation families (e.g., the `translate_up` task represents the "translation" family, which also includes `translate_down`, `translate_left`, etc.). We conducted 10 experiments per setting, each focusing on a single transformation family, resulting in a comprehensive evaluation across all families. We train for 10 epochs on all experiment settings. The environment for all experiments was standardized: a fixed $15 \times 15$ grid with two objects per grid, each no larger than $6 \times 6$. For evaluation, we tested each model on 1,000 unseen samples of the same transformation used during training, maintaining consistent environment and object parameters.

The experiments are organized into settings labeled S-x-i, where $i$ denotes the specific transformation and $x$ indicates the number of training samples (e.g., S-1-i for 100 samples, S-2-i for 1,000 samples, S-3-i for 10,000 samples, and S-4-i for 100,000 samples). In the below outline, we keep $x$ to denote the varying experiment setting (and number of training samples). Below, we outline the experimental setup for the sample-efficiency study.

## E.1 SAMPLE-EFFICIENCY EXPERIMENTS

- S-x-1: Train and test on the atomic transformation `translate_up` as a proxy for all `translate` transformation family.
- S-x-2: Train and test on the atomic transformation `rot_90` as a proxy for all `rotate` transformation family.
- S-x-3: Train and test on the atomic transformation `mirror_horizontal` as a proxy for all `mirror` transformation family.
- S-x-4: Train and test on the atomic transformation `extend_contours_different_color` as a proxy for all `extend` transformation family.
- S-x-5: Train and test on the atomic transformation `empty_inside_pixels` as a proxy for all `empty` transformation family.
- S-x-6: Train and test on the atomic transformation `crop_top_side` as a proxy for all `crop` transformation family.
- S-x-7: Train and test on the atomic transformation `fill_holes_different_color` as a proxy for all `fill` transformation family.
- S-x-8: Train and test on the atomic transformation `double_up` as a proxy for all `duplicate` transformation family.
- S-x-9: Train and test on the atomic transformation `change_shape_color` as a proxy for all `change_color` transformation family.

- S-x-10: Train and test on the atomic transformation `pad_shape` as a proxy for all `pad` transformation family.

For a full list of transformations, we refer the reader back to section D.2.

In the most constrained setting (S1), all models, including the more sophisticated Transformer-based ones, completely fail to learn and generalize in-domain. All reported test accuracies being 0.0 suggests that it is likely that none of the models possess the ability to truly conceptualize simple spatial transformations in a data-scarce setting; this is unlike humans, who are able to generalize from only a few COGITAO examples. This result may reinforce the well-known *sample-inefficiency* of Deep Learning approaches, and more specifically of Transformers, as they appear to continue to rely on statistical pattern matching, which cannot easily emerge from training on only 100 samples while testing on 1000 samples of high (in-domain) diversity.

In S2, we begin to observe a divergence in model performance. The models imbued with strong inductive biases greatly increase their performance with more than an additional 50% accuracy increase. The Vanilla-TF, possessing only a weak inductive bias, achieves only 11.2%. The worse performance (19.9%) of LLaDA compared to the other Transformer-based models is possibly due to nature of its diffusion mechanism as well as the lack of a stronger inductive bias such as through the use of visual tokens.

In S3, the next order of magnitude of 10'000 training examples, all models show a marked improvement. ResNet reaches 96.1%, and Grid-TF achieves 97.1%. The PL-TF model shows the beginning of decrease in improvement at 88.2%, while LLaDA shows a significant gain at 66.5%. Interestingly, Vanilla-TF also catches up (59.3%), but still lags behind models with stronger inductive biases. This narrowing of the performance gap indicates that Transformer-based models being to learn well past some threshold of amount of data, at which point they begin to robustly extract structures and create useful representations from symbolic grids. The fact that the Grid-TF model outperforms even the ResNet model at this scale–typically considered far smaller than Transformer data regimes–suggests a benefit of Transformer models enhanced with grid-aware tokens and relational inductive biases through object and relative positional encodings for abstract visual reasoning tasks.

In the highest data regime S4, all models reach high performance. Grid-TF nearly saturates accuracy at 99.9%, while the simpler baseline models. ResNet and Vanilla-TF, perform even better than the PL-TF and LLaDA. The greater complexity of the learning mechanism of the PL-TF and LLaDA models may partly explain why they slightly underperform compared to the other models. Moreover, the results confirm that Transformer models, even in their vanilla form, can ultimately learn atomic visual transformations when given larger-scale data.

The Sample-Efficiency table in E.1 with all the experiments also shows that some atomic transformations are more difficult to learn than others. For instance, a translation is efficiently learned while a rotation or an extension of contours appears more difficult for all the models.

Overall, the experiments demonstrate that all models are capable of learning all the atomic transformations currently offered by COGITAO. This implies that the individual tasks are not a limitation for the more involved and complex tasks of the experiments part of the other two studies of Systematic Generalization and Compositionality.

| Setting | Transf. Family | ResNet ID | Vanilla-TF ID | Grid-TF ID | PL-TF ID | LLaDA ID |
|---------|----------------|-----------|---------------|------------|----------|----------|
| **S1** | translate | 0.0 | 0.0 | 0.0 | 0.0 | 0.0 |
| | rotate | 0.0 | 0.0 | 0.0 | 0.0 | 0.0 |
| | mirror | 0.0 | 0.0 | 0.0 | 0.0 | 0.0 |
| | extend | 0.0 | 0.0 | 0.0 | 0.0 | 0.0 |
| | empty | 0.0 | 0.0 | 0.0 | 0.0 | 0.0 |
| | crop | 0.0 | 0.0 | 0.0 | 0.0 | 0.0 |
| | fill | 0.0 | 0.0 | 0.0 | 0.0 | 0.0 |
| | duplicate | 0.0 | 0.0 | 0.0 | 0.0 | 0.0 |
| | change_color | 0.0 | 0.0 | 0.0 | 0.0 | 0.0 |
| | pad | 0.0 | 0.0 | 0.0 | 0.0 | 0.0 |
| **S2** | translate | 100.0 | 0.0 | 100.0 | 83.7 | 78.6 |
| | rotate | 5.7 | 0.0 | 0.0 | 0.0 | 0.0 |
| | mirror | 27.4 | 2.3 | 2.9 | 4.6 | 0.1 |
| | extend | 32.0 | 0.0 | 16.1 | 23.1 | 0.0 |
| | empty | 100.0 | 9.2 | 100.0 | 80.8 | 8.4 |
| | crop | 97.9 | 0.1 | 94.0 | 84.4 | 34.0 |
| | fill | 100.0 | 0.0 | 98.1 | 82.6 | 6.2 |
| | duplicate | 16.4 | 0.0 | 0.2 | 19.7 | 0.0 |
| | change_color | 100.0 | 100.0 | 100.0 | 100.0 | 79.1 |
| | pad | 97.1 | 0.0 | 95.3 | 89.0 | 0.0 |
| **S3** | translate | 100.0 | 100.0 | 100.0 | 100.0 | 99.9 |
| | rotate | 94.4 | 0.0 | 96.6 | 88.0 | 28.1 |
| | mirror | 98.3 | 3.8 | 97.0 | 82.6 | 50.8 |
| | extend | 95.8 | 36.6 | 80.3 | 79.9 | 28.5 |
| | empty | 100.0 | 91.2 | 100.0 | 99.5 | 97.7 |
| | crop | 99.6 | 90.5 | 98.8 | 89.0 | 95.9 |
| | fill | 100.0 | 67.8 | 99.9 | 79.2 | 94.9 |
| | duplicate | 72.9 | 13.5 | 98.6 | 84.8 | 65.1 |
| | change_color | 100.0 | 100.0 | 100.0 | 100.0 | 99.9 |
| | pad | 100.0 | 89.6 | 100.0 | 79.5 | 94.0 |
| **S4** | translate | 100.0 | 100.0 | 100.0 | 99.4 | 100.0 |
| | rotate | 99.7 | 96.9 | 99.8 | 98.5 | 99.0 |
| | mirror | 99.8 | 98.8 | 100.0 | 99.1 | 98.9 |
| | extend | 100.0 | 95.9 | 99.4 | 79.8 | 90.5 |
| | empty | 100.0 | 99.9 | 100.0 | 100.0 | 99.9 |
| | crop | 100.0 | 99.3 | 100.0 | 99.7 | 99.9 |
| | fill | 100.0 | 99.6 | 100.0 | 87.4 | 99.9 |
| | duplicate | 86.3 | 99.1 | 100.0 | 91.7 | 97.8 |
| | change_color | 100.0 | 100.0 | 100.0 | 100.0 | 100.0 |
| | pad | 100.0 | 100.0 | 100.0 | 82.2 | 100.0 |

Table 2: In-domain (ID) test grid accuracy for the Sample-Efficiency settings S1–S4, across the models ResNet, Vanilla-TF, Grid-TF, PL-TF and LLaDA for each transformation family. S1 trains with 100 samples, S2 trains with 1,000 samples, S3 trains with 10,000 samples, and S4 trains with 100,000 samples.

# F EXPERIMENT DETAILS

We designed our benchmark experiment to exemplify the use of our generator, and show how state-of-the-art vision models still consistently fail at such elementary tasks as they require compositionality. We provide in this appendix further details into each experiment.

**Compositional Generalization Study (CompGen)**   For all the *CompGen* study, we fix the number of objects to 2, the grid size to 20x20, the object dimension to be smaller or equal to 6x6, and all objects to be fully connected (no unconnected parts; see D.1 "Connectivity"). For each CompGen experiment setting, we generate 100,000 training samples, 1000 in-distribution (ID) validation samples, 1000 out-of-distribution (OOD) validation samples, 1000 ID test samples, and 1000 OOD test samples. The results we report are based on the two aforementioned test sets. Based on these parameters, we then design all experiment settings in such a way that the training and testing sets are built from different "transformation sequences" (i.e. sequence of transformations). In the *CompGen* study, the transformation sequence varies within and between data samples. We thus provide the model with information on which transformation sequence it must apply by appending a task code to the input sequence. The task code is a sequence of tokens that indicates to the model the transformations it must apply (in the correct order). To account for varying "depth" of transformation sequence, we simply pad the input sequence with "identity transformation" tokens–as such, we always append a task code of depth 4–which is the maximum depth of transformation sequence we consider in the entirety of our experiments. We chose the transformations sequences to be from different transformation families, and to be easily composable with one another within the constrained object and grid dimensions. The following is a detailed summary of our experiment settings and experiments.

- **C1 - From Restricted Composite Tasks and Atomic Tasks to Unseen Composite**: We train on a set of composite tasks made of some atomic transformations, and test on composite tasks of the same depth, but not seen during training
  - C1-1: Train on the atomic transformations `translate_up`, `rotate90`, `mirror_horizontal` and all of their mutual compositions (depth $d = 2$), except the specific composition `translate_up` − `rotate90` (and its commutable reverse), which we leave for OOD testing.
  - C1-2: Train on the atomic transformations `change_object_color`, `pad_right`, `fill_holes_different_color` and all of their mutual compositions (depth $d = 2$), except the specific composition `change_object_color` − `pad_right`, which we leave for OOD testing.
  - C1-3: Train on the atomic transformations `crop_bottom_side`, `rotate_90`, `pad_top` and all of their mutual compositions (depth $d = 2$), except the specific composition `rotate_90` − `crop_bottom_side`, which we leave for OOD testing.
  - C1-4: Train on the atomic transformations `double_right`, `crop_contours`, `change_shape_color` and all of their mutual compositions (depth $d = 2$), except the specific composition `double_right` − `crop_contours`, which we leave for OOD testing.
  - C1-5: Train on the atomic transformations `extend_contours_same_color`, `mirror_vertical`, `pad_left` and all of their mutual compositions (depth $d = 2$), except the specific composition `pad_left` − `extend_contours_same_color`, which we leave for OOD testing.

- **C2 - From Restricted Composite Tasks to Unseen Composite**: We train on a set of composite tasks made of some atomic transformations, and test on composite tasks of the same depth, but not seen during training.
  - C2-1: Train on all the mutual compositions (depth $d = 2$) of `translate_up`, `rotate90`, `mirror_horizontal`, except the specific composition `translate_up` − `rotate90` (and its commutable reverse), which we leave for OOD testing.

- C2-2: Train on all the mutual compositions (depth $d = 2$) of `change_object_color`, `pad_right`, `fill_holes_different_color`, except the specific composition `change_object_color` − `pad_right`, which we leave for OOD testing.
  - C2-3: Train on all the mutual compositions (depth $d = 2$) of `crop_bottom_side`, `rotate_90`, `pad_top`, except the specific composition `rotate_90` − `crop_bottom_side`, which we leave for OOD testing.
  - C2-4: Train on the mutual compositions (depth $d = 2$ transformations `double_right`, `crop_contours`, `change_shape_color`, except the specific composition `double_right` − `crop_contours`, which we leave for OOD testing.
  - C2-5: Train on all the mutual compositions (depth $d = 2$) of `extend_contours_same_color`, `mirror_vertical`, `pad_left`, except the specific composition `pad_left` − `extend_contours_same_color`, which we leave for OOD testing.

- **C3 - From Composite Tasks to Deeper Composite Tasks**: We train on a set of composite tasks made of some atomic transformations, and test on composite tasks with the same transformations, but with one additional level of depth.
  - C3-1: Train on all the atomic transformations and mutual compositions (depth $d = 2$) of `translate_up`, `rotate90`, `mirror_horizontal`, and OOD test on all the compositions of depth $d = 3$ of these transformations.
  - C3-2: Train on all the atomic transformations and mutual compositions (depth $d = 2$) of `change_object_color`, `pad_right`, `fill_holes_different_color`, and OOD test on all the compositions of depth $d = 3$ of these transformations..
  - C3-3: Train on all the atomic transformations and mutual compositions (depth $d = 2$) of `crop_bottom_side`, `rotate_90`, `pad_top`, and OOD test on all the compositions of depth $d = 3$ of these transformations.
  - C3-4: Train on all the atomic transformations and mutual compositions (depth $d = 2$) of `double_right`, `crop_contours`, `change_shape_color`, and OOD test on all the compositions of depth $d = 3$ of these transformations.
  - C3-5: Train on all the atomic transformations and mutual compositions (depth $d = 2$) of `extend_contours_same_color`, `mirror_vertical`, `pad_left`, and OOD test on all the compositions of depth $d = 3$ of these transformations.

**Environment Generalization Study (EnvGen)**   For each *EnvGen* experiment setting, we generate 100,000 training samples, 1000 in-distribution (ID) validation samples, 1000 out-of-distribution (OOD) validation samples, 1000 ID test samples, and 1000 OOD test samples. The results we report are based on the two aforementioned test sets. For each *EnvGen* experiment, we fix the transformation sequence (as described below depending on the experiment) and vary some parameters, such as the grid size, the number of objects, the object dimensions or the object properties. The varying parameters constitutes what changes between the in-distribution and OOD testing sets, and forms the basis of the "generalization" experiment. As opposed to the *CompGen* setting, where there are multiple transformation sequences per set, we do not need to provide a task code to the model, but only a single grid on which it must perform the transformation based on the grid settings. To account for varying grid sizes in some of the experiments, we simply pad the the input sequence to the max size which can be observed during the experiment.

- **G1 - Number of Objects Difficulty**: We train on grids with 1 or 2 objects, and OOD test on grids with 3 or 4 objects. We fix the grid size to 15x15.
  * G1-1: Perform experiment with the `translate_up` transformation.
  * G1-2: Perform experiment with the `rotate_90` transformation.
  * G1-3: Perform experiment with the `mirror_horizontal` transformation.

* G1-4: Perform experiment with the `crop_top_side` transformation.
* G1-5: Perform experiment with the `extend_contours_same_color` transformation.

– **G2 - Grid Size Difficulty**: We train on grid sizes between 10x10 and 15x15, and OOD test on grid sizes between 16x16 and 20x20. We fix the number of objects to 2.
  * G2-1: Perform experiment with the `translate_up` transformation.
  * G2-2: Perform experiment with the `rotate_90` transformation.
  * G2-3: Perform experiment with the `mirror_horizontal` transformation.
  * G2-4: Perform experiment with the `crop_top_side` transformation.
  * G2-5: Perform experiment with the `extend_contours_same_color` transformation.

– **G3 - Object Dimension Difficulty**. We train on grids with objects of size between 1x1 and 5x5, and OOD test on grids with objects of size between 6x6 and 10x10.
  * G3-1: Perform experiment with the `translate_up` transformation.
  * G3-2: Perform experiment with the `rotate_90` transformation.
  * G3-3: Perform experiment with the `mirror_horizontal` transformation.
  * G3-4: Perform experiment with the `crop_top_side` transformation.
  * G3-5: Perform experiment with the `extend_contours_same_color` transformation.

– **G4 - Object Complexity Difficulty** We train on grids with symmetric and single-colored objects, and OOD test on grids with asymmetric and multi-colored objects. We fix the number of objects to 2, the grid size to 15x15, and the object dimension to smaller than 6x6.
  * G4-1: Perform experiment with the `translate_up` transformation.
  * G4-2: Perform experiment with the `rotate_90` transformation.
  * G4-3: Perform experiment with the `mirror_horizontal` transformation.
  * G4-4: Perform experiment with the `crop_top_side` transformation.
  * G4-5: Perform experiment with the `extend_contours_same_color` transformation.

– **G5 - All Difficulties Combined** We train on grids with symmetric, single-colored objects of size between 1x1 and 5x5, with 1 or 2 objects, and grid sizes between 10x10 and 15x15. We OOD test on grids with asymmetric, multi-colored objects of size between 6x6 and 10x10, with 3 or 4 objects, and grid sizes between 16x16 and 20x20.
  * G5-1: Perform experiment with the `translate_up` transformation.
  * G5-2: Perform experiment with the `rotate_90` transformation.
  * G5-3: Perform experiment with the `mirror_horizontal` transformation.
  * G5-4: Perform experiment with the `crop_top_side` transformation.
  * G5-5: Perform experiment with the `extend_contours_same_color` transformation.

# G METRICS

We chose to exact grid accuracy to maintain consistency in the field, as this is what has been standard for a lot of ARC-like datasets (to mention a few (28; 40)). However, we acknowledge the strictness of "Exact Grid Match." To address this, we have expanded our evaluation suite to include: Per-Pixel Accuracy (PPA): Accuracy over all grid pixels. Object-PPA: Accuracy calculated only on non-background pixels (to prevent high scores from empty space). ID-Predicted (Novel Metric): This measures "stubbornness." It calculates the percentage of OOD tasks where the model predicted a transformation sequence seen during training (ID) instead of the correct novel sequence.

We report results on all these metrics in our Learning Regime experiments (see appendix H), and find that in many failure cases, ID-Predicted is high, indicating models are over-fitting to the training distribution's compositional structures rather than learning the underlying rules of composition.

We below present the detailed equations of each metric.

## G.1 METRIC 1: PERFECT GRID ACCURACY (GA)

The perfect **grid accuracy** is computed as a percentage of perfectly reconstructed grids. Specifically, given predicted grid $\hat{\mathbf{Y}}_k$ and ground truth grid $\mathbf{Y}_k$

$$\text{Perfect Grid Accuracy Indicator:} \quad \mathbb{I}(\hat{\mathbf{Y}}_k = \mathbf{Y}_k) = \begin{cases} 1 & \text{if all pixels in } \hat{\mathbf{Y}}_k \text{ match } \mathbf{Y}_k \\ 0 & \text{otherwise} \end{cases}$$

$$\text{GA} = \frac{1}{N} \sum_{k=1}^{N} \mathbb{I}(\hat{\mathbf{Y}}_k = \mathbf{Y}_k) \times 100\%$$

## G.2 METRIC 2: PER PIXEL ACCURACY (PPA)

The **Per Pixel Accuracy** informs us on the percentage of wrongly predicted pixels.

As the model's output layer always produces a fixed grid of size $(m \times n)$ (with $m$ and $n$ possibly changing per experiments) each grid is flattened into a vector in $\mathbb{R}^{mn}$. Thus, every predicted grid $\hat{\mathbf{Y}}_k \in \mathbb{R}^{mn}$ and ground-truth grid $\mathbf{Y}_k \in \mathbb{R}^{mn}$ have exactly $P = mn$ pixels, regardless of the original (unpadded) grid size.

We can thus calculate the per-pixel accuracy for sample $k$ as follows:

$$\text{Per-Pixel Accuracy}_k = \frac{1}{mn} \sum_{i=1}^{mn} \mathbb{I}(\hat{\mathbf{Y}}_{k,i} = \mathbf{Y}_{k,i}).$$

The overall per-pixel accuracy across $N$ samples is:

$$\text{PPA} = \frac{1}{N} \sum_{k=1}^{N} \left( \frac{1}{mn} \sum_{i=1}^{mn} \mathbb{I}(\hat{\mathbf{Y}}_{k,i} = \mathbf{Y}_{k,i}) \right) \times 100\%.$$

## G.3 METRIC 3: OBJECT ACCURACY

The **Per Object Pixel Accuracy** metric that informs us on the extent to which objects were mispredicted.

For each sample $k$, let the predicted grid be $\hat{\mathbf{Y}}_k \in \mathbb{R}^{mn}$ and the ground-truth grid $\mathbf{Y}_k \in \mathbb{R}^{mn}$, where each pixel can be either zero (background) or a non-zero object label.

The **object accuracy** counts mismatches **only on pixels $i$ where at least one of the grids is non-zero**.

$$\text{ObjectError}_k = \sum_{i=1}^{mn} \mathbb{I}\left( \begin{array}{c} (\hat{\mathbf{Y}}_{k,i} \neq 0 \ \vee \ \mathbf{Y}_{k,i} \neq 0) \\ \wedge \ \hat{\mathbf{Y}}_{k,i} = \mathbf{Y}_{k,i} \end{array} \right)$$

The **per-object accuracy** for sample $k$ is then:

$$\text{Object Accuracy}_k = \frac{\text{ObjectError}_k}{mn}.$$

Averaged across $N$ samples:

$$\text{Obj-PPA} = \frac{1}{N} \sum_{k=1}^{N} \text{Object Accuracy}_k \times 100\%.$$

### G.4 METRIC 4: MODEL "STUBBORNESS" (STBN)

As it is difficult from the previous metrics to evaluate the precise failure modes of models, we introduce a new metric, which we term "Model *Stubborness*". Model stubborness is specifically introduced for the CompGen out-of-distribution (OOD) test settings, for which models need to predict a grid following a different transformation sequence than seen during in-distribution (ID) training.

We hypothesize that models may behave in a *stubborn* manner at test time - i.e., despite facing a novel transformation sequence, they may still produce an output grid consistent with one of the ID (in-distribution) training transformation sequences.

Formally, let:

- $T_{\text{ID}} = \{\tau_1^{ID}, \tau_2^{ID}, \ldots, \tau_M^{ID}\}$ denote the set of all ID transformation sequences, which is **disjoint** from the OOD transformation set $T_{\text{OOD}} = \{\tau_1^{OOD}, \tau_2^{OOD}, \ldots, \tau_M^{OOD}\}$, i.e., $T_{\text{ID}} \cap T_{\text{OOD}} = \emptyset$.
- $\mathbf{X}_k^{\text{OOD}}$ be an input grid which should be transformed following a given transformation sequence $\tau_\lambda^{OOD}$
- $\hat{\mathbf{Y}}_k$ be the model prediction for sample $k$,
- and let $\tau(\cdot)$ denote the deterministic application of a transformation sequence.

We say the model is **stubborn** on sample $\mathbf{X}_k^{\text{OOD}}$ if:

$$\exists \tau \in T_{\text{ID}} \quad \text{s.t.} \quad \hat{\mathbf{Y}}_k = \tau(\mathbf{X}_k^{\text{OOD}}).$$

The stubbornness indicator is defined as:

$$\mathbb{I}_{\text{stubborn}}(k) = \begin{cases} 1 & \text{if } \exists \tau \in T_{\text{ID}} : \hat{\mathbf{Y}}_k = \tau(\mathbf{X}_k^{\text{OOD}}), \\ 0 & \text{otherwise.} \end{cases}$$

The **Model Stubbornness Metric** over $N$ OOD samples is:

$$\mathcal{STBN} = \frac{1}{N} \sum_{k=1}^{N} \mathbb{I}_{\text{stubborn}}(k) \times 100\%.$$

Intuitively, the metric measures how often the model ignores the novel OOD transformation sequence and instead outputs a grid that could have been produced by an ID transformation rule.

# H   COGITAO LEARNING REGIME

To assess whether the selected training regime of 10 Epochs for 100k samples was reasonable, we conducted two follow up experiments to further justify our choice, as well as the validity of COGITAO's experimental framework and dataset. Specifically, we conducted:

1. Scaling Experiments with the Grid-TF model through increasing dataset size to 1 Million samples and training for up to 100 epochs on CompGen-C1 and CompGen-C3.

2. Introduction of "Experiment 0", which matches the logic of CompGen-C1 and CompGen-C3, but only uses compositions of *translation* tasks (as opposed to more compelx compositions - see F) which are trivial to learn individually (as shown in E). This experiment 0 is also run with our Grid-TF on different splits up to 1 millions samples with training for up to 100 epochs.

We report the complete metrics outlined in G for these experiments, which allows us to shed light in specific failure modes and model behaviours on the COGITAO dataset, which we discuss below.

| Setting | Exp | #Samples | #Epochs | GA | | PPA | | Obj-PPA | | "Stubborness" |
|---------|-----|----------|---------|-----|------|-----|------|-----|------|---------------|
| | | | | ID | OOD | ID | OOD | ID | OOD | |
| C1 | 0 | 100k | 100 | 0.780 | 0.360 | 0.983 | 0.996 | 0.859 | 0.962 | 0.000 |
| | 0 | 250k | 50 | 0.713 | 0.507 | 0.982 | 0.997 | 0.860 | 0.970 | 0.000 |
| | 0 | 500k | 20 | 0.503 | 0.341 | 0.964 | 0.996 | 0.712 | 0.996 | 0.000 |
| | 0 | 1M | 10 | 0.764 | 0.298 | 0.983 | 0.994 | 0.857 | 0.923 | 0.000 |
| | 1 | 100k | 100 | 0.537 | 0.000 | 0.984 | 0.916 | 0.836 | 0.293 | 0.118 |
| | 1 | 250k | 50 | 0.779 | 0.001 | 0.994 | 0.943 | 0.927 | 0.462 | 0.166 |
| | 1 | 500k | 20 | 0.454 | 0.000 | 0.977 | 0.926 | 0.800 | 0.342 | 0.000 |
| | 1 | 1M | 10 | 0.773 | 0.000 | 0.994 | 0.916 | 0.922 | 0.298 | 0.733 |
| C3 | 0 | 100k | 100 | 0.791 | 0.178 | 0.985 | 0.940 | 0.876 | 0.503 | 0.888 |
| | 0 | 250k | 50 | 0.752 | 0.083 | 0.985 | 0.933 | 0.879 | 0.464 | 0.741 |
| | 0 | 500k | 20 | 0.631 | 0.266 | 0.982 | 0.950 | 0.867 | 0.606 | 0.760 |
| | 0 | 1M | 10 | 0.600 | 0.302 | 0.975 | 0.955 | 0.770 | 0.617 | 0.751 |
| | 1 | 100k | 100 | 0.449 | 0.059 | 0.981 | 0.965 | 0.711 | 0.494 | 0.307 |
| | 1 | 250k | 50 | 0.998 | 0.012 | 1.000 | 0.965 | 1.000 | 0.486 | 0.524 |
| | 1 | 500k | 20 | 0.672 | 0.009 | 0.991 | 0.963 | 0.860 | 0.468 | 0.610 |
| | 1 | 1M | 10 | 0.462 | 0.036 | 0.983 | 0.965 | 0.758 | 0.490 | 0.514 |

Table 3: CompGen learning regime results and introduction of Experiment 0.

We find that models still fail to systematically compose simple translations OOD compared to ID. Increasing sample size by 10× does not close the generalization gap. The low OOD performance is not a result of insufficient data or under-training, but rather a structural inability of the architecture to generalize compositionally. This is further evident when looking at the incapacity of models to compose despite the simplicity of the translation transformations, as shown in the Experiment 0 results. Importantly, in the newly added Experiment 0 (translation-only, see response to Q1)–which is designed in its configuration to be easier than all the the other CompGen experiments–models do show signs of OOD generalization through OOD performance of 15-50% compared to the usual near 0%, even though a significant ID to OOD drop indicating

a difficulty in extrapolation remains. This confirms that our experimental framework is sound and that the failures in the other, slightly more difficult experiments–for which training regime scaling did not improve performance–does reflect genuine limitations in compositional reasoning. This also further highlights the pertinence of COGITAO allowing for fine-grained control of tasks difficulty and the interest in experiments with incremental difficulties for which true compositional reasoning should allow great performance at different levels and not only at the more trivial ones.

Looking at the model "stubborness" meta-metric, we can see that indeed, in many cases, the model fails to correctly predict OOD as it still tries to predict in-distribution sequences - highlighting that there is no compositional generalization capacity in the model. Conceptually, we can thus compartementalize failures into two interpretable categories:

- **ID Bias:** The model applies the transformation it observes during training (e.g., translate_right) even when the OOD task specifies another transformation (translate_up).
- **Structural composition failure:** When moving from depth-1 compositions to depth-2 (CompGen Setting 3), models are able to apply the transformation representing a sequence of atomic transformations–highlighted by a high ID performance–but they fail to decompose such sequences to extract their atomic transformations and recompose them as expected for the OOD case; this again highlights a concrete failure in compositional generalization ability.

Across settings, models retain high pixel-level precision while failing to reorganize familiar components into novel compositions–PPA remains high, Object-PPA moderately high (relatively), while GA collapses OOD (often near-zero). This highlights that models achieve some local correctness yet fail to recover the global transformation–a core capability COGITAO is designed to probe. The "stubbornness" meta metric further shows that OOD predictions frequently mirror ID habits rather than the specified atomic transformations or their order. This distinction is critical: GA strictly identifies whether meaningful compositional reasoning occurred, while PPA and Object-PPA clarify how it failed.

# I MODELS

The selection of encoder networks includes both standard baselines (ResNet, Vanilla TF) and more specialized architectures (Grid TF, Pondering Looped TF, LLaDA) designed to better capture abstract and spatial reasoning, compositionality, and generalization capabilities.

Table 4 summarizes some of the notable architectural and modeling characteristics used for each encoder network part of the models. The models were designed to roughly have the same size of 1.2 Million of parameters in order to propose a fairer performance comparison, in spite of being aware that different architectures may have different modeling requirements to optimize their performance.

| | Architectural | | | | | Modeling | |
| Encoder | APE | RPE | PEMixer | Recurrence | Diffusion | VT | Registers |
|---|---|---|---|---|---|---|---|
| ResNet | — | — | — | — | — | — | — |
| Vanilla-TF | learned | — | sum | — | — | — | — |
| Grid-TF | 2D-sincos w/OPE | RoPE | vec weighted sum | — | — | ✓ | ✓ |
| PL-TF | 2D-sincos | RoPE | sum | ✓ | — | ✓ | — |
| LLaDA | — | RoPE | — | — | ✓ | ✓ | — |

Table 4: Overview of the encoder networks for the different architectural and modeling techniques considered. APE: Absolute Positional Encoding. OPE: Object Positional Encoding. RPE: Relative Positional Encoding. VT: Visual Tokens. Registers: Register tokens. PEMixer: Positional Encoding Mixer, where "vec weighted sum" signifies a vector-weighted sum.

## I.1 RESNET

We employ an architecture based on ResNet(29) as a standard baseline notable for its strong historical performances on vision tasks, principally distinguishing itself from the other models evaluated here by its convolutional inductive bias and lack of attention mechanism.

In our implementation, the input grid is processed as a low-resolution image after an artificial channel dimension is created through one-hot encoding of the token categories that can possibly be predicted (i.e., `num_token_categories`). Then, the core of the encoder consists of a sequence of residual blocks. Crucially, all convolutional operations within these blocks are performed without any spatial downsampling (e.g., using stride 1 convolutions and no pooling layers). Despite missing on a slightly more global receptive field, this design choice is important for our tasks, as it strictly preserves the spatial dimensions of the feature maps throughout the network and thus does not require an approach of upsampling. The stacking of convolutional layers without downsampling should also allow the effective receptive field for each individual output pixel to grow, enabling the model to better capture useful context regions while retaining more precise spatial information.

Our specific ResNet architecture comprises:

- An initial convolutional layer to project the input grid channels to the model's hidden dimension: `num_token_categories` input channels, 32 output channels, kernel size of 1, stride of 1, padding of 0. This is followed by Batch Normalization (59) and a ReLU activation function (60).

- A series of 5 residual blocks:
    1. 32 input channels, 64 output channels, kernel size of 3, stride of 1, padding of 1.
    2. 64 input channels, 128 output channels, kernel size of 3, stride of 1, padding of 1.

3. 128 input channels, 128 output channels, kernel size of 1, stride of 1, padding of 0.
4. 128 input channels, 256 output channels, kernel size of 3, stride of 1, padding of 1.
5. 256 input channels, 128 output channels, kernel size of 1, stride of 1, padding of 0.

- A final convolutional layer (1x1 convolution) to map the features from the last residual block to the required embed dimension per pixel/token: 128 input channels, `embed_dim` output channels, kernel size of 1, stride of 1, padding of 0.

### I.2 VANILLA TF

We adapt the standard Vision Transformer architecture, as defined in (48), to our grid-based tasks. The (padded and one-hot encoded) input grid is divided into a sequence of non-overlapping patches, which are effectively single tokens since we use a patch size and stride of 1 when linearly projecting the grid image into the embedding dimension using a 2D convolution.

A basic approach to incorporate spatial information is to use (randomly initialized) 1D learnable positional embeddings to add to the embedded input sequence before it is passed to the encoder network.

This vanilla Transformer encoder consists of multiple layers, each containing a multi-head self-attention (MHSA) mechanism followed by a position-wise feed-forward network. We use Pre-Layer Normalization (Pre-LN) (61; 62) and thus apply Layer Normalization before both the MHSA and feed-forward sub-layers. Residual connections are also used around each sub-layer, as is standard. To be able to produce the output grid using an MLP head, a linear layer is applied to each output token of the sequence (from which the special extra tokens have been truncated) at the end of the encoder network in order to map the embedding dimension to that of the MLP head which will then predict the logits for all of the tokens which, once softmax applied and spatially reshaped, form the output grid.

Notable hyperparameters are:

- Input grid partitioning: Patch size of 1, stride of 1. The grid is partitioned at the pixel-level.
- Embedding dimension: 128.
- Number of encoder layers: 6.
- Number of attention heads in MHSA: 4.
- Dimensionality of the feed-forward layer: Factor of 4 times the embedding dimension, thus 512.
- Activation Functions: GELU (63).
- Output projection: A linear layer maps each embed_dim-dimensional output token (excluding the extra tokens such as the register tokens or the task embedding) to num_classes logits, where num_classes depends on the number of tokens to predict (e.g., 15 if Visual Tokens are used, 11 otherwise, as there are ten for the 0-9 symbols and one for padding).

### I.3 GRID TF

The Grid-TF is a variant of the Vision Transformer architecture adapted to improve performance on abstract visual reasoning tasks, especially in grid-like environments, similar to the COGITAO data. It incorporates several modifications to improve spatial and abstract visual reasoning. Similar to the Vanilla TF, the input grid is embedded as patches at the pixel level (i.e., patch size and stride of 1 when performing the 2D convolution to transform the grid into an embedded sequence), thus–because deemed inadequate in this context–not exactly appropriating one of the main techniques of Vision Transformers: larger patches.

The key architectural modifications from the Vanilla-TF are:

- Positional Encodings:
  - 2D Absolute Positional Encoding (APE): We use 2D sinusoidal absolute positional encodings, extended from the 1D sinusoidal absolute positional encodings used in (49), which are fixed (i.e., not learned) and directly reflect the 2D nature of the input grid. They are added to the patch embeddings following the PEMixer strategy.

  - Object Positional Encoding (OPE): OPE is used as part of the APE, as described in (28) where half of the APE dimension is allocated to the object positions while the other half is used for the x and y positions in the grid. The OPE is also typically coupled with an appropriate PEMixer strategy, such as a vector-weighted sum. We make the choice to compute the object positions *after* having padded (whether with Visual Tokens or with simple padding) the input grid, as opposed to before.

  - Positional Encoding Mixer (PEMixer): The PEMixer presented in (28) defines the strategy by which we encode the absolute positional information into the input embeddings from the absolute positional embeddings. It allows different strategies such as a sum (the standard one), a weighted sum, a vector-weighted sum, etc.

  - Relative Positional Encoding (RPE): Due to the importance of the relative positions of the grid tokens when transforming objects, an RPE scheme comes in as a natural consideration. Among possible techniques of RPE, we choose to use Rotary Position Embedding (RoPE) (52), incorporated into the self-attention mechanism in order to inject relative spatial information between the grid tokens. After initial experiments with ALiBi (64), another RPE method, extended to 2D as in (28), we found that RoPE yields comparable performance on our tasks and thus decided to use RoPE for its simplicity.

- Registers: We prepend (6) register tokens to the input sequence, following results from (50) and a drawn parallel to slots for object-centric learning in (65). Those registers are additional, randomly initialized and learnable tokens appended to the sequence of patch embeddings in order to possibly improve model performance by functioning as containers for less informative "background" regions of the grid. Thus, they do not correspond to any specific input informing the model and should be leveraged by the attention mechanism to improve global context aggregation and internal representations. The positional encodings are not used for those extra tokens.

Another modification to the Vanilla-TF is the use of dropout. We apply dropout with a rate of 0.1 after the multi-head self-attention layer and after the feed-forward layer in each Transformer encoder block.

The remaining notable hyperparameters are the same as for the Vanilla-TF.

To better visualize the overall architecture of the Grid-TF encoder, we provide an overview diagram in figure 6.

### I.4 PONDERING LOOPED TF (PL-TF)

The Pondering Looped Transformer model closely follows the PonderNet paper by Banino et al. (26), using a looped Transformer with weight-sharing as the encoder. As this architecture is less popular than the other ones, we also provide a simplified overview of the architecture in figure 7.

The PL-TF fundamentally differs from the other transformer models in two aspects:

- It possesses an inductive bias of recurrent architecture leveraging weight-sharing through an iterative looping process over the same block structure (27; 66; 67; 68). At each step, the model updates the latent

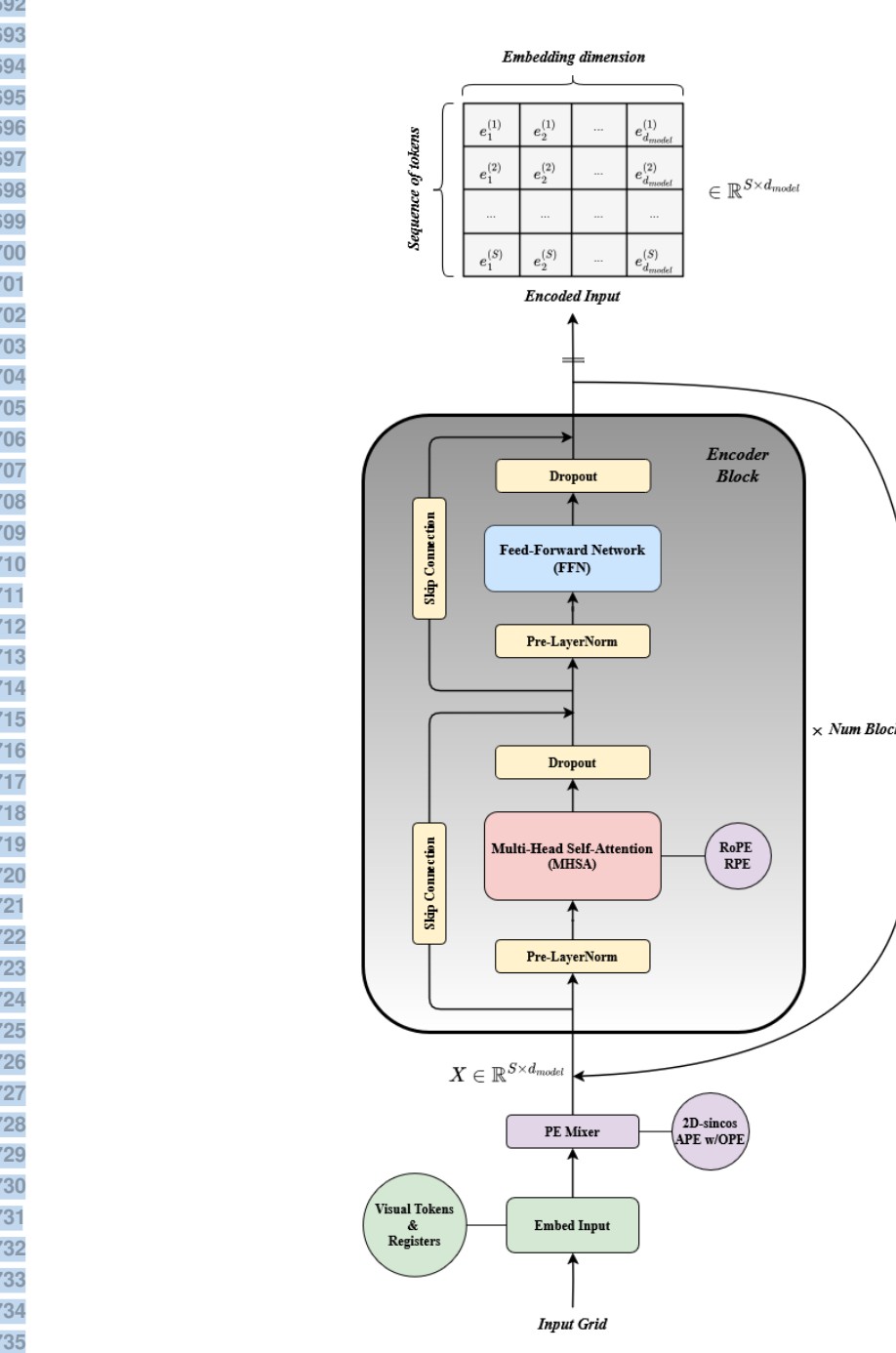

Figure 6: Overview of the Grid Transformer encoder.

representation, which can be useful to mimic a multi-stage reasoning process for tasks that require multiple sequential operations on objects.

- It makes use of a carefully designed adaptive compute time framework–named PonderNet–controlling the number of iterations and computation effort spent on each data sample (26). Essentially, the model learns a halting probability at each iterative step from an aggregated representation of the current hidden state, which allows it to dynamically and probabilistically decide the number of iterations needed for a given input.

Since parameter-efficiency is not a primary goal, in order to obtain a learnable parameter count comparable to that of the other models, the embedding dimension is set to $d_{model} = 256$–instead of $d_{model} = 128$ as in the other transformers–and the number of attention heads is set to eight. We use a single encoder block (although more can be used) with weight-sharing.

The computational resource constraints led to the setting of a relatively low maximum number of pondering steps, although a larger maximum number of steps is likely required to fully harness the capabilities of the recurrent and pondering modules. We note that with those decisions the total number of passes through an encoder block is not too far off from that of the other transformers. Furthermore, the pondering-specifc hyperparameters used are: 2-layer halting node with hidden dimension of 128, $\epsilon$ of 0.01, $\lambda_p$ of 0.05, $\beta$ of 0.01.

The encoder block is the same as the Grid-TF with pre-LN, additive skip connections, dropout of 0.1 and a RoPE RPE scheme in the MHSA module. The PL-TF leverages Visual Tokens as the Grid-TF but it uses a 2D sincos APE scheme without object positional encoding or PEMixer.

### I.5 LLADA

We include LLaDA (25) in our set of models due to its demonstrated strength on logical and arithmetic tasks. LLaDA operates non-autoregressively and exploits bidirectional dependencies over masked target sequences, making it well-suited for structured input-output mappings. We adapt the original LLaDA setup skipping pretraining and training it from scratch via supervised learning. Each instance of input data is represented as a flattened concatenation of a task embedding, input sequence, and a partially masked target sequence. During training, a random portion of target tokens is masked (mask ratio sampled uniformly in $[0, 1]$), and the model is trained to reconstruct the masked tokens. At inference, the target is fully masked, and reconstruction proceeds over 32 denoising steps, each resolving a fraction of the masked tokens based on confidence.

In contrast to the original architecture, We use a lightweight 6-layer version with approximately 1.2M parameters. The architecture uses an embedding dimension of 128, a feedforward hidden size of 384 (MLP ratio of 4), and 4 attention heads. The model incorporates RMSNorm (69) with affine parameters and uses SiLU activation functions (70). Rotary positional embeddings (RoPE) (52) are applied independently within each transformer layer, using a shared base frequency parameter of $\theta = 500,000$. Weights are initialized via the Mitchell method with a standard deviation of 0.02. No dropout is used throughout training. The model's vocabulary comprises 12 tokens, reflecting the tokenized representation used in COGITAO, and includes special tokens for padding and masking.

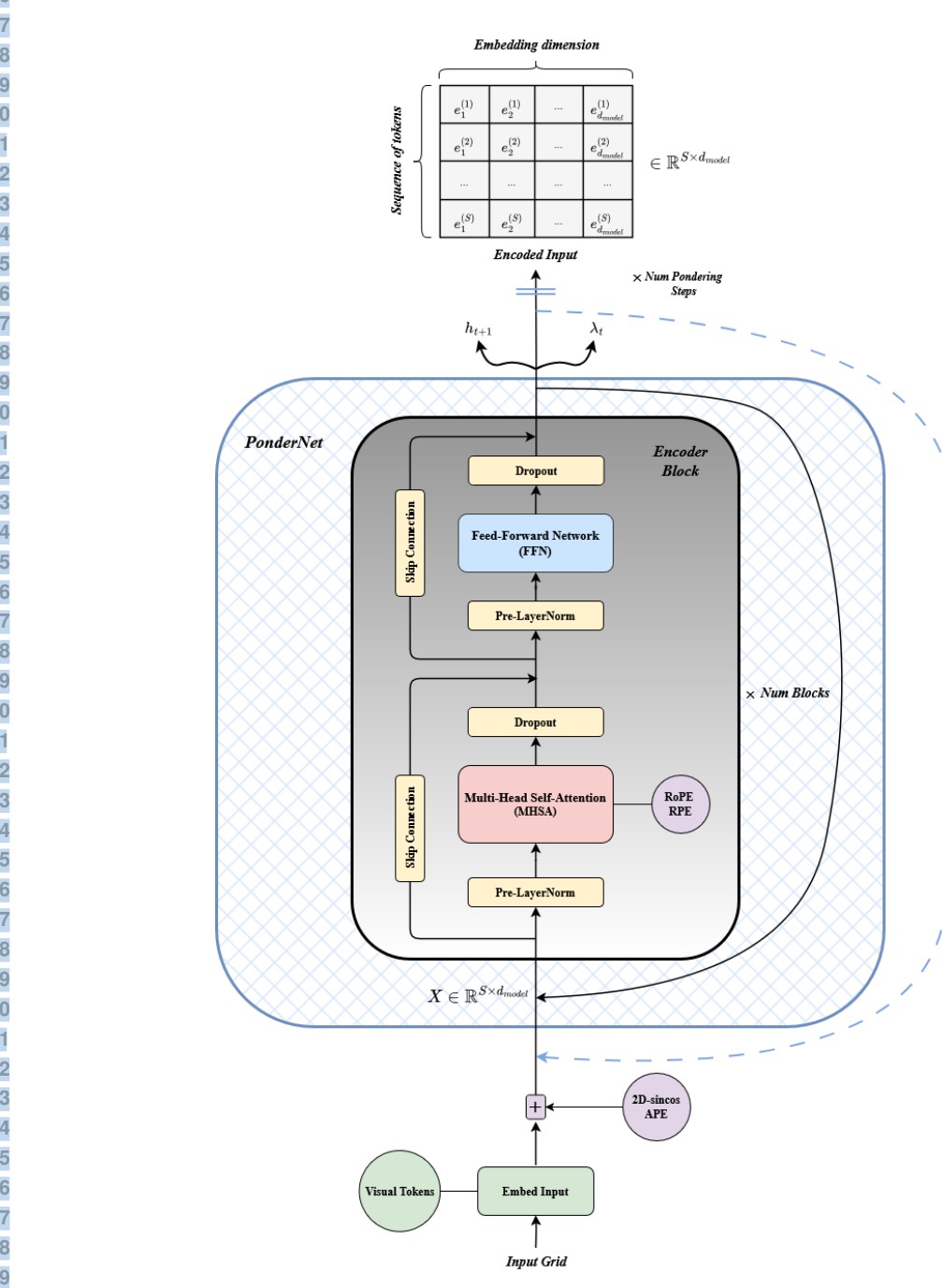

Figure 7: Overview of the Pondering Looped Transformer encoder.

## J  TRAINING PROCEDURE

To ensure a fairer comparison across different models, we adopted a consistent training and evaluation framework whenever possible. Key aspects are detailed as follows:

- Training Procedure and Data: All models were trained from scratch using supervised learning. We used a dataset of 100,000 unique samples. Training proceeded for 10 epochs with a batch size of 64, meaning each model saw a total of 1,000,000 samples, This implies a sample-efficient mode of experimentation, as observed through the training learning curves hinting at a non-terminal convergence for several experiments and considering the typical quantity of samples that Transformer-based models require.

- Validation and Testing: Model performance was monitored during training on a validation set of 1,000 samples. After training, models were evaluated on a distinct held-out test set of 1,000 samples. We always compute the performance on ID *and* OOD val/test sets.

- Modeling Strategy:
  - An input grid with shape `[H, W]` is converted to an artificial image of shape `[C, H, W]` by the creation of a channel dimension through one-hot encoding of the categories of tokens that can be predicted.

  - For the experiments within the *CompGen* study, the transformations sequence vary within the trainin set (and w.r.t. the OOD sets). We therefore provide the model with some context in the form of a task embedding in order to inform it of what task it should perform given the input grid. For that purpose, we considered two approaches: task tokens and in-context example. We decided to use the first approach, where we provide the model with a sequence of tokens representing the atomic transformations composing the task to perform. The second approach was to provide the model with an example (i.e., an input-output pair of grids) of the task to perform. However, it was not experimented with due to time and resource constraints.

- Early Stopping and Model Selection: No early stopping was performed as the number of epochs was restricted and in most experiments the models showed steady convergence throughout the 10 epochs. The checkpoint used for final testing was chosen based on the best (lowest) validation loss achieved on the OOD validation set.

- Loss Function: We employed a pixel-wise cross-entropy loss, calculated between the entire predicted token grid and the ground-truth target grid, both padded to a maximum size equal to the largest grid size within the dataset.

- Evaluation Metric: The primary metric for reporting performance is grid accuracy. This is the percentage of predicted grids that perfectly match the ground-truth target grids. Both grids were padded to the maximum grid size observed in the dataset before comparison. The padding is either composing of usual padding tokens or of special padding tokens named "Visual Tokens" (VTs)(28). This means that when VTs are used as part of the modeling strategy for Transformer-based models, the model has to predict correctly the whole padded grid, with its special padding tokens, in order to perform well since the metric computation does not discard the tokens/pixels outside of the boundaries denoted by the true grid size.

- Optimizer and Learning Rate: The AdamW optimizer (53) was used, with a learning rate following a Cosine Annealing schedule that monitors the OOD validation loss at each epoch.

- Weight Initialization: Model weights were initialized from a truncated normal distribution.

- Training Precision: Training was conducted using FP16 mixed precision.

- Training Duration: A training run lasted between 7 minutes and several hours depending on the model, hardware and specific experiment.

## J.1 RESNET

The learning rate and weight decay used for ResNet are of 1e-3, same as for the Vanilla-TF and Grid-TF models.

ResNet is trained without Visual Tokens, as their purpose is mainly to mitigate the loss of the 2D spatial structure when going from a 2D grid to a flattened sequence, as it is the case for Transformer models, as well as defining more explicit grid boundaries.

For the Compositionality experiments, the task embedding is appended (along the spatial dimension) to the spatially flattened output of the ResNet encoder network. This means that the task embedding only enters the input transformation process at a late stage, similar to what is done in (15), compared to how the task embedding is used with the Transformer-based models working with sequences. Practically, it is the MLP head that has to leverage the information provided by the task embedding in order to make better-informed predictions.

## J.2 VANILLA-TF & GRID-TF

The two transformer models use exactly the same hyperparameters. The learning rate and weight decay are 1e-3. Before the initial learning rate is set to change with the Cosine Annealing schedule, a linear warm-up phase takes place for 200 steps.

A key difference between the Vanilla-TF and the Grid-TF is that the latter uses Visual Tokens while the former does not.

The input sequence to the transformer results from the one-hot encoding of the input grid for which embedding patches are created using a 2D convolution with patch size of 1 and stride of 1.

For the Compositionality experiments, the task embedding is appended (along the spatial dimension) to the input sequence *before* being encoded by the Transformer encoder. This provides the full task context at the start of the processing of the input by the model, thus increasing the information about the task to perform propagated through the model.

## J.3 PONDERING LOOPED TF

The PL-TF is trained similarly to the two previous transformer models and follows the PonderNet paper (e.g., with regard to the update of the reconstruction and regularization losses). It mainly uses the same hyperparameters as the aforementioned models, but with a smaller batch size of 32 due to the possible increase in computations and resource constraints. It uses data modeled with Visual Tokens as for the Grid-TF.

## J.4 LLADA

We train the LLaDA model using the same hyperparameter configuration as that employed for the vision transformer baselines, including the Vanilla ViT and Grid ViT architectures. However, we introduce two key modifications: Firstly, while retaining the AdamW optimizer (53), we reduce the learning rate to $2 \times 10^{-4}$ and set the weight decay to $0.01$. Empirically, this lower learning rate leads to significantly faster convergence for LLaDA during training. Second, we extend the training schedule to 20 epochs, doubling the number

of training epochs compared to the baseline setup. This adjustment compensates for the fact that LLaDA masks only approximately 50% of the target tokens on average, thereby ensuring that all models are trained to predict a comparable total number of tokens and enabling a fair evaluation.

# K  RESULTS

We present a set of results for each model and each of the 40 individual experiments considered in the studies of CompGen and EnvGen. The results reported are the average of three runs with different random seeds. A few experiments showed a high sensitivity to the random seed, up to a 15% difference, while the others yielded similar results for different runs. Tables 5 6 7 8 display the in-domain (ID) and out-of-domain (OOD) test grid accuracies (i.e., the predicted grid and the ground-truth target grid, both padded to a max. size, should perfectly match) for all models considered across all the experiment settings (i.e., C1-C3 and G1-G5). While the main paper provides a summary in the form of averaged results over each setting for clarity and space constraints, the exhaustive results here enable a more granular understanding of model behavior given different transformations and compositions thereof.

We include here results of a ResNet-like model baseline, which is omitted from the main text table and discussion. As a standard convolutional neural network performing well on vision tasks - even when it includes a level of reasoning (15), ResNet is poorly designed for compositionality tasks, and is more sample efficient than Transformer-based architectures (15). Consequently, it was not deemed to be of focus in this paper, alth-ough it could provide few insights. Its inclusion here also serves to put into perspective the performance of models with clearly different inductive biases. In the main paper, we focus on models better designed for abstraction and generalization, with a Vanilla-TF as starting point, followed by Grid-TF, PL-TF and LLaDA models.

The results clearly highlight that compositional generalization remains a major challenge, which COGITAO allows to explore. Across most CompGen experiments, models perform poorly in the OOD setting. Notably, some transformations stand out as yielding easier or more difficult tasks. For example, translate_up is by far the easier transformation followed for example by extend_contours_same and crop_top_side. More difficult transformations include mirror_horizontal and rotate90 which empirically, but also intuitively, are seen to be more complex. A transformation such as crop_top_side may be easier as it mainly introduces localized changes, allowing models, especially those with strong inductive biases for locality, to generalize more effectively. In contrast, transformations such as mirror_horizontal and rotate90 result in drastic spatial reconfigurations of visual features, which can severely disrupt learned representations, in particular when not explicitly trained on such variations.

In the *EnvGen* experiments, ResNet performed competitively, and even better for settings focused on changes of grid size and number of objects. These experiments involve systematic variations of properties of the grid environment and scene (e.g., the grid size, the number of objects, their spatial distribution) which can still be captured through localized pixel/symbol statistics and spatial patterns that do not require compositional abilities. ResNet's strong performance here could be attributed to its architectural inductive biases: overlapping receptive fields, translation invariance from convolutions, and the ability to encode detailed local structures without aggressive downsampling (as per our implementation). We suspect that these properties enable it to effectively maintain useful representations when the domain shift involves more spatially coherent variations, as opposed to more abstract ones based on a compositional prior.

These detailed tables serve not only as a complement to the averaged results in the main paper, but also as a valuable diagnostic tool for evaluating the specific generalization challenges posed by different transformations. They reinforce the necessity of designing models and training regimes that can robustly handle a wide range of compositional and systematic shifts in visual input with respect to diverse transformations implying fundamentally different types of changes in the grid. Consequently, an exhaustive table with more experiments for all the transformations enabled by COGITAO would be a natural extension of those results.

| Setting | Experiment | ResNet | | Vanilla-TF | |
|---------|-----------|--------|-----|------------|-----|
| | | ID | OOD | ID | OOD |
| | experiment-1 | $0.1 \pm 0.1$ | $0.0 \pm 0.0$ | $23.3 \pm 5.6$ | $0.0 \pm 0.0$ |
| | experiment-2 | $8.9 \pm 3.1$ | $0.0 \pm 0.0$ | $22.7 \pm 3.1$ | $0.0 \pm 0.0$ |
| C1 | experiment-3 | $1.0 \pm 0.1$ | $0.0 \pm 0.0$ | $5.4 \pm 0.7$ | $0.0 \pm 0.0$ |
| | experiment-4 | $2.3 \pm 0.9$ | $0.0 \pm 0.0$ | $16.7 \pm 5.3$ | $0.0 \pm 0.0$ |
| | experiment-5 | $0.6 \pm 0.3$ | $0.0 \pm 0.0$ | $14.4 \pm 0.4$ | $0.0 \pm 0.0$ |
| | experiment-1 | $0.7 \pm 0.2$ | $0.0 \pm 0.0$ | $26.8 \pm 0.5$ | $0.0 \pm 0.0$ |
| | experiment-2 | $12.2 \pm 6.5$ | $0.0 \pm 0.0$ | $14.6 \pm 0.5$ | $0.0 \pm 0.0$ |
| C2 | experiment-3 | $0.6 \pm 0.2$ | $0.0 \pm 0.0$ | $8.8 \pm 1.2$ | $0.0 \pm 0.0$ |
| | experiment-4 | $7.7 \pm 2.0$ | $0.0 \pm 0.0$ | $17.3 \pm 0.3$ | $0.0 \pm 0.0$ |
| | experiment-5 | $0.2 \pm 0.1$ | $0.0 \pm 0.0$ | $21.6 \pm 1.5$ | $0.0 \pm 0.0$ |
| | experiment-1 | $0.0 \pm 0.0$ | $0.0 \pm 0.0$ | $13.8 \pm 1.2$ | $0.8 \pm 0.1$ |
| | experiment-2 | $10.4 \pm 3.2$ | $3.2 \pm 1.1$ | $37.4 \pm 5.8$ | $5.5 \pm 2.1$ |
| C3 | experiment-3 | $3.0 \pm 0.6$ | $0.1 \pm 0.0$ | $12.6 \pm 2.0$ | $1.1 \pm 0.3$ |
| | experiment-4 | $0.7 \pm 0.3$ | $0.0 \pm 0.0$ | $68.3 \pm 8.1$ | $12.3 \pm 3.3$ |
| | experiment-5 | $0.1 \pm 0.1$ | $0.0 \pm 0.0$ | $15.1 \pm 6.1$ | $0.3 \pm 0.1$ |

Table 5: ID and OOD test grid accuracy with SEM (Standard Error of the Mean) for the *CompGen* experiments (C1–C3) across the models ResNet and Vanilla-TF.

| Setting | Experiment | Grid-TF | | PL-TF | | LLaDA | |
|---|---|---|---|---|---|---|---|
| | | ID | OOD | ID | OOD | ID | OOD |
| C1 | experiment-1 | $43.5 \pm 16.9$ | $0.0 \pm 0.0$ | $65.8 \pm 1.0$ | $0.3 \pm 0.2$ | $27.7 \pm 1.4$ | $0.0 \pm 0.0$ |
| | experiment-2 | $47.2 \pm 18.6$ | $0.0 \pm 0.0$ | $82.8 \pm 0.6$ | $0.0 \pm 0.0$ | $67.0 \pm 0.4$ | $0.0 \pm 0.0$ |
| | experiment-3 | $56.7 \pm 7.7$ | $0.0 \pm 0.0$ | $83.7 \pm 1.3$ | $0.0 \pm 0.0$ | $33.3 \pm 2.5$ | $0.0 \pm 0.0$ |
| | experiment-4 | $91.7 \pm 0.3$ | $0.0 \pm 0.0$ | $89.0 \pm 1.5$ | $0.0 \pm 0.0$ | $64.0 \pm 29.1$ | $0.0 \pm 0.0$ |
| | experiment-5 | $59.7 \pm 11.9$ | $0.0 \pm 0.0$ | $83.7 \pm 0.5$ | $0.0 \pm 0.0$ | $32.6 \pm 7.5$ | $0.0 \pm 0.0$ |
| C2 | experiment-1 | $56.2 \pm 5.8$ | $0.0 \pm 0.0$ | $59.3 \pm 1.0$ | $0.5 \pm 0.5$ | $31.2 \pm 0.8$ | $0.0 \pm 0.0$ |
| | experiment-2 | $94.3 \pm 1.5$ | $0.0 \pm 0.0$ | $76.1 \pm 12.8$ | $0.0 \pm 0.0$ | $72.0 \pm 1.2$ | $0.0 \pm 0.0$ |
| | experiment-3 | $56.6 \pm 5.8$ | $0.0 \pm 0.0$ | $87.4 \pm 0.7$ | $0.0 \pm 0.0$ | $27.2 \pm 1.2$ | $0.0 \pm 0.0$ |
| | experiment-4 | $92.3 \pm 2.2$ | $0.0 \pm 0.0$ | $86.3 \pm 3.5$ | $0.0 \pm 0.0$ | $68.3 \pm 22.6$ | $0.0 \pm 0.0$ |
| | experiment-5 | $42.9 \pm 9.6$ | $0.0 \pm 0.0$ | $86.1 \pm 1.0$ | $0.0 \pm 0.0$ | $32.2 \pm 3.6$ | $0.0 \pm 0.0$ |
| C3 | experiment-1 | $40.5 \pm 18.8$ | $1.2 \pm 0.5$ | $85.1 \pm 1.6$ | $1.8 \pm 0.1$ | $91.7 \pm 3.3$ | $1.2 \pm 0.1$ |
| | experiment-2 | $68.1 \pm 29.5$ | $15.0 \pm 7.5$ | $79.0 \pm 4.7$ | $19.8 \pm 1.8$ | $98.4 \pm 0.6$ | $24.5 \pm 1.1$ |
| | experiment-3 | $51.4 \pm 1.2$ | $2.6 \pm 0.4$ | $77.5 \pm 3.2$ | $2.6 \pm 0.3$ | $82.7 \pm 5.9$ | $5.5 \pm 0.8$ |
| | experiment-4 | $89.7 \pm 8.9$ | $22.0 \pm 4.7$ | $85.3 \pm 1.5$ | $11.3 \pm 1.5$ | $63.3 \pm 10.3$ | $6.0 \pm 4.6$ |
| | experiment-5 | $66.7 \pm 14.1$ | $0.6 \pm 0.3$ | $82.9 \pm 3.2$ | $0.5 \pm 0.0$ | $84.3 \pm 9.9$ | $1.8 \pm 0.7$ |

Table 6: ID and OOD test grid accuracy with SEM (Standard Error of the Mean) for the *CompGen* experiments (C1–C3) across the models Grid-TF, PL-TF, and LLaDA.

| Setting | Experiment | ResNet | | Vanilla-TF | |
|---------|-----------|--------|--------|------------|--------|
| | | ID | OOD | ID | OOD |
| **G1** | experiment-1 | $100.0 \pm 0.0$ | $100.0 \pm 0.0$ | $100.0 \pm 0.0$ | $100.0 \pm 0.0$ |
| | experiment-2 | $99.8 \pm 0.0$ | $96.5 \pm 0.1$ | $97.3 \pm 0.2$ | $55.4 \pm 6.6$ |
| | experiment-3 | $99.8 \pm 0.1$ | $98.5 \pm 0.2$ | $97.5 \pm 0.6$ | $77.5 \pm 3.4$ |
| | experiment-4 | $100.0 \pm 0.0$ | $99.9 \pm 0.0$ | $98.9 \pm 0.6$ | $77.9 \pm 3.5$ |
| | experiment-5 | $99.9 \pm 0.1$ | $99.3 \pm 0.1$ | $98.2 \pm 0.3$ | $83.7 \pm 1.4$ |
| **G2** | experiment-1 | $100.0 \pm 0.0$ | $99.9 \pm 0.1$ | $98.7 \pm 0.3$ | $1.9 \pm 1.9$ |
| | experiment-2 | $88.9 \pm 1.3$ | $78.2 \pm 7.2$ | $57.1 \pm 21.5$ | $0.2 \pm 0.2$ |
| | experiment-3 | $99.3 \pm 0.3$ | $98.2 \pm 0.8$ | $81.1 \pm 11.3$ | $8.2 \pm 5.4$ |
| | experiment-4 | $100.0 \pm 0.0$ | $100.0 \pm 0.0$ | $96.3 \pm 1.5$ | $0.0 \pm 0.0$ |
| | experiment-5 | $100.0 \pm 0.0$ | $100.0 \pm 0.0$ | $76.9 \pm 2.3$ | $0.2 \pm 0.2$ |
| **G3** | experiment-1 | $100.0 \pm 0.0$ | $100.0 \pm 0.0$ | $100.0 \pm 0.0$ | $88.6 \pm 9.6$ |
| | experiment-2 | $93.7 \pm 0.5$ | $0.0 \pm 0.0$ | $0.0 \pm 0.0$ | $0.0 \pm 0.0$ |
| | experiment-3 | $96.2 \pm 1.9$ | $0.1 \pm 0.1$ | $2.3 \pm 0.3$ | $0.1 \pm 0.0$ |
| | experiment-4 | $98.9 \pm 1.0$ | $0.0 \pm 0.0$ | $93.0 \pm 2.2$ | $0.0 \pm 0.0$ |
| | experiment-5 | $99.4 \pm 0.0$ | $35.3 \pm 2.7$ | $92.6 \pm 1.4$ | $23.8 \pm 3.3$ |
| **G4** | experiment-1 | $100.0 \pm 0.0$ | $23.7 \pm 5.8$ | $100.0 \pm 0.0$ | $90.6 \pm 0.9$ |
| | experiment-2 | $30.7 \pm 1.2$ | $0.0 \pm 0.0$ | $3.8 \pm 2.9$ | $0.0 \pm 0.0$ |
| | experiment-3 | $58.8 \pm 2.9$ | $0.1 \pm 0.1$ | $14.8 \pm 1.0$ | $0.0 \pm 0.0$ |
| | experiment-4 | $95.2 \pm 0.6$ | $64.1 \pm 4.0$ | $51.9 \pm 2.8$ | $9.0 \pm 1.4$ |
| | experiment-5 | $97.1 \pm 0.9$ | $36.9 \pm 0.5$ | $60.4 \pm 14.7$ | $9.4 \pm 2.3$ |
| **G5** | experiment-1 | $100.0 \pm 0.0$ | $9.1 \pm 2.0$ | $97.7 \pm 0.2$ | $0.0 \pm 0.0$ |
| | experiment-2 | $98.3 \pm 0.3$ | $0.0 \pm 0.0$ | $91.9 \pm 3.3$ | $0.0 \pm 0.0$ |
| | experiment-3 | $98.3 \pm 0.6$ | $0.0 \pm 0.0$ | $32.7 \pm 14.5$ | $0.0 \pm 0.0$ |
| | experiment-4 | $100.0 \pm 0.0$ | $0.0 \pm 0.0$ | $98.3 \pm 0.9$ | $0.0 \pm 0.0$ |
| | experiment-5 | $99.7 \pm 0.2$ | $0.6 \pm 0.2$ | $41.9 \pm 18.4$ | $0.0 \pm 0.0$ |

Table 7: ID and OOD test grid accuracy with SEM (Standard Error of the Mean) as error bars for the *EnvGen* experiments (G1–G5) across the models ResNet and Vanilla-TF.

| Setting | Experiment | Grid-TF | | PL-TF | | LLaDA | |
|---|---|---|---|---|---|---|---|
| | | ID | OOD | ID | OOD | ID | OOD |
| G1 | experiment-1 | $100.0 \pm 0.0$ | $100.0 \pm 0.0$ | $100.0 \pm 0.0$ | $99.9 \pm 0.1$ | $100.0 \pm 0.0$ | $99.9 \pm 0.1$ |
| | experiment-2 | $99.7 \pm 0.0$ | $92.7 \pm 1.3$ | $90.2 \pm 5.0$ | $90.0 \pm 4.2$ | $99.0 \pm 0.3$ | $83.5 \pm 3.6$ |
| | experiment-3 | $99.8 \pm 0.1$ | $95.3 \pm 0.4$ | $84.1 \pm 1.7$ | $82.0 \pm 1.4$ | $99.1 \pm 0.1$ | $84.2 \pm 2.6$ |
| | experiment-4 | $99.2 \pm 0.1$ | $88.5 \pm 2.5$ | $96.7 \pm 1.3$ | $72.5 \pm 9.6$ | $99.6 \pm 0.0$ | $88.9 \pm 1.4$ |
| | experiment-5 | $97.8 \pm 0.5$ | $73.9 \pm 4.9$ | $98.0 \pm 0.8$ | $82.9 \pm 2.2$ | $99.8 \pm 0.1$ | $94.1 \pm 1.5$ |
| G2 | experiment-1 | $100.0 \pm 0.0$ | $90.5 \pm 4.9$ | $100.0 \pm 0.0$ | $38.4 \pm 11.2$ | $100.0 \pm 0.0$ | $60.6 \pm 6.6$ |
| | experiment-2 | $97.1 \pm 1.8$ | $68.5 \pm 3.6$ | $96.4 \pm 3.4$ | $75.9 \pm 2.8$ | $95.5 \pm 1.3$ | $45.3 \pm 3.4$ |
| | experiment-3 | $97.9 \pm 2.1$ | $90.6 \pm 5.8$ | $85.7 \pm 7.3$ | $65.4 \pm 17.3$ | $97.7 \pm 1.8$ | $47.8 \pm 16.8$ |
| | experiment-4 | $99.5 \pm 0.2$ | $94.2 \pm 1.1$ | $98.5 \pm 1.1$ | $67.2 \pm 5.0$ | $99.7 \pm 0.1$ | $89.7 \pm 1.2$ |
| | experiment-5 | $95.6 \pm 1.1$ | $41.4 \pm 6.2$ | $82.2 \pm 4.1$ | $27.7 \pm 6.1$ | $99.6 \pm 0.2$ | $68.7 \pm 7.8$ |
| G3 | experiment-1 | $100.0 \pm 0.0$ | $99.7 \pm 0.3$ | $100.0 \pm 0.0$ | $100.0 \pm 0.0$ | $100.0 \pm 0.0$ | $99.1 \pm 0.5$ |
| | experiment-2 | $33.4 \pm 33.2$ | $0.0 \pm 0.0$ | $90.0 \pm 1.1$ | $0.0 \pm 0.0$ | $18.8 \pm 18.5$ | $0.0 \pm 0.0$ |
| | experiment-3 | $97.5 \pm 0.3$ | $0.1 \pm 0.0$ | $83.1 \pm 10.9$ | $0.1 \pm 0.1$ | $96.8 \pm 0.6$ | $0.1 \pm 0.1$ |
| | experiment-4 | $98.2 \pm 1.0$ | $0.0 \pm 0.0$ | $98.4 \pm 0.6$ | $0.0 \pm 0.0$ | $97.2 \pm 1.3$ | $0.0 \pm 0.0$ |
| | experiment-5 | $95.7 \pm 1.4$ | $33.1 \pm 2.4$ | $88.3 \pm 5.8$ | $35.9 \pm 3.1$ | $99.2 \pm 0.5$ | $34.7 \pm 0.8$ |
| G4 | experiment-1 | $100.0 \pm 0.0$ | $79.4 \pm 9.2$ | $100.0 \pm 0.0$ | $100.0 \pm 0.0$ | $100.0 \pm 0.0$ | $99.6 \pm 0.3$ |
| | experiment-2 | $79.6 \pm 5.2$ | $0.0 \pm 0.0$ | $87.5 \pm 1.4$ | $0.1 \pm 0.1$ | $28.3 \pm 28.2$ | $0.0 \pm 0.0$ |
| | experiment-3 | $61.8 \pm 30.9$ | $0.0 \pm 0.0$ | $86.0 \pm 0.7$ | $19.3 \pm 4.2$ | $88.5 \pm 3.6$ | $0.9 \pm 0.2$ |
| | experiment-4 | $97.0 \pm 1.0$ | $15.2 \pm 6.3$ | $82.3 \pm 0.4$ | $35.8 \pm 3.5$ | $91.6 \pm 2.2$ | $24.7 \pm 2.9$ |
| | experiment-5 | $96.0 \pm 0.8$ | $0.1 \pm 0.1$ | $78.0 \pm 10.3$ | $32.3 \pm 10.0$ | $98.3 \pm 0.7$ | $34.7 \pm 6.7$ |
| G5 | experiment-1 | $100.0 \pm 0.0$ | $1.1 \pm 0.5$ | $100.0 \pm 0.0$ | $44.4 \pm 12.7$ | $99.9 \pm 0.1$ | $50.4 \pm 11.1$ |
| | experiment-2 | $77.8 \pm 13.6$ | $0.0 \pm 0.0$ | $57.8 \pm 25.9$ | $0.0 \pm 0.0$ | $12.6 \pm 8.3$ | $0.0 \pm 0.0$ |
| | experiment-3 | $98.8 \pm 0.4$ | $0.0 \pm 0.0$ | $56.0 \pm 25.7$ | $0.0 \pm 0.0$ | $43.3 \pm 10.3$ | $0.0 \pm 0.0$ |
| | experiment-4 | $100.0 \pm 0.0$ | $0.0 \pm 0.0$ | $98.1 \pm 0.8$ | $0.0 \pm 0.0$ | $95.0 \pm 1.4$ | $0.0 \pm 0.0$ |
| | experiment-5 | $98.3 \pm 0.5$ | $0.0 \pm 0.0$ | $91.6 \pm 0.4$ | $0.0 \pm 0.0$ | $99.3 \pm 0.4$ | $0.0 \pm 0.0$ |

Table 8: ID and OOD test grid accuracy with SEM (Standard Error of the Mean) for the *EnvGen* experiments (G1–G5) across the models Grid-TF, PL-TF, and LLaDA.

# L LLM PROMPTS

## L.1 PROMPTS PER EXPERIMENT SETTING

### L.1.1 IN-DISTRIBUTION TASKS (COMPGEN-ID1 AND COMPGEN-ID2):

```
BASE_PROMPT_IDCOMP = """You are a system specialized in solving object-
    transformation puzzles on ASCII grids.

TASK:
1.You will see several input-output pairs as ASCII grids (numbers represent
    colors/objects)
2.Each pair includes labeled transformations explaining how input becomes
    output
3.You must infer what each transformation does from these examples
4.For the test input, you will be told which specific transformations to apply.
5.Apply ONLY the specified transformations to the test input in the order given

OUTPUT FORMAT:
Return ONLY the resulting grid as a Python list of lists.
- No explanations
- No markdown code blocks
- No text before or after
- Just the raw list, e.g.: [[0, 1], [1, 0]]
"""
```

### L.1.2 OUT-OF-DISTRIBUTION TASKS (COMPGEN-OOD1, COMPGEN-OOD2, COMPGEN-OOD3):

```
BASE_PROMPT_OODCOMP = """You are a system specialized in solving object-
    transformation puzzles on ASCII grids.

TASK:
1. You will see several inputoutput pairs as ASCII grids (numbers represent
    colors/objects)
2. Each pair includes labeled transformations explaining how input becomes
    output
3. You must infer what each transformation does from these examples
4. For the test input, you will be told which specific transformations to apply
     (possibly in a new combination not seen during training).
"IMPORTANT: The test uses out-of-distribution composition. Each individual
    transformation appears in the training examples, but the test requires
    combining them in a way not seen during training. You must:
- Identify what each transformation does independently
- Apply the specified transformations to the test input, even though this exact
     combination is new"
5. Apply ONLY the specified transformations to the test input in the order
    given

OUTPUT FORMAT:
Return ONLY the resulting grid as a Python list of lists.
- No explanations
- No markdown code blocks
- No text before or after
- Just the raw list, e.g.: [[0, 1], [1, 0]]
```

```
"""
```

### L.1.3 OBJECT GENERALIZATION (ENVGEN-OBJECTS):

```
BASE\_PROMPT\_OBJGEN = """You are a system specialized in solving object-
    transformation puzzles on ASCII grids.

TASK:
1. You will see several inputoutput pairs as ASCII grids (numbers represent
    colors/objects)
2. Each pair includes labeled transformations explaining how input becomes
    output
3. You must infer how each transformation affects individual objects
4. Apply the same transformation(s) to the test input

IMPORTANT: The test evaluates object-count generalization. The transformation
    logic is identical to training, but the test input contains more objects
    than any training example. You must apply the learned transformation to
    each object, regardless of how many are present.

OUTPUT FORMAT:
Return ONLY the resulting grid as a Python list of lists.
- No explanations
- No markdown code blocks
- No text before or after
- Just the raw list, e.g.: [[0, 1], [1, 0]]
"""
```

### L.1.4 GRID GENERALIZATION (ENVGEN-GRID):

```
BASE\_PROMPT\_GRIDGEN = """You are a system specialized in solving object-
    transformation puzzles on ASCII grids.

TASK:
1. You will see several inputoutput pairs as ASCII grids (numbers represent
    colors/objects)
2. Each pair includes labeled transformations explaining how input becomes
    output
3. You must infer how each transformation affects objects and the grid
4. Apply the same transformation(s) to the test input

IMPORTANT: The test evaluates grid-size generalization. The transformation
    logic is identical to training, but the test grid is a different size than
    any training example. You must apply the learned transformation correctly
    regardless of grid dimensions.

OUTPUT FORMAT:
Return ONLY the resulting grid as a Python list of lists.
- No explanations
- No markdown code blocks
- No text before or after
- Just the raw list, e.g.: [[0, 1], [1, 0]]
"""
```

## L.2 FULL PROMPT EXAMPLES

### L.2.1 FULL "EXPLICIT" PROMPT EXAMPLE (COMPGEN-OOD-1-1)

```
    You are a system specialized in solving object-transformation puzzles on
    ASCII grids.

TASK:
1. You will see several inputoutput pairs as ASCII grids (numbers represent
    colors/objects)
2. Each pair includes labeled transformations explaining how input becomes
    output
3. You must infer what each transformation does from these examples
4. For the test input, you will be told which specific transformations to apply
     (possibly in a new combination not seen during training).
"IMPORTANT: The test uses out-of-distribution composition. Each individual
    transformation appears in the training examples, but the test requires
    combining them in a way not seen during training. You must:
- Identify what each transformation does independently
- Apply the specified transformations to the test input, even though this exact
     combination is new"
5. Apply ONLY the specified transformations to the test input in the order
    given

OUTPUT FORMAT:
Return ONLY the resulting grid as a Python list of lists.
- No explanations
- No markdown code blocks
- No text before or after
- Just the raw list, e.g.: [[0, 1], [1, 0]]
Here is a new example with transformations: ['translate_up', 'mirror_horizontal
    '].
Input 1[[0, 0, 0, 0, 0, 0, 0, 0, 0, 0, 0, 0, 0, 0, 0], [0, 0, 0, 0, 0, 0, 0, 0,
    0, 0, 0, 0, 0, 0, 0], [0, 0, 0, 0, 0, 0, 0, 0, 0, 0, 0, 0, 0, 0, 0], [0, 0,
    0, 0, 0, 0, 0, 0, 0, 0, 0, 0, 0, 0, 0], [0, 0, 0, 0, 0, 0, 0, 0, 0, 0, 0,
    0, 0, 0, 0], [0, 0, 0, 0, 0, 0, 0, 0, 0, 0, 6, 0, 0, 0, 0], [0, 0, 3, 0, 0,
    0, 0, 0, 0, 0, 0, 6, 0, 0, 0], [0, 2, 9, 5, 0, 0, 0, 0, 0, 0, 0, 0, 0, 0,
    0], [9, 2, 2, 2, 7, 0, 0, 0, 0, 0, 0, 0, 0, 0, 0], [0, 5, 5, 4, 0, 0, 0, 0,
    0, 0, 0, 0, 0, 0, 0], [0, 0, 2, 0, 0, 0, 0, 0, 0, 0, 0, 0, 0, 0, 0], [0, 0,
    0, 0, 0, 0, 0, 0, 0, 0, 0, 0, 0, 0, 0], [0, 0, 0, 0, 0, 0, 0, 0, 0, 0, 0, 0,
    0, 0, 0], [0, 0, 0, 0, 0, 0, 0, 0, 0, 0, 0, 0, 0, 0, 0], [0, 0, 0, 0, 0,
    0, 0, 0, 0, 0, 0, 0, 0, 0, 0]]
Output 1[[0, 0, 0, 0, 0, 0, 0, 0, 0, 0, 0, 0, 0, 0, 0], [0, 0, 0, 0, 0, 0, 0, 0,
    0, 0, 0, 0, 0, 0, 0], [0, 0, 0, 0, 0, 0, 0, 0, 0, 0, 0, 0, 0, 0, 0], [0, 0,
    0, 0, 0, 0, 0, 0, 0, 0, 0, 0, 0, 0, 0], [0, 0, 0, 0, 0, 0, 0, 0, 0, 0, 0, 0,
    6, 0, 0, 0], [0, 0, 2, 0, 0, 0, 0, 0, 0, 6, 0, 0, 0, 0], [0, 5, 5, 4, 0,
    0, 0, 0, 0, 0, 0, 0, 0, 0, 0], [9, 2, 2, 2, 7, 0, 0, 0, 0, 0, 0, 0, 0, 0, 0,
    0], [0, 2, 9, 5, 0, 0, 0, 0, 0, 0, 0, 0, 0, 0, 0], [0, 0, 3, 0, 0, 0, 0, 0,
    0, 0, 0, 0, 0, 0, 0], [0, 0, 0, 0, 0, 0, 0, 0, 0, 0, 0, 0, 0, 0, 0], [0, 0,
    0, 0, 0, 0, 0, 0, 0, 0, 0, 0, 0, 0], [0, 0, 0, 0, 0, 0, 0, 0, 0, 0, 0, 0, 0,
    0, 0, 0], [0, 0, 0, 0, 0, 0, 0, 0, 0, 0, 0, 0, 0, 0, 0], [0, 0, 0, 0, 0,
    0, 0, 0, 0, 0, 0, 0, 0, 0, 0]]
Here is a new example with transformations: ['rot90', 'rot90'].
Input 2[[0, 0, 0, 0, 0, 0, 0, 0, 0, 0, 0, 0, 0, 0, 0], [0, 0, 0, 0, 0, 0, 0, 0,
    0, 0, 0, 0, 0, 0, 0], [0, 0, 0, 0, 0, 7, 7, 7, 7, 0, 0, 0, 0, 0, 0], [0, 0,
```

```
         0, 0, 0, 7, 0, 7, 7, 0, 0, 0, 0, 0, 0], [0, 0, 0, 0, 0, 0, 0, 0, 7, 0, 0,
         0, 0, 0, 0], [0, 0, 0, 0, 0, 0, 0, 7, 7, 0, 0, 0, 0, 0, 0], [0, 0, 0, 0, 0,
         0, 0, 0, 0, 0, 0, 0, 0, 0, 0], [0, 0, 0, 0, 0, 0, 0, 0, 0, 0, 0, 0, 0, 0,
         0], [0, 0, 0, 0, 0, 0, 0, 0, 0, 0, 0, 0, 0, 0, 0], [0, 0, 0, 0, 0, 0, 0, 0,
         0, 0, 0, 0, 0, 0, 0], [0, 0, 0, 0, 9, 9, 5, 5, 0, 0, 0, 0, 0, 0, 0], [0, 0,
         0, 0, 9, 9, 5, 5, 0, 0, 0, 0, 0, 0, 0], [0, 0, 0, 0, 9, 9, 5, 5, 0, 0, 0,
         0, 0, 0, 0], [0, 0, 0, 0, 9, 9, 5, 5, 0, 0, 0, 0, 0, 0, 0], [0, 0, 0, 0, 0,
         0, 0, 0, 0, 0, 0, 0, 0, 0, 0]]
Output 2[[0, 0, 0, 0, 0, 0, 0, 0, 0, 0, 0, 0, 0, 0, 0], [0, 0, 0, 0, 0, 0, 0, 0,
         0, 0, 0, 0, 0, 0, 0], [0, 0, 0, 0, 0, 7, 7, 0, 0, 0, 0, 0, 0, 0, 0], [0, 0,
         0, 0, 0, 7, 0, 0, 0, 0, 0, 0, 0, 0, 0], [0, 0, 0, 0, 0, 7, 7, 0, 7, 0, 0,
         0, 0, 0, 0], [0, 0, 0, 0, 0, 7, 7, 7, 7, 0, 0, 0, 0, 0, 0], [0, 0, 0, 0, 0,
         0, 0, 0, 0, 0, 0, 0, 0, 0, 0], [0, 0, 0, 0, 0, 0, 0, 0, 0, 0, 0, 0, 0, 0,
         0], [0, 0, 0, 0, 0, 0, 0, 0, 0, 0, 0, 0, 0, 0, 0], [0, 0, 0, 0, 0, 0, 0, 0,
         0, 0, 0, 0, 0, 0, 0], [0, 0, 0, 0, 5, 5, 9, 9, 0, 0, 0, 0, 0, 0, 0], [0, 0,
         0, 0, 5, 5, 9, 9, 0, 0, 0, 0, 0, 0, 0], [0, 0, 0, 0, 5, 5, 9, 9, 0, 0, 0,
         0, 0, 0, 0], [0, 0, 0, 0, 5, 5, 9, 9, 0, 0, 0, 0, 0, 0, 0], [0, 0, 0, 0, 0,
         0, 0, 0, 0, 0, 0, 0, 0, 0, 0]]
Here is a new example with transformations: ['translate_up'].
Input 3[[0, 0, 0, 0, 0, 0, 0, 0, 0, 0, 0, 0, 0, 0, 0], [0, 0, 0, 0, 0, 0, 0, 0,
         0, 0, 0, 0, 0, 0, 0], [0, 0, 0, 0, 0, 0, 0, 0, 0, 0, 0, 0, 0, 0, 0], [0, 0,
         0, 0, 0, 0, 0, 0, 0, 0, 0, 0, 0, 0, 0], [0, 0, 0, 0, 0, 7, 1, 0, 0, 0, 0,
         0, 0, 0, 0], [0, 0, 0, 0, 0, 7, 1, 0, 0, 0, 0, 0, 0, 0, 0], [0, 0, 0, 0, 0,
         7, 1, 0, 0, 0, 0, 0, 0, 0, 0], [0, 0, 0, 0, 0, 7, 1, 0, 0, 0, 0, 0, 0, 0,
         0], [0, 0, 0, 0, 0, 7, 1, 0, 0, 0, 0, 0, 0, 0, 0], [0, 0, 0, 0, 0, 0, 0, 0,
         0, 0, 0, 0, 0, 0, 0], [0, 0, 0, 0, 0, 0, 0, 0, 0, 0, 0, 0, 0, 0, 0], [0, 0,
         0, 0, 0, 0, 0, 0, 0, 0, 0, 0, 0, 0, 0], [0, 0, 0, 0, 0, 0, 0, 2, 2, 0, 0,
         0, 0, 0], [0, 0, 0, 0, 0, 0, 1, 1, 1, 1, 0, 0, 0, 0, 0], [0, 0, 0, 0, 0,
         0, 0, 2, 2, 0, 0, 0, 0, 0, 0]]
Output 3[[0, 0, 0, 0, 0, 0, 0, 0, 0, 0, 0, 0, 0, 0, 0], [0, 0, 0, 0, 0, 0, 0, 0,
         0, 0, 0, 0, 0, 0, 0], [0, 0, 0, 0, 0, 0, 0, 0, 0, 0, 0, 0, 0, 0, 0], [0, 0,
         0, 0, 0, 7, 1, 0, 0, 0, 0, 0, 0, 0, 0], [0, 0, 0, 0, 0, 7, 1, 0, 0, 0, 0,
         0, 0, 0], [0, 0, 0, 0, 0, 7, 1, 0, 0, 0, 0, 0, 0, 0, 0], [0, 0, 0, 0, 0,
         7, 1, 0, 0, 0, 0, 0, 0, 0, 0], [0, 0, 0, 0, 0, 7, 1, 0, 0, 0, 0, 0, 0, 0,
         0], [0, 0, 0, 0, 0, 0, 0, 0, 0, 0, 0, 0, 0, 0, 0], [0, 0, 0, 0, 0, 0, 0, 0,
         0, 0, 0, 0, 0, 0, 0], [0, 0, 0, 0, 0, 0, 0, 0, 0, 0, 0, 0, 0, 0, 0], [0, 0,
         0, 0, 0, 0, 2, 2, 0, 0, 0, 0, 0, 0], [0, 0, 0, 0, 0, 0, 1, 1, 1, 1, 0,
         0, 0, 0, 0], [0, 0, 0, 0, 0, 0, 2, 2, 0, 0, 0, 0, 0, 0], [0, 0, 0, 0, 0,
         0, 0, 0, 0, 0, 0, 0, 0, 0, 0]]
Here is a new example with transformations: ['rot90'].
Input 4[[0, 0, 0, 0, 0, 0, 0, 0, 0, 0, 0, 0, 0, 0, 0], [0, 0, 0, 0, 0, 0, 0, 0,
         0, 0, 0, 0, 0, 0, 0], [0, 0, 0, 0, 0, 0, 0, 0, 0, 0, 0, 0, 0, 0, 0], [0, 0,
         0, 0, 0, 0, 0, 0, 0, 0, 0, 0, 0], [0, 0, 0, 0, 0, 0, 0, 0, 0, 0, 0, 0, 0, 0,
         0, 0, 0], [0, 0, 0, 0, 0,
         0, 6, 6, 6, 0, 0, 0, 0, 0, 0], [0, 0, 0, 0, 0, 0, 0, 6, 6, 0, 0, 0, 0, 0,
         0], [0, 0, 9, 3, 0, 0, 0, 0, 6, 0, 0, 0, 0, 0, 0], [0, 3, 9, 3, 0, 0, 0, 0,
         0, 0, 0, 0, 0, 0, 0], [0, 3, 9, 0, 0, 0, 0, 0, 0, 0, 0, 0, 0, 0, 0], [0, 0,
         0, 0, 0, 0, 0, 0, 0, 0, 0, 0, 0, 0, 0], [0, 0, 0, 0, 0, 0, 0, 0, 0, 0, 0,
         0, 0, 0], [0, 0, 0, 0, 0, 0, 0, 0, 0, 0, 0, 0, 0, 0, 0], [0, 0, 0, 0, 0,
         0, 0, 0, 0, 0, 0, 0, 0, 0]]
Output 4[[0, 0, 0, 0, 0, 0, 0, 0, 0, 0, 0, 0, 0, 0, 0], [0, 0, 0, 0, 0, 0, 0, 0,
         0, 0, 0, 0, 0, 0, 0], [0, 0, 0, 0, 0, 0, 0, 0, 0, 0, 0, 0, 0, 0, 0], [0, 0,
         0, 0, 0, 0, 0, 0, 0, 0, 0, 0, 0, 0], [0, 0, 0, 0, 0, 0, 0, 0, 0, 0, 0, 0,
         0, 0, 0], [0, 0, 0, 0, 0, 0, 0, 0, 0, 0, 0, 0, 0, 0, 0], [0, 0, 0, 0, 0,
         0, 6, 6, 6, 0, 0, 0, 0, 0, 0], [0, 0, 0, 0, 0, 0, 6, 6, 0, 0, 0, 0, 0, 0,
```

```
0], [0, 3, 3, 0, 0, 0, 6, 0, 0, 0, 0, 0, 0, 0, 0], [0, 9, 9, 9, 0, 0, 0, 0,
    0, 0, 0, 0, 0, 0, 0], [0, 0, 3, 3, 0, 0, 0, 0, 0, 0, 0, 0, 0, 0, 0], [0, 0,
    0, 0, 0, 0, 0, 0, 0, 0, 0, 0, 0, 0, 0], [0, 0, 0, 0, 0, 0, 0, 0, 0, 0, 0,
    0, 0, 0, 0], [0, 0, 0, 0, 0, 0, 0, 0, 0, 0, 0, 0, 0, 0, 0], [0, 0, 0, 0, 0,
    0, 0, 0, 0, 0, 0, 0, 0, 0, 0]]
Here is a new example with transformations: ['mirror_horizontal'].
Input 5[[0, 0, 0, 0, 0, 0, 0, 0, 0, 0, 0, 0, 0, 0, 0], [0, 0, 0, 0, 0, 0, 0, 0,
    0, 0, 0, 0, 0, 0, 0], [0, 0, 0, 0, 0, 0, 0, 0, 0, 0, 0, 0, 0, 0, 0], [0, 0,
    0, 0, 0, 0, 0, 0, 0, 0, 0, 0, 0], [0, 0, 0, 0, 0, 0, 0, 0, 0, 0, 0, 0, 0,
    0, 0, 0], [0, 0, 0, 0, 0, 0, 0, 0, 0, 0, 0, 0, 0, 0, 0], [0, 0, 0, 0, 0, 0,
    0, 0, 0, 0, 0, 0, 0, 0, 0, 0], [0, 0, 0, 0, 0, 0, 0, 0, 0, 0, 0, 0, 0, 0, 0,
    0], [0, 0, 0, 0, 0, 0, 0, 0, 0, 0, 0, 0, 0, 0, 0, 0], [0, 0, 0, 0, 0, 0, 0,
    0, 0, 0, 0, 0, 0, 0, 0], [0, 0, 0, 0, 0, 0, 0, 0, 0, 0, 0, 0, 0, 0, 0], [0, 0,
    0, 0, 2, 3, 2, 0, 0, 0, 0, 0, 0, 0], [0, 0, 0, 0, 0, 2, 3, 2, 0, 0, 0, 0,
    0, 0, 0, 0], [0, 0, 0, 0, 2, 3, 2, 0, 0, 0, 5, 0, 0, 0], [0, 0, 0, 0, 0,
    0, 0, 0, 0, 9, 5, 0, 0, 0]]
Output 5[[0, 0, 0, 0, 0, 0, 0, 0, 0, 0, 0, 0, 0, 0, 0], [0, 0, 0, 0, 0, 0, 0, 0,
    0, 0, 0, 0, 0, 0], [0, 0, 0, 0, 0, 0, 0, 0, 0, 0, 0, 0, 0, 0, 0], [0, 0,
    0, 0, 0, 0, 0, 0, 0, 0, 0, 0, 0, 0, 0], [0, 0, 0, 0, 0, 0, 0, 0, 0, 0, 0, 0,
    0, 0, 0, 0], [0, 0, 0, 0, 0, 0, 0, 0, 0, 0, 0, 0, 0, 0, 0], [0, 0, 0, 0, 0,
    0, 0, 0, 0, 0, 0, 0, 0, 0, 0, 0], [0, 0, 0, 0, 0, 0, 0, 0, 0, 0, 0, 0, 0, 0, 0,
    0], [0, 0, 0, 0, 0, 0, 0, 0, 0, 0, 0, 0, 0, 0, 0, 0], [0, 0,
    0, 0, 2, 3, 2, 0, 0, 0, 0, 0, 0, 0], [0, 0, 0, 0, 2, 3, 2, 0, 0, 0, 0,
    0, 0, 0, 0], [0, 0, 0, 0, 2, 3, 2, 0, 0, 9, 5, 0, 0, 0], [0, 0, 0, 0, 0,
    0, 0, 0, 0, 5, 0, 0, 0]]
Now, apply the following transformations: ['translate_up', 'rot90'] to the test
    input below.
Test Input: [[0, 0, 0, 0, 0, 0, 0, 0, 0, 0, 0, 0, 0, 0, 0], [0, 0, 0, 0, 0, 0,
    0, 0, 0, 0, 0, 0, 0, 0, 0], [0, 0, 0, 0, 0, 0, 0, 0, 0, 0, 0, 0, 0, 0, 0],
    [0, 0, 0, 0, 0, 0, 0, 9, 0, 4, 3, 0, 0, 0, 0], [0, 0, 0, 0, 0, 0, 0, 3, 4,
    4, 3, 0, 0, 0, 0], [0, 0, 0, 0, 0, 0, 0, 0, 0, 0, 4, 0, 0, 0, 0, 0], [0, 0, 0,
    0, 0, 0, 0, 0, 0, 3, 9, 0, 0, 0, 0], [0, 0, 0, 0, 0, 0, 0, 0, 0, 0, 0, 0, 0,
    0, 0], [0, 0, 0, 0, 0, 0, 0, 0, 0, 0, 0, 0, 0, 0, 0], [0, 0, 0, 0, 0, 0, 0,
    0, 0, 0, 0, 0, 0, 0, 0], [0, 0, 0, 0, 0, 5, 0, 0, 0, 0, 0, 0, 0, 0, 0],
    [0, 0, 0, 0, 5, 5, 0, 0, 0, 0, 0, 0, 0, 0, 0], [0, 0, 0, 0, 0, 0, 0, 0, 0, 0,
    0, 0, 0, 0, 0, 0], [0, 0, 0, 0, 0, 0, 0, 0, 0, 0, 0, 0, 0, 0, 0], [0, 0, 0,
    0, 0, 0, 0, 0, 0, 0, 0, 0, 0, 0, 0]]
```

### L.2.2   FULL "CODED" PROMPT EXAMPLE (OOD-1-1)

```
You are a system specialized in solving object-transformation puzzles on
ASCII grids.

TASK:
1. You will see several inputoutput pairs as ASCII grids (numbers represent
    colors/objects)
2. Each pair includes labeled transformations explaining how input becomes
    output
3. You must infer what each transformation does from these examples
4. For the test input, you will be told which specific transformations to apply
    (possibly in a new combination not seen during training).
```

```
"IMPORTANT: The test uses out-of-distribution composition. Each individual
    transformation appears in the training examples, but the test requires
    combining them in a way not seen during training. You must:
-  Identify what each transformation does independently
-  Apply the specified transformations to the test input, even though this exact
    combination is new"
5. Apply ONLY the specified transformations to the test input in the order
    given

OUTPUT FORMAT:
Return ONLY the resulting grid as a Python list of lists.
-  No explanations
-  No markdown code blocks
-  No text before or after
-  Just the raw list, e.g.: [[0, 1], [1, 0]]
Here is a new example with transformations: ['t3', 't2'].
Input 1[[0, 0, 0, 0, 0, 0, 0, 0, 0, 0, 0, 0, 0, 0, 0], [0, 0, 0, 0, 0, 0, 0, 0,
    0, 0, 0, 0, 0, 0, 0], [0, 0, 0, 0, 0, 0, 0, 0, 0, 0, 0, 0, 0, 0, 0], [0, 0,
    0, 0, 0, 0, 0, 0, 0, 0, 0, 0, 0, 0], [0, 0, 0, 0, 0, 0, 0, 0, 0, 0, 0,
    0, 0, 0, 0], [0, 0, 0, 0, 0, 0, 0, 0, 0, 0, 6, 0, 0, 0, 0], [0, 0, 3, 0, 0,
    0, 0, 0, 0, 0, 0, 6, 0, 0, 0], [0, 2, 9, 5, 0, 0, 0, 0, 0, 0, 0, 0, 0, 0,
    0], [9, 2, 2, 2, 7, 0, 0, 0, 0, 0, 0, 0, 0, 0, 0], [0, 5, 5, 4, 0, 0, 0, 0,
    0, 0, 0, 0, 0, 0], [0, 0, 2, 0, 0, 0, 0, 0, 0, 0, 0, 0, 0, 0, 0], [0, 0,
    0, 0, 0, 0, 0, 0, 0, 0, 0, 0, 0, 0], [0, 0, 0, 0, 0, 0, 0, 0, 0, 0, 0, 0,
    0, 0, 0], [0, 0, 0, 0, 0, 0, 0, 0, 0, 0, 0, 0, 0, 0, 0], [0, 0, 0, 0, 0,
    0, 0, 0, 0, 0, 0, 0, 0, 0, 0]]
Output 1[[0, 0, 0, 0, 0, 0, 0, 0, 0, 0, 0, 0, 0, 0, 0], [0, 0, 0, 0, 0, 0, 0, 0,
    0, 0, 0, 0, 0, 0, 0], [0, 0, 0, 0, 0, 0, 0, 0, 0, 0, 0, 0, 0, 0, 0], [0, 0,
    0, 0, 0, 0, 0, 0, 0, 0, 0, 0, 0, 0], [0, 0, 0, 0, 0, 0, 0, 0, 0, 0, 0, 0,
    6, 0, 0, 0], [0, 0, 2, 0, 0, 0, 0, 0, 0, 0, 6, 0, 0, 0, 0], [0, 5, 5, 4, 0,
    0, 0, 0, 0, 0, 0, 0, 0, 0, 0], [9, 2, 2, 2, 7, 0, 0, 0, 0, 0, 0, 0, 0, 0,
    0], [0, 2, 9, 5, 0, 0, 0, 0, 0, 0, 0, 0, 0, 0, 0], [0, 0, 3, 0, 0, 0, 0, 0,
    0, 0, 0, 0, 0, 0], [0, 0, 0, 0, 0, 0, 0, 0, 0, 0, 0, 0, 0, 0, 0], [0, 0,
    0, 0, 0, 0, 0, 0, 0, 0, 0, 0, 0], [0, 0, 0, 0, 0, 0, 0, 0, 0, 0, 0, 0, 0,
    0, 0, 0], [0, 0, 0, 0, 0, 0, 0, 0, 0, 0, 0, 0, 0, 0, 0], [0, 0, 0, 0, 0,
    0, 0, 0, 0, 0, 0, 0, 0, 0, 0]]
Here is a new example with transformations: ['t1', 't1'].
Input 2[[0, 0, 0, 0, 0, 0, 0, 0, 0, 0, 0, 0, 0, 0, 0], [0, 0, 0, 0, 0, 0, 0, 0,
    0, 0, 0, 0, 0, 0, 0], [0, 0, 0, 0, 0, 7, 7, 7, 7, 0, 0, 0, 0, 0, 0], [0, 0,
    0, 0, 0, 7, 0, 7, 7, 0, 0, 0, 0, 0], [0, 0, 0, 0, 0, 0, 0, 0, 0, 7, 0, 0,
    0, 0, 0, 0], [0, 0, 0, 0, 0, 0, 0, 7, 7, 0, 0, 0, 0, 0, 0], [0, 0, 0, 0, 0,
    0, 0, 0, 0, 0, 0, 0, 0, 0, 0, 0], [0, 0, 0, 0, 0, 0, 0, 0, 0, 0, 0, 0, 0, 0,
    0], [0, 0, 0, 0, 0, 0, 0, 0, 0, 0, 0, 0, 0, 0, 0], [0, 0, 0, 0, 0, 0, 0, 0,
    0, 0, 0, 0, 0], [0, 0, 0, 0, 9, 9, 5, 5, 0, 0, 0, 0, 0, 0, 0], [0, 0,
    0, 0, 9, 9, 5, 5, 0, 0, 0, 0, 0, 0], [0, 0, 0, 0, 9, 9, 5, 5, 0, 0, 0,
    0, 0, 0], [0, 0, 0, 0, 9, 9, 5, 5, 0, 0, 0, 0, 0, 0], [0, 0, 0, 0, 0,
    0, 0, 0, 0, 0, 0, 0, 0]]
Output 2[[0, 0, 0, 0, 0, 0, 0, 0, 0, 0, 0, 0, 0, 0, 0], [0, 0, 0, 0, 0, 0, 0, 0,
    0, 0, 0, 0, 0, 0, 0], [0, 0, 0, 0, 0, 7, 7, 0, 0, 0, 0, 0, 0, 0, 0], [0, 0,
    0, 0, 0, 7, 0, 0, 0, 0, 0, 0, 0, 0], [0, 0, 0, 0, 0, 7, 7, 0, 7, 0, 0,
    0, 0, 0], [0, 0, 0, 0, 0, 7, 7, 7, 7, 0, 0, 0, 0, 0, 0], [0, 0, 0, 0, 0,
    0, 0, 0, 0, 0, 0, 0, 0, 0, 0, 0], [0, 0, 0, 0, 0, 0, 0, 0, 0, 0, 0, 0, 0, 0,
    0], [0, 0, 0, 0, 0, 0, 0, 0, 0, 0, 0, 0, 0, 0, 0], [0, 0, 0, 0, 0, 0, 0, 0,
    0, 0, 0, 0, 0], [0, 0, 0, 0, 5, 5, 9, 9, 0, 0, 0, 0, 0, 0], [0, 0,
    0, 0, 5, 5, 9, 9, 0, 0, 0, 0, 0, 0], [0, 0, 0, 0, 5, 5, 9, 9, 0, 0, 0,
```

```
0, 0, 0, 0], [0, 0, 0, 0, 5, 5, 9, 9, 0, 0, 0, 0, 0, 0, 0], [0, 0, 0, 0, 0,
0, 0, 0, 0, 0, 0, 0, 0, 0, 0]]
Here is a new example with transformations: ['t3'].
Input 3[[0, 0, 0, 0, 0, 0, 0, 0, 0, 0, 0, 0, 0, 0, 0], [0, 0, 0, 0, 0, 0, 0, 0,
0, 0, 0, 0, 0, 0, 0], [0, 0, 0, 0, 0, 0, 0, 0, 0, 0, 0, 0, 0, 0, 0], [0, 0,
0, 0, 0, 0, 0, 0, 0, 0, 0, 0, 0, 0], [0, 0, 0, 0, 0, 7, 1, 0, 0, 0, 0,
0, 0, 0, 0], [0, 0, 0, 0, 0, 7, 1, 0, 0, 0, 0, 0, 0, 0, 0], [0, 0, 0, 0, 0,
7, 1, 0, 0, 0, 0, 0, 0, 0, 0], [0, 0, 0, 0, 0, 7, 1, 0, 0, 0, 0, 0, 0, 0,
0], [0, 0, 0, 0, 0, 7, 1, 0, 0, 0, 0, 0, 0, 0, 0], [0, 0, 0, 0, 0, 0, 0, 0,
0, 0, 0, 0, 0, 0, 0], [0, 0, 0, 0, 0, 0, 0, 0, 0, 0, 0, 0, 0, 0, 0], [0, 0,
0, 0, 0, 0, 0, 0, 0, 0, 0, 0, 0, 0], [0, 0, 0, 0, 0, 0, 0, 2, 2, 0, 0,
0, 0, 0], [0, 0, 0, 0, 0, 0, 1, 1, 1, 1, 0, 0, 0, 0, 0], [0, 0, 0, 0, 0,
0, 0, 2, 2, 0, 0, 0, 0, 0, 0]]
Output 3[[0, 0, 0, 0, 0, 0, 0, 0, 0, 0, 0, 0, 0, 0, 0], [0, 0, 0, 0, 0, 0, 0, 0,
0, 0, 0, 0, 0, 0, 0], [0, 0, 0, 0, 0, 0, 0, 0, 0, 0, 0, 0, 0, 0, 0], [0, 0,
0, 0, 0, 7, 1, 0, 0, 0, 0, 0, 0, 0], [0, 0, 0, 0, 0, 7, 1, 0, 0, 0, 0,
0, 0, 0, 0], [0, 0, 0, 0, 0, 7, 1, 0, 0, 0, 0, 0, 0, 0, 0], [0, 0, 0, 0, 0,
7, 1, 0, 0, 0, 0, 0, 0, 0, 0], [0, 0, 0, 0, 0, 7, 1, 0, 0, 0, 0, 0, 0, 0,
0], [0, 0, 0, 0, 0, 0, 0, 0, 0, 0, 0, 0, 0, 0, 0], [0, 0, 0, 0, 0, 0, 0, 0,
0, 0, 0, 0, 0, 0, 0], [0, 0, 0, 0, 0, 0, 0, 0, 0, 0, 0, 0, 0, 0, 0], [0, 0,
0, 0, 0, 0, 2, 2, 0, 0, 0, 0, 0], [0, 0, 0, 0, 0, 0, 1, 1, 1, 1, 0,
0, 0, 0, 0], [0, 0, 0, 0, 0, 0, 0, 2, 2, 0, 0, 0, 0, 0], [0, 0, 0, 0, 0,
0, 0, 0, 0, 0, 0, 0, 0, 0, 0]]
Here is a new example with transformations: ['t1'].
Input 4[[0, 0, 0, 0, 0, 0, 0, 0, 0, 0, 0, 0, 0, 0, 0], [0, 0, 0, 0, 0, 0, 0, 0,
0, 0, 0, 0, 0, 0, 0], [0, 0, 0, 0, 0, 0, 0, 0, 0, 0, 0, 0, 0, 0, 0], [0, 0,
0, 0, 0, 0, 0, 0, 0, 0, 0, 0, 0, 0], [0, 0, 0, 0, 0, 0, 0, 0, 0, 0, 0, 0,
0, 0, 0, 0], [0, 0, 0, 0, 0, 0, 0, 0, 0, 0, 0, 0, 0, 0, 0], [0, 0, 0, 0, 0,
0, 6, 6, 6, 0, 0, 0, 0, 0, 0], [0, 0, 0, 0, 0, 0, 0, 0, 6, 6, 0, 0, 0, 0, 0,
0], [0, 0, 9, 3, 0, 0, 0, 0, 6, 0, 0, 0, 0, 0, 0], [0, 3, 9, 3, 0, 0, 0, 0,
0, 0, 0, 0, 0, 0], [0, 3, 9, 0, 0, 0, 0, 0, 0, 0, 0, 0, 0, 0, 0], [0, 0,
0, 0, 0, 0, 0, 0, 0, 0, 0, 0, 0, 0], [0, 0, 0, 0, 0, 0, 0, 0, 0, 0, 0, 0,
0, 0, 0, 0], [0, 0, 0, 0, 0, 0, 0, 0, 0, 0, 0, 0, 0, 0], [0, 0, 0, 0, 0,
0, 0, 0, 0, 0, 0, 0, 0, 0, 0]]
Output 4[[0, 0, 0, 0, 0, 0, 0, 0, 0, 0, 0, 0, 0, 0, 0], [0, 0, 0, 0, 0, 0, 0, 0,
0, 0, 0, 0, 0, 0, 0], [0, 0, 0, 0, 0, 0, 0, 0, 0, 0, 0, 0, 0, 0, 0], [0, 0,
0, 0, 0, 0, 0, 0, 0, 0, 0, 0, 0, 0], [0, 0, 0, 0, 0, 0, 0, 0, 0, 0, 0, 0,
0, 0, 0, 0], [0, 0, 0, 0, 0, 0, 0, 0, 0, 0, 0, 0, 0, 0, 0], [0, 0, 0, 0, 0,
0, 6, 6, 6, 0, 0, 0, 0, 0, 0], [0, 0, 0, 0, 0, 0, 6, 6, 0, 0, 0, 0, 0, 0,
0], [0, 3, 3, 0, 0, 0, 6, 0, 0, 0, 0, 0, 0, 0, 0], [0, 9, 9, 9, 0, 0, 0, 0,
0, 0, 0, 0, 0, 0], [0, 0, 3, 3, 0, 0, 0, 0, 0, 0, 0, 0, 0, 0, 0], [0, 0,
0, 0, 0, 0, 0, 0, 0, 0, 0, 0, 0, 0], [0, 0, 0, 0, 0, 0, 0, 0, 0, 0, 0, 0,
0, 0, 0], [0, 0, 0, 0, 0, 0, 0, 0, 0, 0, 0, 0, 0, 0, 0], [0, 0, 0, 0, 0,
0, 0, 0, 0, 0, 0, 0, 0, 0, 0]]
Here is a new example with transformations: ['t2'].
Input 5[[0, 0, 0, 0, 0, 0, 0, 0, 0, 0, 0, 0, 0, 0, 0], [0, 0, 0, 0, 0, 0, 0, 0,
0, 0, 0, 0, 0, 0, 0], [0, 0, 0, 0, 0, 0, 0, 0, 0, 0, 0, 0, 0, 0, 0], [0, 0,
0, 0, 0, 0, 0, 0, 0, 0, 0, 0, 0], [0, 0, 0, 0, 0, 0, 0, 0, 0, 0, 0, 0, 0, 0,
0, 0, 0], [0, 0, 0, 0, 0, 0, 0, 0, 0, 0, 0, 0, 0, 0, 0, 0, 0, 0,
0], [0, 0, 0, 0, 0, 0, 0, 0, 0, 0, 0, 0, 0, 0, 0], [0, 0, 0, 0, 0, 0, 0, 0,
0, 0, 0, 0, 0], [0, 0, 0, 0, 0, 0, 0, 0, 0, 0, 0, 0, 0, 0, 0, 0], [0, 0,
0, 0, 2, 3, 2, 0, 0, 0, 0, 0, 0], [0, 0, 0, 0, 2, 3, 2, 0, 0, 0, 0,
0, 0, 0], [0, 0, 0, 0, 2, 3, 2, 0, 0, 0, 0, 5, 0, 0, 0], [0, 0, 0, 0, 0,
0, 0, 0, 0, 0, 9, 5, 0, 0, 0]]
```

```
Output 5[[0, 0, 0, 0, 0, 0, 0, 0, 0, 0, 0, 0, 0, 0, 0], [0, 0, 0, 0, 0, 0, 0, 0,
    0, 0, 0, 0, 0, 0, 0], [0, 0, 0, 0, 0, 0, 0, 0, 0, 0, 0, 0, 0, 0, 0], [0, 0,
    0, 0, 0, 0, 0, 0, 0, 0, 0, 0, 0, 0, 0], [0, 0, 0, 0, 0, 0, 0, 0, 0, 0, 0,
    0, 0, 0, 0], [0, 0, 0, 0, 0, 0, 0, 0, 0, 0, 0, 0, 0, 0, 0], [0, 0, 0, 0, 0,
    0, 0, 0, 0, 0, 0, 0, 0, 0, 0], [0, 0, 0, 0, 0, 0, 0, 0, 0, 0, 0, 0, 0, 0,
    0], [0, 0, 0, 0, 0, 0, 0, 0, 0, 0, 0, 0, 0, 0, 0], [0, 0, 0, 0, 0, 0, 0, 0,
    0, 0, 0, 0, 0, 0, 0], [0, 0, 0, 0, 0, 0, 0, 0, 0, 0, 0, 0, 0, 0, 0], [0, 0,
    0, 0, 2, 3, 2, 0, 0, 0, 0, 0, 0, 0, 0], [0, 0, 0, 0, 2, 3, 2, 0, 0, 0, 0,
    0, 0, 0, 0], [0, 0, 0, 0, 2, 3, 2, 0, 0, 0, 9, 5, 0, 0, 0], [0, 0, 0, 0, 0,
    0, 0, 0, 0, 0, 0, 0, 5, 0, 0, 0]]
Now, apply the following transformations: ['t3', 't1'] to the test input below.
Test Input: [[0, 0, 0, 0, 0, 0, 0, 0, 0, 0, 0, 0, 0, 0, 0], [0, 0, 0, 0, 0, 0,
    0, 0, 0, 0, 0, 0, 0, 0], [0, 0, 0, 0, 0, 0, 0, 0, 0, 0, 0, 0, 0, 0, 0, 0],
    [0, 0, 0, 0, 0, 0, 0, 9, 0, 4, 3, 0, 0, 0, 0], [0, 0, 0, 0, 0, 0, 0, 3, 4,
    4, 3, 0, 0, 0, 0], [0, 0, 0, 0, 0, 0, 0, 0, 0, 0, 4, 0, 0, 0, 0, 0], [0, 0, 0,
    0, 0, 0, 0, 0, 0, 3, 9, 0, 0, 0, 0], [0, 0, 0, 0, 0, 0, 0, 0, 0, 0, 0, 0, 0,
    0, 0, 0], [0, 0, 0, 0, 0, 0, 0, 0, 0, 0, 0, 0, 0, 0, 0], [0, 0, 0, 0, 0, 0,
    0, 0, 0, 0, 0, 0, 0, 0, 0], [0, 0, 0, 0, 0, 5, 0, 0, 0, 0, 0, 0, 0, 0, 0, 0],
    [0, 0, 0, 0, 5, 5, 0, 0, 0, 0, 0, 0, 0, 0, 0], [0, 0, 0, 0, 0, 0, 0, 0, 0, 0,
    0, 0, 0, 0, 0, 0], [0, 0, 0, 0, 0, 0, 0, 0, 0, 0, 0, 0, 0, 0, 0], [0, 0, 0,
    0, 0, 0, 0, 0, 0, 0, 0, 0, 0]]
```