# OpenReview forum: "COGITAO: A Procedural Object-Centric Framework to Study Compositional Generalization"
_ICLR.cc/2026/Conference — Submitted to ICLR 2026_

### Official Review · Reviewer_mCiE · 2025-10-17

**Soundness:** 3
**Presentation:** 3
**Contribution:** 3
**Rating:** 6
**Confidence:** 4

**Summary:**

The paper proposes an object-centric procedural generative framework for studying compositional generalization of deep learning models. It features grid-based objects (but also supports "to RGB" rendering for vision models) and 28 atomic transformations that can be freely composed to generate a large number of downstream tasks. Using this framework, the paper presents two major use cases (studying compositional and environmental generalization), establishing multiple benchmarks.

**Strengths:**

- Generally, one risk of procedurally generated datasets is that most tasks may overlap. COGITAO instead presents non-redundant & non-conflicting (in a loose sense) transformations which makes it convincingly flexible and composable, if properly used.
- The chosen two use cases (and the resulting benchmarks) are meaningful and interesting to assess the intended generalization capabilities. Moreover, I see another use case (also mentioned briefly by the authors) which could broaden the scope and impact of COGITAO: it can be used to study implicit biases of architectures.
- The paper's writing quality is good, the choice of images and tables help the presentation and the contribution is clear.

**Weaknesses:**

The paper's contribution is solid. However, there are two major points that require attention.

**W1.**

Although presenting a framework to study compositional generalization, I found complete lack of acknowledgement about _provable_ object-centric learning literature & compositional generalization (eg. see [1,2] and their related works). The only referenced work I could find is a classical work on Slot attention (Locatello et al., 2020), for which a similar mechanism is implemented inside Grid-TF.

-----------------------

**W2.**

On the same note of **W1.**, being COGITAO grid-based (eventually extended to vision), I feel the paragraph "Compositional Generalization in Language" in the Related Works section is out of scope. I'd argue the paper would rather benefit from a section dedicated to compositional generalization methods.

-----------------------

**_References:_**

[1] Brady, Jack, et al. "Provably learning object-centric representations." ICML 2023.

[2] Wiedemer, Thaddäus, et al. "Provable compositional generalization for object-centric learning." ICLR 2024.

**Questions:**

Thanking in advance for their response, I'd kindly ask the authors the following questions.

- **(Continuing from W1.)** Being the dataset akin to toy settings in [1,2] the paper could benefit from trying also their methods. For instance, [2] provably allows for compositional generalization, although I'm unsure all of their assumptions might hold in the benchmarks supported by COGITAO.
- Do the authors plan to extend their code base to include also a slot-based differentiable renderer? (even implemented just as finite differences. For instance, allowing for taking derivatives of grid cells with respect to slots as done in [1,2]). This would make COGITAO also very useful for supporting theoretical research.
- I'd kindly ask the authors to address **W2.** in the Weaknesses section of this review.

---

> ### Author Response · Authors · 2025-12-03
>
> We thank the reviewer for recognizing the advantage of COGITAO in its non-redundancy and flexibility, as well as recognizing the usefulness of our two use cases (CompGen and EnvGen). We agree with reviewer mCiE that COGITAO can be used to study implicit biases of architecture - which is indeed one important reasons we believe our framework to be useful for.
>
> # W1 & Q1: Inclusion of Related Works
>
> While we did not discuss in depth the mentionned "provable object-centric learning" references, we did reference one work which aligns with this line of work mentioned by the reviewer [1] (line 037), from the same authors as referenced [2] from the reviewer’s response. The reason we do not discuss more this line of work is because of the difference in focus in aspects of compositional generalization. Indeed, we note that COGITAO and [2,3] focus on different aspects:
>
> 1. **Different compositional axis:** Wiedemer et al. [3] and Brady et al. [2] study generalization to novel combinations of *objects* (slot-supported subsets), whereas COGITAO studies generalization to novel compositions of *transformations*. In their framework, the compositional structure lies in how objects combine in scenes (representations); in ours, the focus lies in how operations compose transformations (tasks) sequentially.
> 2. **Different task formulation:** Their framework concerns with autoencoder-based representation learning with slot identifiability, while COGITAO involves supervised prediction of transformation outcomes. Their theoretical assumptions (additivity, compositional consistency) specifically constrain decoder architectures for reconstruction tasks.
>
> Essentially, the key assumptions from Wiedemer et al. (e.g. additive decoders and compositional consistency regularization) are tailored to autoencoders learning object-centric representations. Directly applying these to COGITAO would require reformulating our tasks within an autoencoder framework with explicit slot representations, which represents an interesting but architecturally different approach.
>
> We however welcome the reviewer's pertinent suggestion, and incorporated the references in our related works section to the revised manuscript, and refer to this as interesting future works.
>
> # W2: Integration of Language in Related Works
>
> We thank the reviewer for brining up this interesting discussion point. We however do believe that a section on Compositional Generalization in Language does fit in the related works section of COGITAO. First, we believe that the inductive biases needed for compositionality translate between vision and language, hence the interest in including both of them. Furthermore, COGITAO takes significant inspiration from compositional datasets that are related to language (e.g. SCAN), which is why we felt the use of discussing compositional datasets in language. Finally, as we encode our 2D grids into 1D sequence and use learning methods that overlap between vision and language, we also felt it was meaningful to discuss the Compositional Generalization problem in the context of language
>
> # Q2: Extension to a slot-based differential renderer
>
> As of now, COGITAO's generator operates with discrete grids (color indices 0–9) and procedural object placement. The generation process is not currently differentiable, as objects are sampled and placed programmatically, and transformations are applied via discrete operations. Implementing a slot-based differentiable renderer, while partially theoretically feasible, would involve a substantial amount of rework to the current work. Specifically, we would need to
>
> * Representing each object as a continuous slot vector encoding position (x, y), scale, color, and shape parameters.
> * Implementing a soft/differentiable rendering function that maps slots to grids—similar to spatial transformer networks or the additive decoders in [2]. The RGB extension we provide (Appendix B) could serve as a starting point, as continuous pixel values are more amenable to differentiation than discrete color indices.
> * Reformulate our 28 transformations as differentiable functions operating on slot representations to enable end-to-end gradient flow, - which is simply not feasible. Some transformations, such as translations or rotations could possibly be made into differentiable operations with substantial reworking, but object duplication or deletion for instance, would be practically impossible to implement in a differentiable form. As a result, the COGITAO framework would be significantly reduced in its expressivity and flexibility.
>
> This would essentially become a different system than COGITAO.
>
> # References
>
> 1. Wiedemer, et al. Compositional Generalization from First Principles. In Advances in Neural Information Processing Systems (NeurIPS)
> 2. Brady, Jack, et al. "Provably learning object-centric representations." ICML 2023.
> 3. Wiedemer, Thaddäus, et al. "Provable compositional generalization for object-centric learning." ICLR 2024.

---

### Official Review · Reviewer_E5Z9 · 2025-10-25

**Soundness:** 3
**Presentation:** 2
**Contribution:** 3
**Rating:** 4
**Confidence:** 4

**Summary:**

This work introduces COGITAO, a procedural, object-centric framework designed to generate millions of unique tasks for evaluating compositional generalization in models by composing 28 atomic transformations on grid-based objects.
Experimental results reveal that despite strong in-domain performance, state-of-the-art models, including specialized architectures like Grid Transformers and recurrent Pondering Looped Transformers, struggle significantly with out-of-distribution generalization, highlighting the challenge of systematic compositional reasoning in novel settings.

**Strengths:**

- The transformation proposed in this work is highly flexible and procedurally designed. The framework supports the composition of 28 distinct atomic transformations, such as translation, rotation, mirroring, and cropping, applied to objectis in grid-based environments, allowing to be processed in a sequence with any abitrary depth.
- This generator also allows fine-grained control over environmental parameters, including grid size, number of objects, object complexity, and color patterns, which facilitates systemtic testing of both compositional generalization and environmental generalization.
- The framework is comprehensive and extendable to Sequential-COGITAO, which outputs intermediate transformation steps, and to COGITAO-RGB, which renders tasks in a natural image format to bridge the gap to real-world vision applications.

**Weaknesses:**

- The proposed 28 atomic transformation rules are not able to perform on real-world concepts, like natural objects. They are only available for the proposed grid-like inputs.
- To improve clarity, line 111 should direct the reader to Appendices A and B for a discussion of the RGB and sequential frameworks, which are detailed there but not in the main body of the paper. Moreover, the author does not provide any experimental study on RGB and sequential settings.
- The experimental settings lack clarity in the main text. For instance, the mathematical formulation of the supervision signal mentioned in line 331 is not explicitly defined. Based on the metric described in Table 1's caption ("test grid accuracy (i.e., % of perfect matches"), I infer the task is a per-grid-cell classification. This key detail about the learning objective should be stated explicitly before.
- The manuscript's organization could be improved, particularly by providing deeper insights and analysis in the experimental results section. The proposed benchmark is supposed to show why SOTA models fail to generalize to OOD cases, especially with more composition.

**Questions:**

- In Figure 1, what is the transformation process for example task A? How to generate the output grid? What is "(mod 9 + 1) color" in line 819?
- Are there any two different transformation processes that will produce the same output grids for the same input grids?
- Where is Table 4.4 in line 345 and line 367? Is it Table 1?
- In Table 1, intuitively, C1 is easier than C3 since C1 tests on depth 2 and C3 tests on depth 3. Why does C3 have better ID and OOD performance than C1 for all the cases in Table 1?
- What are the key differences between COGITAO and CLEVR?

---

> ### Author Response · Authors · 2025-12-03
>
> We thank the reviewer for recognizing the fine-grained control of our generator and the comprehensiveness of the framework. We appreciate the constructive feedback on presentation and clarity. We have addressed the specific questions and concerns below.
>
> # W1 & W2: Applicability of COGITAO transformations to the real world and COGITAO-RGB
>
> The abstract nature of COGITAO is a deliberate design choice, following the tradition of benchmarks like ARC, SVRT, and RAVEN. By using a controlled environment, we isolate compositional reasoning capabilities from the confounding noise of visual recognition in natural images.
>
> However, the operations in COGITAO functionally correspond to various real-world tasks and solving them would advance various fields. For example, COGITAO addresses a core challenge in Offline Reinforcement Learning [5] and Vision-Language-Action (VLA) [6,7] models: learning compositional action policies from pre-recorded, static datasets.
>
> A model learning to predict the outcome of _[translate, rotate]_ is algorithmically equivalent to a VLA agent learning robotic manipulation sequences from video demonstrations. COGITAO serves as a necessary "unit test": if architectures fail to generalize compositionally in this noise-free domain, they face fundamental bottlenecks in learning robust, multi-step planning policies in noisy real-world environments.
> Functional Isomorphism: Furthermore, the operations (e.g., translation, filling, duplicating) are functionally isomorphic to standard tasks in image editing and object-centric world modeling [1, 2], where agents must distinguish independent causal mechanisms.
>
> While we agree that pixel-level statistics differ, the reasoning logic required to execute composed actions is shared. As noted by the reviewer, our RGB extension (Appendix A) is the designated bridge to natural images, which we aim to explore in future work.
>
> Note: We added the reference to the relevant appendices in line 111 as requested by reviewer E5Z9.
>
> # W2: Clarification of the learning objective and metrics
>
> We thank the reviewer for this relevant point. We added the following details in the appendix to further clarify.
>
> ## Clarification of the learning objective
>
> Our learning objective, similar to the approach in [4], is a cross-entropy loss computed as the sum over pixels. Specifically, after the final decoder layer, the output embedding $\mathbf{h}_i^N$ for patch $i$ is projected linearly. A Softmax function is then applied to predict the individual pixel-wise value $\hat{\mathbf{y}}_i$:
>
> $$\hat{\mathbf{y}}_i = \text{Softmax}(\text{Linear}(\mathbf{h}_i^N))$$
>
> The overall $\textbf{cross-entropy loss}$ ($\mathcal{L}$) is computed as the sum over all pixels $i$ where $\mathbf{y}_i$ is the ground truth:
>
> $$\mathcal{L} = - \sum_{i} \mathbf{y}_{i} \log{(\hat{\mathbf{y}}_i)}$$
>
> ## Clarification of the Metrics
>
> We reported perfect grid accuracy in our original paper. Following comments from reviewer E5Z9 and Eqxe requesting more in-depth analysis of our results, we suggest to add three more metrics - "per pixel accuracy", "object accuracy" and model "stubbornness". We present the exact formulation of Grid Accuracy as requested by the reviewer, and refer the reviewers to our revised manuscript (Appendix G) for the new metrics exact formulation.
>
> ### Metric 1: Perfect Grid Reconstruction
>
> The perfect grid reconstruction is computed as a percentage of perfectly reconstructed grids. Specifically, given predicted grid $\hat{\mathbf{Y}}_k$ and ground truth grid $\mathbf{Y}_k$
>
> $$\text{Perfect Grid Accuracy Indicator: } \quad \mathbb{I}(\hat{\mathbf{Y}}_k = \mathbf{Y}_k) =
> \begin{cases}
> 1 & \text{if all pixels in } \hat{\mathbf{Y}}_k \text{ match } \mathbf{Y}_k \\
> 0 & \text{otherwise}
> \end{cases}
> $$
>
> $$\text{GA} = \frac{1}{N} \sum_{k=1}^{N} \mathbb{I}(\hat{\mathbf{Y}}_k = \mathbf{Y}_k) \times 100\%$$
>
> # W3 & W4: Further analysis on the experimental result section
>
> We thank the reviewer for this insightful point. As it aligns closely with the elements raised in Reviewer Eqxe’s W3, we provide a thorough response in that section and kindly refer Reviewer E5Z9 to this answer, which addresses all aspects of this question in detail.

---

> ### Author Response · Authors · 2025-12-03
>
> # Q1: Clarifying the transformation process for task A and (mod 9 + 1)
>
> The transformation in Figure 1.A is "extend object". As described in our Appendix (line 819), it "Extends the outermost edges of the object with either the same color or (mod 9 + 1) color."
>
> We thank the reviewer for pointing out the need for a clearer explanation of the “mod 9 + 1” color-change rule. We briefly outline this below, and added this definition to the appendix (line 918):
>
> Object colors take values $c \in \{1, \dots, 9\}$, with $c = 0$ reserved for **background**. The color-change operation is defined as a cyclic increment over the nine object colors:
>
> $$
> c' =
> \begin{cases}
> ((c - 1) \bmod 9) + 1, & c \in \{1,\dots,9\} \\
> 0, & c = 0 .
> \end{cases}
> $$
>
> This rule **increments** each object color by one in a **cyclic fashion** (sending $9 \mapsto 1$) while ensuring that the **background value** remains unchanged.
>
> # Q2: Would two transformation processes produce the same output grids
>
> As noted in line 801-805, different transformation processes can produce the same output grids. For example, we can consider the obvious example [*translate_up, translate_up, translate_down]*, which will always be equivalent to *[translate_up]*.
> This mimics real-world redundancy; a robust model should learn that these distinct execution paths yield equivalent states. Therefore, we argue that this by no means an issue but an important evaluation property.
>
> # Q3: Table 4.4
> We thank the reviewer for bringing this to our attention. It indeed is Table 1, and will correct this typo in the manuscript.
>
> # Q4: Why does C3 have better ID than OOD performance than C1?
>
> * C1 (Specific): The C1-OOD set tests a single, specific transformation sequence (repeated across samples). If the model fails to grasp that specific logic, accuracy is 0%. It is "unforgiving."
> * C3 (Diverse): The C3-OOD set is an aggregate over all possible depth-3 combinations of known atoms. While the depth is higher, the diversity allows the model to succeed on easier combinations within that set, resulting in a higher average accuracy than the strict "all-or-nothing" nature of C1.
>
> # Key differences between COGITAO and CLEVR
>
> * **1. Generative vs. Discriminative:** CLEVR is primarily a Visual Question Answering (VQA) benchmark (output = text/label). COGITAO is a generative benchmark (output = grid/image), requiring the model to actively execute rules rather than just identify them.
> * **2. Transformation vs. Property:** CLEVR composes object properties (color, shape). COGITAO composes actions(transformations), making it more suitable for studying planning and world modeling.
> * **3. Flexible generator + Benchmark vs Benchmark Only:** COGITAO provides extensive control and flexibility, supporting composition over a set of 28 interoperable transformations at adjustable composition-depth, along with control over grid parametrization and object properties. This flexibility enables the creation of millions of unique task rules which researchers can adjust to their liking. CLEVR however offers a fixed dataset, which users are bound to use as provided.
>
> # References
>
> 1. Danijar Hafner, Jurgis Pasukonis, Jimmy Ba, and Timothy Lillicrap. Mastering diverse domains through world models.
> 2. Jake Bruce, Michael D Dennis, Ashley Edwards, Jack Parker-Holder, Yuge Shi, Edward Hughes, Matthew Lai, Aditi Mavalankar, Richie Steigerwald, Chris Apps, et al. Genie: Generative interactive environments.
> 3. Stefano Ferraro, Pietro Mazzaglia, Tim Verbelen, and Bart Dhoedt. Focus: object-centric world models for robotic manipulation.
> 4. Li, Wenhao, Yudong Xu, Scott Sanner, and Elias Boutros Khalil. "Tackling the abstraction and reasoning corpus with vision transformers: the importance of 2d representation, positions, and objects."
> 5. Hussing, Marcel, Jorge A. Mendez, Anisha Singrodia, Cassandra Kent, and Eric Eaton. "Robotic manipulation datasets for offline compositional reinforcement learning."
> 6. Li, Haoxin, and Boyang Li. "Enhancing vision-language compositional understanding with multimodal synthetic data."
> 7. Zeng, Y., Huang, Y., Zhang, J., Jie, Z., Chai, Z. and Wang, L., 2024. Investigating compositional challenges in vision-language models for visual grounding.

---

### Official Review · Reviewer_Eqxe · 2025-10-28

**Soundness:** 2
**Presentation:** 3
**Contribution:** 1
**Rating:** 2
**Confidence:** 4

**Summary:**

The paper introduces COGITAO (COmpositional Generalization In Transformations And Objects), a dataset for evaluating compositional generalization in abstract visual domains. Inspired by the ARC-AGI benchmark, COGITAO aims to provide a large-scale, controllable environment that enables scientific study of compositional generalization. The dataset consists of input–output grid pairs, where the output is generated by applying a sequence of transformations (e.g., rotation, translation) to the input. The dataset defines two generalization regimes: (1) compositional generalization, which tests generalization to novel transformation sequences, and (2) environmental generalization, which varies object and grid properties. The authors evaluate four architectures: Vanilla Transformer, Grid Transformer, Pondering Looped Transformer, and LLaDA and showed that they fail to generalize.

**Strengths:**

1. Originality: The dataset is based on prior datasets such as ARC and PGM, but introduced more diverse transformations (28 atomic total) and objects.
2. Quality: The authors clearly defined the two generalizations (compositional and environmental) and carefully designed the transformations and generalization splits. The authors also tested diverse architectures.
3. Clarity: The paper is well-organized and clearly written
4. Significance: COGITAO provides a generative framework for analyzing and improving compositional generalization in vision models. It offers a large-scale benchmark that can guide the development of models for challenging tasks such as ARC.

**Weaknesses:**

1. Motivation Clarity: The motivation for why this new benchmark is necessary could be improved. The manuscript argues that ARC’s dataset is too small and heterogeneous for systematic study. However, ARC was intentionally designed to be small to probe generalization without overfitting through large-scale training, and related benchmarks such as PGM (Hill et al., 2019) already operate at million-scale sample sizes. Thus, dataset size alone does not fully justify the need for COGITAO. To clarify its unique contribution, the authors could provide concrete examples of the new scientific questions or analyses that COGITAO enables, but prior benchmarks cannot.

2. Connection to Broader Context: The paper deliberately focuses on training models (e.g., transformers) via supervised learning and omits large pretrained models or in-context learning, arguing that training details for foundation models are opaque. While this focus is understandable from a scientific control perspective, it leaves unclear how COGITAO relates to the current frontier of model capabilities. Recent progress, such as reports of strong performance by O3 on ARC, raises the question of whether large-scale pretrained systems may already solve the generalization task. e.g. recent report on ARC by O3? If so, what specific research questions remain unresolved that make COGITAO valuable? Does the dataset primarily expose limitations of small architectures trained with supervised loss? The paper would benefit from at least a baseline evaluation or discussion of how large pretrained models perform on COGITAO.

3. Evaluation Metrics: The paper relies on exact grid match accuracy as the sole evaluation metric, but this choice is neither well-justified nor fully aligned with the goal of studying generalization mechanisms. The metric is extremely strict: a one-pixel offset or minor transformation error, results in a total failure. The authors should clarify how this metric captures the underlying compositional reasoning ability they aim to evaluate and whether alternative or complementary metrics (e.g., per-object accuracy) might provide a more informative picture of model behavior. In addition, no error analysis is presented to reveal what kinds of transformations or compositional structures cause failure, or how performance degrades with transformation sequence length.

4. Finally, there is no human evaluation. The  example tasks appear challenging even to humans. It would be valuable to include a human baseline to contextualize model performance

**Questions:**

1. Dataset Difficulty and Learning Regime: The reported in-domain accuracies are relatively low (e.g., 30% for Transformer, 60% for Grid-Transformer, 82% for PL-TF, and 84% for LLaDA). This raises the question of whether the current training regime truly enables the study of compositional generalization, or whether models are still struggling with basic task learning. Could the authors comment on whether the dataset might be too difficult or insufficiently calibrated for the intended analysis? Why were models trained for only 10 epochs given the modest dataset size

2. The models were trained on 100K examples—smaller than datasets like PGM, which operate at the million-sample scale. Given this, how did the authors determine that the dataset size was sufficient for convergence and reliable evaluation of generalization?

3. Model Selection: Why were only Transformer-based models evaluated? Have the authors considered testing architectures explicitly designed for relational reasoning, such as Relational Networks?

---

> ### Author Response · Authors · 2025-12-03
>
> We thank the reviewer for recognizing the originality of our dataset, as well as the quality and clarity of our work. We appreciate the constructive feedback regarding motivation and the broader model context. We have addressed these concerns with significant new experiments, including evaluations on frontier LLMs (e.g., O3, Gemini 3 Pro) and dataset scaling studies (up to 1M samples).
>
> # W1: Motivation Clarity
>
> **Clarifying COGITAO's Contribution:** COGITAO's main contribution is to allow controlled and flexible study of compositional generalization, focusing on both width (number of transformations) and depth (sequence length) of composition. While similar works exist in language (e.g. SCAN [1]), COGITAO is the first to enable such compositional flexibility in object-centric, visual domains.
>
> **Examples of scientific questions that COGITAO enables and that other benchmarks do not:** Our CompGen experiments (C1, C2 and C3 settings)  enable precise control over the shift between training and testing distributions. This allows us to isolate
> whether a model fails at perceiving an object or executing a sequence step. Unlike PGM (which focuses on pattern recognition/discrimination) or ARC (which relies on broad, loose priors), COGITAO tests the specific capability of generative execution of transformation rules. The core scientific question COGITAO enables (which PGM cannot) is: Can a model that perfectly perceive an object and perfectly executes a single transformation T, successfully execute the composition T_2​(T_1​(x)) without explicit supervision on that specific sequence?
>
> Thus, we summarize the differences:
> * **vs. PGM:** We test generative composition of transformations, not just discriminative reasoning.
> * **vs. ARC:** We provide controllable training distributions to isolate specific inductive biases, whereas ARC relies on opaque, broad priors.
>
> # W2: Connection to Broader Context:
>
> While we agree that there has been substantial progress with LLMs and some reasoning benchmarks such as ARC (e.g. O3, Grok, Gemini etc..), we believe that the purpose of COGITAO is not only to be "fully solved" (as would a standard benchmark), but rather to be solved *in light of a specific data training distribution*, which would underlie the capacity of given models to perform transformation composition. We see it as a “microscope” for architectural inductive biases. As such, testing models on frontier LLMs, for which we have close to no information on the training data or procedure (in addition to the known risks of data contamination), would not serve to answer our research question, and so even if frontier models come to solve our dataset.
>
> We however acknowledge that ignoring capabilities of large pretrained models leaves a gap in the context of modern AI. We have thus conducted In-Context Learning (ICL) experiments on frontier models to provide a reference baseline. We hereby summarize our findings, which are also detailed in the revised Appendix C.
>
> ## Experimental Setup:
>
> We prompted frontier LLMs in different conditions which resemble the various experiment settings we have. Specifically, we give 5 in context examples to the LLMs, as well as an input example that it must apply the novel transformation on. We do so according to the following experimental logic (see appendix C for details)
>
> * **CompGen-ID1:** context built of a set of 5 transformations . Test on one of the composed ones seen during training.
> * **CompGen-OOD1:** context with a set of 5 transformations (simple + composed). Test on a composition not seen during training.
> * **EnvGen-GridSize:** context built of 5 examples with 10x10-15x15 grids. Test on the same transformation but 16x16-20x20 grids.
>
> We tested two prompting regimes:
>
> * **Explicit:** Task codes reveal the transformation name (e.g., ["translate_up", "rotate90"])
> * **Coded:** Task codes are abstract (e.g., ["t1", "t2"]), requiring the model to infer the rule from context
>
> ## Results (with coded task embedding):
>
> | |deepseek-r1t2|openai gpt-4.1-mini|openai o3|x-ai/grok-code-fast-1|gemini-3-pro-preview|
> |:----|:----|:----|:----|:----|:----|
> |CompGen-ID1|5 / 50|1 / 50|13 / 50|1 / 49|28 / 50|
> |CompGen-ID2|11 / 49|2 / 50|20 / 50|5 / 49|27 / 50|
> |CompGen-OOD1|0 / 50|0 / 50|8 / 50|0 / 46|18 / 50|
> |CompGen-OOD2|0 / 50|0 / 50|1 / 50|0 / 47|4 / 49|
> |CompGen-OOD3|1 / 49|0 / 50|13 / 50|0 / 48|15 / 50|
> |EnvGen_GridSize|13 / 50|1 / 50|24 / 50|5 / 45|36 / 49|
> |EnvGen_Objects|13 / 50|4 / 50|15 / 50|5 / 49|39 / 50|
>
> ## Result Interpretations
>
> Even SOTA models like Gemini 3.5 Pro and OpenAI o3 exhibit the exact failure mode COGITAO highlights. While they achieve reasonable performance on In-Distribution (ID) tasks and Environmental Generalization (size/objects), their performance drops precipitously on Out-of-Distribution (OOD) composition. This confirms that COGITAO is not only solved by scale; it exposes a fundamental reasoning gap in how current architectures decompose and recompose information.

---

> ### Author Response · Authors · 2025-12-03
>
> # W3: Evaluation Metrics
>
> **Why Grid Accuracy (GA) is appropriate and not “too strict”**. GA is the standard metric in ARC-like benchmarks (2,3,4) where tasks operate in low-noise, low-dimensional, discretized environments with precise spatial structure. In these settings, a single incorrect cell can alter object identity, break a transformation, or invalidate a rule. Because COGITAO tasks aim to test whether models have inferred the correct underlying transformation rather than approximated it, GA usefully distinguishes genuine rule recovery and understanding from correlation-based approximations. This aligns with the cognitive motivation of the benchmark: if a rule is understood, the model should produce the exact outcome of applying the transformation for any unseen grid.
>
> However, we acknowledge the strictness of "Exact Grid Match." To address this, we have expanded our evaluation suite to include complementary metrics, reported in the Appendix:
>
> * Per-Pixel Accuracy (PPA): evaluates similarity of prediction to target at the pixel level.
> * Object-PPA: PPA but restricted to object pixels (values 1–9).
> * ID-Predicted % "Stubborness": It calculates the percentage of OOD tasks where the model predicted a transformation sequence seen during training (ID) instead of the correct novel sequence.
>
> These additional metrics capture nuanced behaviors (e.g., high PPA but low Object-PPA suggesting object-structure failures, or vice versa), and confirm when models are not guessing but remain constrained by what they learned ID. However, these insights do not yet yield strong global patterns across experiments, which we acknowledge, but they substantively inform our interpretation of model behavior.
>
> **How the chosen metrics collectively assess generalization:** Across settings, models retain high pixel-level precision while failing to reorganize familiar components into novel compositions--PPA remains high, Object-PPA moderately high (relatively), while GA collapses OOD (often near-zero). This highlights that models achieve some local correctness yet fail to recover the global transformation--a core capability COGITAO is designed to probe. The “stubbornness” meta metric further shows that OOD predictions frequently mirror ID habits rather than the specified atomic transformations or their order. This distinction is critical: GA strictly identifies whether meaningful compositional reasoning occurred, while PPA and Object-PPA clarify how it failed.
>
> Importantly, in the newly added Experiment 0 (translation-only, see response to Q1)--which is designed in its configuration to be easier than all the the other CompGen experiments--models do show signs of OOD generalization through OOD performance of 15-50% compared to the usual near 0%, even though a significant ID to OOD drop indicating a difficulty in extrapolation remains. This confirms that our experimental framework is sound and that the failures in the other, slightly more difficult experiments--for which training regime scaling did not improve performance--does reflect genuine limitations in compositional reasoning.
>
> **Performance degradation with sequence length.** In CompGen C3, depth-1 transformations are used for training/ID testing and depth-2 for OOD testing. Performance degrades sharply from high ID scores to near-zero OOD GA. Because collapse already occurs at depth-2, deeper compositions would not currently provide informative signal. Moreover, increasing depth for ID-only evaluation would not assess compositional reasoning, as models may continue to rely on pattern matching rather than decomposition and composition of transformations.
>
> # W4: Human Evaluation
>
> While formal human baselines are not included, the tasks are designed based on atomic operations (rotation, translation) that are intuitively solvable by humans, unlike the complex abstract pattern matching in some PGM configurations. But we agree with the reviewer that it would be an interesting addition, which we leave for future work.

---

> ### Author Response · Authors · 2025-12-03
>
> # Q1 & Q2: Dataset Difficulty and Learning Regime
>
> We thank the reviewer for these two relevant questions.
>
> We stopped at 10 epochs in the main paper because learning had plateaued. Furthermore, we observed that training on more than 100k samples did not impact the generalization performance - this is particularly relevant since all transformations are individually perfectly learnable at ~10k samples (see Appendix E - Transformations Learnability).
>
> To answer the reviewer's concerns and empirically support our decision, we ran the following two supplementary experiments:
> 1. Scaling Experiments with the Grid-Transformer, increasing dataset size to 1 Million samples and training for up to 100 epochs.
> 2. We added a new experiment across settings and regimes: "Experiment 0". It is desgined to be easier than the other experiments by being formed of a composition of translation-only transformations (e.g., translate_left), which proved to be easily learnable individually (only necessiting ~1k samples as per the Transformation Learnability results (Appendix E)). This experiment 0, as in the aforementionned scaled experiment, is also run with our Grid-TF on different splits up to 1 millions samples with training for up to 100 epochs.
>
> Below is a summary of our results (see appendix H for full details):
>
> | Setting | Exp | #Samples | #Epochs | GA ID | GA OOD | PPA ID | PPA OOD | Obj-PPA ID | Obj-PPA OOD | "Stubborness" |
> |---------|-----|----------|---------|-------|--------|--------|---------|-------------|-------------|--------------|
> | **CompGen-1** |||||||||||
> | | 0 | 100k | 100 | 0.780 | 0.360 | 0.983 | 0.996 | 0.859 | 0.962 | 0.000 |
> | | 0 | 250k | 50  | 0.713 | 0.507 | 0.982 | 0.997 | 0.860 | 0.970 | 0.000 |
> | | 0 | 1M   | 10  | 0.764 | 0.298 | 0.983 | 0.994 | 0.857 | 0.923 | 0.000 |
> ||||||||||||
> | | 1 | 100k | 100 | 0.537 | 0.000 | 0.984 | 0.916 | 0.836 | 0.293 | 0.118 |
> | | 1 | 250k | 50  | 0.779 | 0.001 | 0.994 | 0.943 | 0.927 | 0.462 | 0.166 |
> | | 1 | 1M   | 10  | 0.773 | 0.000 | 0.994 | 0.916 | 0.922 | 0.298 | 0.733 |
> ||||||||||||
> | **CompGen-3** |||||||||||
> | | 0 | 100k | 100 | 0.791 | 0.178 | 0.985 | 0.940 | 0.876 | 0.503 | 0.888 |
> | | 0 | 250k | 50  | 0.752 | 0.083 | 0.985 | 0.933 | 0.879 | 0.464 | 0.741 |
> | | 0 | 1M   | 10  | 0.600 | 0.302 | 0.975 | 0.955 | 0.770 | 0.617 | 0.751 |
> ||||||||||||
> | | 1 | 100k | 100 | 0.449 | 0.059 | 0.981 | 0.965 | 0.711 | 0.494 | 0.307 |
> | | 1 | 250k | 50  | 0.998 | 0.012 | 1.000 | 0.965 | 1.000 | 0.486 | 0.524 |
> | | 1 | 1M   | 10  | 0.462 | 0.036 | 0.983 | 0.965 | 0.758 | 0.490 | 0.514 |
>
> Models still fail to systematically compose simple translations OOD compared to ID. Increasing sample size by 10× does not close the generalization gap. The low OOD performance is not a result of insufficient data or under-training, but rather a structural inability of the architecture to generalize compositionally. This is further evident in Experiment 0 when looking at the remaining sharp drop of OOD performance of models to compose despite the simplicity of the translation transformations.
>
> Looking at the model stubborness metric, we can see that indeed, in many cases, the model fails to correctly predict OOD as it still tries to predict in-distribution sequences - highlighting that there is no compositional generalization capacity in the model. We believe both these news results further reinforce the validity of the current training regime for the intended analysis.
>
> # Q3: Models used
>
> We focused on Transformers (Vanilla, Grid, Looped, LLaDA) and ResNets as they represent the dominant paradigm in current vision and language modeling. While Relational Networks (RNs) are indeed effective for explicit relation modeling (e.g., in PGM), our goal was to establish a baseline for the architectures powering the current wave of foundation models. We agree that evaluating RNs is a valuable direction for future work.
>
> # References
>
> 1. Lake, Brenden, and Marco Baroni. "Generalization without systematicity: On the compositional skills of sequence-to-sequence recurrent networks." In International conference on machine learning, pp. 2873-2882. PMLR, 2018.
> 2. Chollet, François. "On the measure of intelligence."
> 3. Li, Wenhao, Yudong Xu, Scott Sanner, and Elias Boutros Khalil. "Tackling the abstraction and reasoning corpus with vision transformers: the importance of 2d representation, positions, and objects."
> 4. Jolicoeur-Martineau, Alexia. "Less is more: Recursive reasoning with tiny networks."

---

### Meta-Review · Area_Chair_eUBm · 2026-01-04

**Summary:**

The main contribution of this paper is the newly proposed dataset. The dataset can be used to evaluate the compositional generalization capability of a vision model. Different from other related benchmarks, this paper mainly focuses on the generalization capability of compositional transforms.

The evaluation metric used in the original version of the paper is not propriate. One reviewer points out this and the authors promised to add a new set of metrics into the paper. However, as these new metrics will significantly change the experimental results, if we evaluated the paper by considering the new experiments, it may not be fair to other papers. Rebuttal is mainly for clarification of reviewers potential mis-understanding rather than adding a set of brand new experiments.

Based on this, this paper may not be included in ICLR26, and, I encourage the authors to resubmit it to other venues.

**Reviewer Concerns:**

The authors have conducted In-Context Learning (ICL) experiments on frontier models to provide a reference baseline per Reviewer Eqxe
They added a set of new metrices and related experiments to address reviewer Eqxe’s question.

**Reviewer Scores:**

Reviewer Eqxe gives a set of good suggestions.

---

### Decision · Program_Chairs · 2026-01-26

Reject